# DECENTRALIZED NONSMOOTH NONCONVEX OPTIMIZATION WITH CLIENT SAMPLING

## ABSTRACT

This paper considers decentralized nonsmooth nonconvex optimization problem with Lipschitz continuous local functions. We propose an efficient stochastic first-order method with client sampling, achieving the $(\delta, \epsilon)$-Goldstein stationary point with the overall sample complexity of $\mathcal{O}(\delta^{-1}\epsilon^{-3})$, the computation rounds of $\mathcal{O}(\delta^{-1}\epsilon^{-3})$, and the communication rounds of $\tilde{\mathcal{O}}(\gamma^{-1/2}\delta^{-1}\epsilon^{-3})$, where $\gamma$ is the spectral gap of the mixing matrix for the network. Our results achieve the optimal sample complexity and the sharper communication complexity than existing methods. We also extend our ideas to zeroth-order optimization. Moreover, the numerical experiments show the empirical advantage of our methods.

## 1 INTRODUCTION

The large scale nonsmooth nonconvex optimization covers many applications in fields such as machine learning (Nair & Hinton, 2010; Xiao et al., 2024), statistics (Fan & Li, 2001; Zhang, 2010a;b), and economics (Duffie, 2010; Stadtler, 2014). In this paper, we focus on the decentralized stochastic optimization problem

$$\min_{\mathbf{x}\in\mathbb{R}^d} f(\mathbf{x}) \triangleq \frac{1}{n}\sum_{i=1}^{n} f_i(\mathbf{x}) \tag{1}$$

over the network with $n$ clients, where the local function at the $i$th client has the form of

$$f_i(\mathbf{x}) \triangleq \mathbb{E}_{\boldsymbol{\xi}_i \sim \mathcal{D}_i}[F_i(\mathbf{x}; \boldsymbol{\xi}_i)] \tag{2}$$

such that the stochastic component $F_i(\,\cdot\,; \boldsymbol{\xi}_i)$ is Lipschitz continuous but possibly nonconvex nonsmooth and the random index $\boldsymbol{\xi}_i \in \Xi_i$ follows the distribution $\mathcal{D}_i$. It is well known that achieving approximate stationary points in terms of the classical Clarke subdifferential (Clarke, 1990) for the general Lipschitz continuous function is intractable (Jordan et al., 2022; Kornowski & Shamir, 2021; Tian & So, 2024; Zhang et al., 2020). Instead, we typically target to find $(\delta, \epsilon)$-Goldstein stationary points (Zhang et al., 2020). This criterion suggests studying the convex hull of Clarke subdifferential at points in the $\delta$-radius neighborhood of the given point.

The stochastic optimization methods for finding $(\delta, \epsilon)$-Goldstein stationary points in non-distributed setting have been widely studied in recent years (Chen et al., 2023; Cutkosky et al., 2023; Davis et al., 2022; Kornowski & Shamir, 2024; Lin et al., 2022; Tian et al., 2022; Zhang et al., 2020). Specifically, Tian et al. (2022); Zhang et al. (2020) proposed the (perturbed) stochastic interpolated normalized gradient descent with the stochastic first-order oracle (SFO) complexity of $\mathcal{O}(\delta^{-1}\epsilon^{-4})$. In a seminal work, Cutkosky et al. (2023) established the conversion from nonsmooth nonconvex optimization to online learning, achieving the SFO complexity of $\mathcal{O}(\delta^{-1}\epsilon^{-3})$. They also extends the lower bound of Arjevani et al. (2023) to show their SFO complexity is optimal. Another line of research is the zeroth-order optimization. Lin et al. (2022) applied the randomized smoothing (Duchi et al., 2012; Nesterov & Spokoiny, 2017; Shamir, 2017; Yousefian et al., 2012) to design the gradient-free method with the stochastic zeroth-order oracle (SZO) complexity of $\mathcal{O}(d^{3/2}\delta^{-1}\epsilon^{-4})$. Later, Chen et al. (2023) improve this result by incorporating variance reduction techniques (Cutkosky & Orabona, 2019; Fang et al., 2018; Huang et al., 2022; Ji et al., 2019; Levy et al., 2021; Liu et al., 2018; Nguyen et al., 2017; Pham et al., 2020; Wang et al., 2019), achieving the SZO complexity of $\mathcal{O}(d^{3/2}\delta^{-1}\epsilon^{-3})$. Recently, Kornowski & Shamir (2024) established the optimal dimension-dependence SZO complexity of $\mathcal{O}(d\delta^{-1}\epsilon^{-3})$ based on the inclusion property of Goldstein subdifferential.

In decentralized setting, we have to consider the consensus error among the variables on different clients in the network. The popular technique of gradient tracking can successfully bound the consensus error in for smooth optimization problems (Nedic & Ozdaglar, 2009; Qu & Li, 2017; Shi et al., 2015), while it cannot be directly used in the nonsmooth setting since the Lipschitz continuity of the gradient (subgradient) may not hold. Kovalev et al. (2024); Lan et al. (2020) proposed efficient decentralized algorithms based on the primal-dual framework for the nonsmooth objective but only limit to the convex problem. A natural idea for decentralized nonsmooth optimization is using the randomized smoothing to establish the smooth surrogate for the original problem, which works for both the convex (Scaman et al., 2018) and the nonconvex settings (Lin et al., 2024). For example, Lin et al. (2024) extended gradient-free methods (Chen et al., 2023; Lin et al., 2022) for decentralized stochastic nonsmooth nonconvex problem, while their SZO complexity bounds depend on the term of $\mathcal{O}(d^{3/2})$, which does not match the best-known zeroth-order method in non-distributed scenarios (Kornowski & Shamir, 2024). Later, Sahinoglu & Shahrampour (2024) proposed multi-epoch decentralized online learning (ME-DOL) method for both first-order and zeroth-order decentralized stochastic stochastic nonsmooth nonconvex optimization, which incorporates the decentralized online mirror descent (Shahrampour & Jadbabaie, 2017) into the online-to-nonconvex conversion (Cutkosky et al., 2023; Kornowski & Shamir, 2024). The ME-DOL with SFO can find $(\delta, \epsilon)$-Goldstein stationary point with the computation and the communication rounds of $\mathcal{O}(n\gamma^{-2}\delta^{-1}\epsilon^{-3})$, and the ME-DOL with SZO requires the computation rounds and the communication rounds of $\mathcal{O}(nd\gamma^{-2}\delta^{-1}\epsilon^{-3})$, where $\gamma \in (0, 1]$ is the spectral gap of the mixing matrix associated with the network.

The objective in distributed optimization problem (1) naturally has the finite-sum structure in the view of local functions $\{f_i\}_{i=1}^n$. This motivates us to design the partial participated methods, which performs the client sampling during iterations and only executes the computation/communication on the selected clients (Chen et al., 2020; Maranjyan et al., 2022; Mishchenko et al., 2022). Some recent works (Bai et al., 2024; Liu et al., 2024; Luo et al., 2022) studied partial participated methods by considering the balance among the first-order oracle complexity, the computation rounds, and the communication rounds in decentralized optimization. However, these results heavily depend on the smoothness assumptions. To the best of our knowledge, all existing methods (Chen et al., 2020; Kovalev et al., 2024; Lan et al., 2020; Lin et al., 2024; Sahinoglu & Shahrampour, 2024; Wang et al., 2023; Zhang et al., 2024) for decentralized nonsmooth optimization require all clients accessing their local oracle in per computation rounds, which limits the sampling efficiency.

In this paper, we propose the Decentralized Online-to-nonconvex Conversion with Client Sampling (DOC$^2$S), which integrates the partial participated computation and the multi-consensus steps into decentralized optimization. We show that DOC$^2$S with local stochastic first-order oracle (LSFO) can achieve the $(\delta, \epsilon)$-Goldstein stationary points with the total LSFO calls of $\mathcal{O}(\delta^{-1}\epsilon^{-3})$, the computation rounds of $\mathcal{O}(\delta^{-1}\epsilon^{-3})$, and the communication rounds of $\tilde{\mathcal{O}}(\gamma^{-1/2}\delta^{-1}\epsilon^{-3})$. All of these upper bounds are sharper than the state-of-the-arts results achieved by ME-DOL (Sahinoglu & Shahrampour, 2024). Recall that ME-DOL requires the computation rounds of $\mathcal{O}(\gamma^{-2}\delta^{-1}\epsilon^{-3})$ and each of its computation round requires all clients to access their local stochastic gradient, which leads to the total LSFO calls of $\mathcal{O}(n\gamma^{-2}\delta^{-1}\epsilon^{-3})$. In contrast, the total LSFO complexity of our DOC$^2$S does not depend on the number of clients $n$ and spectral gap $\gamma$. Additionally, we also show that DOC$^2$S with local stochastic zeroth-order oracle (LSZO) can achieve the $(\delta, \epsilon)$-Goldstein stationary points with the total LSZO calls of $\mathcal{O}(d\delta^{-1}\epsilon^{-3})$, the computation rounds of $\mathcal{O}(d\delta^{-1}\epsilon^{-3})$, and the communication rounds of $\tilde{\mathcal{O}}(d\gamma^{-1/2}\delta^{-1}\epsilon^{-3})$, also improving the results of ME-DOL (Sahinoglu & Shahrampour, 2024). We summarize theoretical results of our methods and related work in Table 1.

## 2 PRELIMINARIES

In this section, we formalize our problem setting and introduce the background of nonsmooth analysis.

### 2.1 NOTATION AND ASSUMPTIONS

We use $\|\cdot\|$ and $\|\cdot\|_2$ to denote the Frobenius norm and the spectral norm of the matrix, respectively, also the Euclidean norm of the vector. We let $\mathbf{1}_n = [1, \ldots, 1]^\top \in \mathbb{R}^n$ and $\mathbf{I}$ be the identity matrix. The notation $\mathrm{conv}(\cdot)$ denotes the convex hull of given set. Additionally, we use notations $\mathbb{B}^d(\mathbf{x}, \delta)$ and $\mathbb{S}^{d-1}$ to present the Euclidean ball centered at $\mathbf{x} \in \mathbb{R}^d$ with radius $\delta > 0$ and the unit sphere centered at the origin, respectively.

Table 1: We present the upper complexity bounds of our methods and related work for finding $(\delta, \epsilon)$-Goldstein stationary points in stochastic decentralized optimization problem. The sample complexity refers to the overall LSFO/LZSO complexity on all $n$ clients.

| Oracle | Methods | Sample Complexity | Computation Rounds | Communication Rounds |
|---|---|---|---|---|
| 1st | §ME-DOL 
 (Sahinoglu & Shahrampour, 2024) | $\mathcal{O}\left(\dfrac{n^2}{\gamma^2\delta\epsilon^3}\right)$ | $\mathcal{O}\left(\dfrac{n}{\gamma^2\delta\epsilon^3}\right)$ | $\mathcal{O}\left(\dfrac{n}{\gamma^2\delta\epsilon^3}\right)$ |
| 1st | DOC²S 
 Theorem 1 | $\mathcal{O}\left(\dfrac{1}{\delta\epsilon^3}\right)$ | $\mathcal{O}\left(\dfrac{1}{\delta\epsilon^3}\right)$ | $\tilde{\mathcal{O}}\left(\dfrac{1}{\gamma^{1/2}\delta\epsilon^3}\right)$ |
| 0th | †DGFM 
 (Lin et al., 2024) | $\mathcal{O}\left(\dfrac{nd^{3/2}}{\gamma^p\delta\epsilon^4}\right)$ | $\mathcal{O}\left(\dfrac{d^{3/2}}{\gamma^p\delta\epsilon^4}\right)$ | $\mathcal{O}\left(\dfrac{d^{3/2}}{\gamma^p\delta\epsilon^4}\right)$ |
| 0th | †DGFM$^+$ 
 (Lin et al., 2024) | $\mathcal{O}\left(\dfrac{n^{3/2}d^{1/2}}{\delta\epsilon^2}\left(1+\dfrac{d}{n\epsilon}\right)\right)$ | $\mathcal{O}\left(\dfrac{n^{1/2}d^{1/2}}{\delta\epsilon^2}\left(1+\dfrac{d}{n\epsilon}\right)\right)$ | $\mathcal{O}\left(\dfrac{n^{1/2}d^{1/2}}{\gamma^q\delta\epsilon^2}\right)$ |
| 0th | §ME-DOL 
 (Sahinoglu & Shahrampour, 2024) | $\mathcal{O}\left(\dfrac{n^2d}{\gamma^2\delta\epsilon^3}\right)$ | $\mathcal{O}\left(\dfrac{nd}{\gamma^2\delta\epsilon^3}\right)$ | $\mathcal{O}\left(\dfrac{nd}{\gamma^2\delta\epsilon^3}\right)$ |
| 0th | DOC²S 
 Theorem 3 | $\mathcal{O}\left(\dfrac{d}{\delta\epsilon^3}\right)$ | $\mathcal{O}\left(\dfrac{d}{\delta\epsilon^3}\right)$ | $\tilde{\mathcal{O}}\left(\dfrac{d}{\gamma^{1/2}\delta\epsilon^3}\right)$ |

†The dependency on $\gamma$ in the complexity of DGFM and DGFM$^+$ is not provided explicitly (Lin et al., 2024).

§The complexity of ME-DOL (Sahinoglu & Shahrampour, 2024) contains additional dependency on $n$. Please refer to Appendix D for details.

We impose following assumptions for formulations (1)–(2).

**Assumption 1.** We suppose each stochastic component $F_i(\mathbf{x}, \boldsymbol{\xi}_i)$ is $L(\boldsymbol{\xi}_i)$-Lipschitz continuous in $\mathbf{x}$ for given $\boldsymbol{\xi}_i \in \Xi_i$, i.e., it holds that $|F_i(\mathbf{x}; \boldsymbol{\xi}_i) - F_i(\mathbf{y}; \boldsymbol{\xi}_i)| \leq L(\boldsymbol{\xi}_i)\|\mathbf{x} - \mathbf{y}\|$, for all $\mathbf{x}, \mathbf{y} \in \mathbb{R}^d$ and $i \in [n]$. Furthermore, we suppose each $L(\boldsymbol{\xi}_i)$ has a bounded second moment, i.e., there exists $L > 0$ such that $\mathbb{E}_{\boldsymbol{\xi}_i}[L(\boldsymbol{\xi}_i)^2] \leq L^2$ for all $i \in [n]$.

**Assumption 2.** We suppose the objective function $f : \mathbb{R}^d \to \mathbb{R}$ is lower bounded by $f^*$, i.e., it holds $f^* \triangleq \inf_{\mathbf{x} \in \mathbb{R}^d} f(\mathbf{x}) > -\infty$.

We make the following assumption for the local stochastic first-order oracle (LSFO).

**Assumption 3.** We suppose the algorithms can access the local function $f_i : \mathbb{R}^d \to \mathbb{R}$ via the LSFO consisting of local gradient estimator $F_i : \mathbb{R}^d \times \Xi_i \to \mathbb{R}$ and the random index $\boldsymbol{\xi}_i \sim \mathcal{D}_i$ such that $\mathbb{E}_{\boldsymbol{\xi}_i \sim \mathcal{D}_i}[\nabla F_i(\mathbf{x}; \boldsymbol{\xi}_i)] = \nabla f_i(\mathbf{x})$ and $\mathbb{E}_{\boldsymbol{\xi}_i \sim \mathcal{D}_i}[\|\nabla F_i(\mathbf{x}; \boldsymbol{\xi}_i) - \nabla f_i(\mathbf{x})\|^2] \leq \sigma^2$ for some $\sigma \geq 0$. We further suppose there exists some $G \geq 0$ such that $\mathbb{E}_{\boldsymbol{\xi}_i \sim \mathcal{D}_i}[\|\nabla F_i(\mathbf{x}; \boldsymbol{\xi}_i)\|^2] \leq G^2$ for all $\mathbf{x} \in \mathbb{R}^d$ and $\boldsymbol{\xi}_i \in \Xi_i$.

Rademacher's theorem (Evans, 2018) states the Lipschitz continuous function is differentiable almost everywhere. Thus, the LSFO is well-defined almost everywhere under Assumption 1. Besides, we also consider the local stochastic zeroth-order oracle (LSZO) as follows.

**Assumption 4.** We suppose the algorithms can access the local function $f_i : \mathbb{R}^d \to \mathbb{R}$ via the LSZO consisting of local function value estimator $F_i : \mathbb{R}^d \times \Xi_i \to \mathbb{R}$ and the random index $\boldsymbol{\xi}_i \sim \mathcal{D}_i$ such that $\mathbb{E}_{\boldsymbol{\xi}_i \sim \mathcal{D}_i}[F_i(\mathbf{x}; \boldsymbol{\xi}_i)] = f_i(\mathbf{x})$.

We aim for all $n$ clients in the network to collaborate in solving stochastic decentralized optimization problem (1). We use the doubly stochastic matrix $\mathbf{P} = [p_{ij}] \in \mathbb{R}^{n \times n}$ to describe the topology of the network. Specifically, the communication step at the $i$th client is built upon the weighted average $\mathbf{x}_i^+ = \sum_{j=1}^{n} p_{ij}\mathbf{x}_j$, where $\mathbf{x}_j$ is the local variable on the $j$th client. We impose the following standard assumption in decentralized optimization for the matrix $\mathbf{P} \in \mathbb{R}^{n \times n}$ (Schmidt et al., 2017; Scaman et al., 2018).

**Assumption 5.** We suppose that the mixing matrix $\mathbf{P} \in \mathbb{R}^{n \times n}$ associated with the network satisfies: (a) The matrix $\mathbf{P} \in \mathbb{R}^{n \times n}$ is symmetric and holds $p_{ij} \geq 0$ for all $i, j \in [n]$; (b) It holds $p_{ij} \neq 0$ if and only if clients $i$ and $j$ are connected or $i = j$; (c) It holds $\mathbf{0} \preceq \mathbf{P} \preceq \mathbf{I}$ and $\mathbf{P}^\top \mathbf{1}_n = \mathbf{P}\mathbf{1}_n = \mathbf{1}_n$.

Under Assumption 5, the largest eigenvalue of the mixing matrix $\mathbf{P} \in \mathbb{R}^{n \times n}$ is one. Consequently, we define the spectral gap of $\mathbf{P} \in \mathbb{R}^{n \times n}$ as $\gamma = 1 - \lambda_2(\mathbf{P}) \in (0, 1]$, where $\lambda_2(\mathbf{P})$ is the second largest eigenvalue of $\mathbf{P}$.

## 2.2 GOLDSTEIN STATIONARY POINTS

We present the notion of Clarke subdifferential (Clarke et al., 2008) and its relaxation Goldstein subdifferential (Goldstein, 1977) for the Lipschitz continuous objective in the nonconvex nonsmooth problem as follows.

**Definition 1** (Clarke et al. (2008)). The Clarke subdifferential of a Lipschitz continuous function $f : \mathbb{R}^d \to \mathbb{R}$ at a point $\mathbf{x} \in \mathbb{R}^d$ is defined by $\partial f(\mathbf{x}) := \mathrm{conv}\{\mathbf{g} : \mathbf{g} = \lim_{\mathbf{x}_k \to \mathbf{x}} \nabla f(\mathbf{x}_k)\}$.

**Definition 2** (Goldstein (1977)). For given $\delta \geq 0$ and a Lipschitz continuous function $f : \mathbb{R}^d \to \mathbb{R}$, the Goldstein $\delta$-subdifferential of at point $\mathbf{x} \in \mathbb{R}^d$ is defined by $\partial_\delta f(\mathbf{x}) := \mathrm{conv}\big(\cup_{\mathbf{y} \in \mathbb{B}^d(\mathbf{x}, \delta)} \partial f(\mathbf{y})\big)$, where the $\partial f(\mathbf{y})$ is Clarke subdifferential.

We are interested in finding the $(\delta, \epsilon)$-Goldstein stationary point (Zhang et al., 2020), which is defined as follows.

**Definition 3** (Zhang et al. (2020)). For given Lipschitz continuous function $f : \mathbb{R}^d \to \mathbb{R}$, $\delta \geq 0$, and $\mathbf{x} \in \mathbb{R}^d$, we denote $\|\nabla f(\mathbf{x})\|_\delta := \min\{\|\mathbf{g}\| : \mathbf{g} \in \partial_\delta f(\mathbf{x})\}$. We call the point $\mathbf{x}$ a $(\delta, \epsilon)$-Goldstein stationary point of $f$ if $\|\nabla f(\mathbf{x})\|_\delta \leq \epsilon$ holds.

## 2.3 RANDOMIZED SMOOTHING

Randomized smoothing is a popular technique in stochastic optimization (Duchi et al., 2012; Lin et al., 2022; Nesterov & Spokoiny, 2017; Shamir, 2017; Yousefian et al., 2012). This paper focuses on the uniform smoothing as follows.

**Definition 4** (Yousefian et al. (2012)). Given a Lipschitz continuous function $f : \mathbb{R}^d \to \mathbb{R}$, we denote its smooth surrogate as $f_\delta(\mathbf{x}) \triangleq \mathbb{E}_{\mathbf{u} \sim \mathrm{Unif}(\mathbb{B}^d(0,1))}[f(\mathbf{x} + \delta\mathbf{u})]$, where $\mathrm{Unif}(\mathbb{B}^d(0, 1))$ is the uniform distribution on the unit Euclidean ball centered at the origin.

The smooth surrogate $f_\delta$ has the following properties.

**Proposition 1** (Lin et al. (2022, Proposition 2.2), Kornowski & Shamir (2024, Lemma 4)). *Suppose the function $f : \mathbb{R}^d \to \mathbb{R}$ is $L$-Lipschitz, then its smooth surrogate $f_\delta$ holds: (a) $|f_\delta(\cdot) - f(\cdot)| \leq \delta L$; (b) $f_\delta(\cdot)$ is $L$-Lipschitz; (c) $f_\delta(\cdot)$ is differentiable with $c_0\sqrt{d}L\delta^{-1}$-Lipschitz gradients for some numeric constant $c_0 > 0$; (d) $\partial_\mu f_\delta(x) \subseteq \partial_{\mu+\delta} f(x)$ for all $\mu \geq 0$.*

Based on Proposition 1, we can establish unbiased gradient estimators for the smooth surrogate of local functions, which is shown in the following lemma.

**Lemma 1** (Kornowski & Shamir (2024, Lemma 7)). *We let $\mathbf{w} = \mathbf{x} + s\mathbf{\Delta}$ with $s \sim \mathrm{Unif}([0, 1])$ and given $\mathbf{x}, \mathbf{\Delta} \in \mathbb{R}^d$. Under Assumptions 1 and 4, the random vector*

$$\mathbf{g}_i = \frac{d}{2\delta}\Big(F_i(\mathbf{x} + s\mathbf{\Delta} + \delta\mathbf{z}_i; \boldsymbol{\xi}_i) - F_i(\mathbf{x} + s\mathbf{\Delta} - \delta\mathbf{z}_i; \boldsymbol{\xi}_i)\Big)\mathbf{z}_i$$

*with $\mathbf{z}_i \sim \mathrm{Unif}(\mathbb{S}^{d-1})$, $\boldsymbol{\xi}_i \sim \mathcal{D}_i$, and given $\delta \geq 0$ holds that $\mathbb{E}_{\boldsymbol{\xi}_i, \mathbf{z}_i}[\mathbf{g}_i \,|\, s] = \nabla(f_i)_\delta(\mathbf{w})$ and $\mathbb{E}_{\boldsymbol{\xi}_i, \mathbf{z}_i}\big[\|\mathbf{g}_i\|^2 \,|\, s\big] \leq 16\sqrt{2\pi}dL^2$ for all $i \in [n]$.*

## 3 THE ALGORITHM AND MAIN RESULTS

We propose decentralized online-to-nonconvex conversion with client sampling (DOC$^2$S) in Algorithm 1, which incorporates the steps of client sampling and Chebyshev acceleration (Algorithm 2) (Arioli & Scott, 2014; Liu & Morse, 2011; Song et al., 2024; Ye et al., 2023) into the framework of online-to-nonconvex conversion (Cutkosky et al., 2023; Sahinoglu & Shahrampour, 2024) to improve both the computation and the communication efficiency. Furthermore, our DOC$^2$S (Algorithm 1) supports both the first-order and the zeroth-order oracles through subroutines of Algorithms 3 and 4. Following the design of Cutkosky et al. (2023), the double-loop structure in DOC$^2$S can be regarded as minimizing the $K$-shifting regret, i.e., $R_T(\mathbf{u}^1, \ldots, \mathbf{u}^K) = \sum_{k=1}^{K} \sum_{t=1}^{T} \langle \mathbf{g}_{i^t}^{k,t}, \bar{\mathbf{\Delta}}^{k,t-1/2} - \mathbf{u}^k \rangle$ for an arbitrary sequence of $K$ vectors $\mathbf{u}^1, \ldots, \mathbf{u}^K \in \{\mathbf{u} : \|\mathbf{u}\| \leq D\}$ that changes every $T$ iterations. We desire that the algorithm guarantees every $T$ iterations can achieve a shifting regret of $\mathcal{O}(K\sqrt{T})$, where $K = \mathcal{O}(\epsilon^{-1})$ and $T = \mathcal{O}(\epsilon^{-2})$.

---

**Algorithm 1** Decentralized Online-to-Nonconvex Conversion with Client Sampling (DOC$^2$S)

1: **Input:** OracleType $\in$ {0th, 1st}, $\delta' \geq 0$, $K, T, R \in \mathbb{N}$, $\eta, D > 0$, $\mathbf{P} \in \mathbb{R}^{n \times n}$

2: **Initialization:** $\mathbf{y}_i^{0,T} = \mathbf{0}$ for all $i \in [n]$

3: **for** $k = 1$ **to** $K$ **do**

4:     **parallel for** $i = 1$ **to** $n$

5:       $\boldsymbol{\Delta}_i^{k,1/2} = \mathbf{0}$,   $\mathbf{y}_i^{k,0} = \mathbf{y}_i^{k-1,T}$

6:     **end parallel for**

7:     **for** $t = 1$ **to** $T$ **do**

8:       $i^t \sim \text{Unif}(\{1, \ldots, n\})$

9:       **parallel for** $i = 1$ **to** $n$

10:         $\mathbf{x}_i^{k,t} = \begin{cases} \mathbf{y}_i^{k,t-1} + n\boldsymbol{\Delta}_i^{k,t-1/2}, & i = i^t \\ \mathbf{y}_i^{k,t-1}, & i \neq i^t \end{cases}$

11:         $s_i^{k,t} \sim \text{Unif}([0,1])$,   $\mathbf{w}_i^{k,t} = \mathbf{y}_i^{k,t-1} + s_i^{k,t} \boldsymbol{\Delta}_i^{k,t-1/2}$

12:       **end parallel for**

13:       $\{\mathbf{y}_i^{k,t}\}_{i=1}^n = \text{FastGossip}\left(\{\mathbf{x}_i^{k,t}\}_{i=1}^n, \mathbf{P}, R\right)$

14:       $\mathbf{g}_{i^t}^{k,t} = \begin{cases} \text{First-Order-Estimator}\left(F_{i^t}, \mathcal{D}_{i^t}, \mathbf{w}_{i^t}^{k,t}, \delta'\right), & \text{OracleType = 1th} \\ \text{Zeroth-Order-Estimator}\left(F_{i^t}, \mathcal{D}_{i^t}, \mathbf{w}_{i^t}^{k,t}, \delta'\right), & \text{OracleType = 0th} \end{cases}$

15:       **parallel for** $i = 1$ **to** $n$

16:         $\boldsymbol{\Delta}_i^{k,t} = \begin{cases} n \min\left\{1, \dfrac{D}{\|\boldsymbol{\Delta}_i^{k,t-1/2} - \eta\mathbf{g}_i^{k,t}\|}\right\}\left(\boldsymbol{\Delta}_i^{k,t-1/2} - \eta\mathbf{g}_i^{k,t}\right), & i = i^t \\ \mathbf{0}, & i \neq i^t \end{cases}$

17:       **end parallel for**

18:       $\{\boldsymbol{\Delta}_i^{k,t+1/2}\}_{i=1}^n = \text{FastGossip}\left(\{\boldsymbol{\Delta}_i^{k,t}\}_{i=1}^n, \mathbf{P}, R\right)$

19:     **end for**

20: **end for**

21: **Output:** $\mathbf{w}_i^{\text{out}} \sim \text{Unif}\left(\{\hat{\mathbf{w}}_i^1, \ldots, \hat{\mathbf{w}}_i^K\}\right)$ for all $i \in [n]$, where $\hat{\mathbf{w}}_i^k = \frac{1}{T}\sum_{t=1}^T \mathbf{w}_i^{k,t}$

---

The key idea of DOC$^2$S (Algorithm 1) is to perform the client sampling $i^t \sim \text{Unif}(\{1, \ldots, n\})$ at the beginning of each iteration (line 8). Consequently, the local oracle call (line 14) in the iteration is only required on the $i^t$th client, which significantly improve the sample complexity of existing decentralized nonconvex optimization methods that requires all $n$ clients perform the computation in each iteration (Chen et al., 2020; Kovalev et al., 2024; Lan et al., 2020; Lin et al., 2024; Sahinoglu & Shahrampour, 2024; Wang et al., 2023; Zhang et al., 2024). We also include the multi-consensus step with Chebyshev acceleration (Algorithm 2) (Arioli & Scott, 2014; Liu & Morse, 2011; Song et al., 2024; Ye et al., 2023) in iterations (lines 13 and 18 of Algorithm 1), which guarantees the consensus error of the local variables can be well bounded even if only one of the clients performs the local oracle calls in each iteration.

We present the main theoretical results for proposed DOC$^2$S (Algorithm 1) with the local stochastic first-order oracle (Algorithm 3) as follows.

**Theorem 1.** *Under Assumptions 1, 2, 3, and 5, Algorithm 1 with the local stochastic first-order oracle (Algorithm 3) by taking $\delta' = \delta/2$, $K = \mathcal{O}(\delta^{-1}\epsilon^{-1})$, $T = \mathcal{O}(\epsilon^{-2})$, $R = \tilde{\mathcal{O}}(\gamma^{-1/2})$, $\eta = \mathcal{O}(\delta\epsilon^3)$, and $D = \mathcal{O}(\delta\epsilon^2)$ can output $\{\mathbf{w}_i^{\text{out}}\}_{i=1}^n$ such that $\mathbb{E}\left[\|\nabla f(\mathbf{w}_i^{\text{out}})\|_\delta\right] \leq \epsilon$ for all $i \in [n]$.*

**Corollary 2.** *Following the setting of Theorem 1, each client can achieve an $(\delta, \epsilon)$-Goldstein stationary point of the objective within the overall stochastic first-order oracle complexity of $\mathcal{O}(\delta^{-1}\epsilon^{-3})$, the computation rounds of $\mathcal{O}(\delta^{-1}\epsilon^{-3})$, and the communication rounds of $\tilde{\mathcal{O}}(\gamma^{-1/2}\delta^{-1}\epsilon^{-3})$.*

Besides the sharper complexity bounds than ME-DOL (Sahinoglu & Shahrampour, 2024) (see comparison in Table 1), the proposed DOC$^2$S also guarantees every client can achieve a $(\delta, \epsilon)$-Goldstein stationary point in expectation. Recall that the theoretical analysis of ME-DOL (Sahinoglu & Shahram-

**Algorithm 2** FastGossip$\left(\left\{\mathbf{z}_i^{(0)}\right\}_{i=1}^n, \mathbf{P}, R\right)$

1: $\mathbf{z}_i^{(-1)} = \mathbf{z}_i^{(0)}$ for all $i \in [n]$

2: $\phi = \dfrac{1 - \sqrt{1 - (\lambda_2(\mathbf{P}))^2}}{1 + \sqrt{1 - (\lambda_2(\mathbf{P}))^2}}$

3: **parallel for** $r = 0$ **to** $R - 1$

4: $\quad \mathbf{z}_i^{(r+1)} = (1 + \phi) \sum_{j=1}^n p_{ij} \mathbf{z}_j^{(r)} - \phi \mathbf{z}_i^{(r-1)}$

5: **end parallel for**

6: **Output:** $\left\{\mathbf{z}_i^{(R)}\right\}_{i=1}^n$

---

**Algorithm 3** First-Order-Estimator$(F_i, \mathcal{D}_i, \mathbf{w}_i, \mu)$

1: $\boldsymbol{\xi}_i \sim \mathcal{D}_i, \quad \mathbf{z}_i \sim \text{Unif}(\mathbb{B}^d(0, 1))$

2: $\mathbf{g}_i = \nabla F_i(\mathbf{w}_i + \mu \mathbf{z}_i; \boldsymbol{\xi}_i)$

3: **Output:** $\mathbf{g}_i$

---

**Algorithm 4** Zeroth-Order-Estimator$(F_i, \mathcal{D}_i, \mathbf{w}_i, \mu)$

1: $\boldsymbol{\xi}_i \sim \mathcal{D}_i, \quad \mathbf{z}_i \sim \text{Unif}(\mathbb{S}^{d-1})$

2: $\mathbf{g}_i = \dfrac{d}{2\mu}(F_i(\mathbf{w}_i + \mu \mathbf{z}_i; \boldsymbol{\xi}_i) - F_i(\mathbf{w}_i - \mu \mathbf{z}_i; \boldsymbol{\xi}_i))\mathbf{z}_i$

3: **Output:** $\mathbf{g}_i$

---

pour, 2024, Theorem 2) only indicates that there exists some point $\bar{\mathbf{w}}^k = \frac{1}{nT} \sum_{t=1}^T \sum_{i=1}^n \mathbf{w}_i^{k,t}$ which is an $(\delta, \epsilon)$-Goldstein stationary point, where $\mathbf{w}_i^{k,t}$ is generated on the $i$th client. However, achieving such mean vector $\bar{\mathbf{w}}^k$ is non-trivial in practice for the decentralized setting. In contrast, the point $\mathbf{w}_i^{\text{out}}$ in our algorithm and theory only depends on its local variables (line 21 of Algorithm 1).

Similarly, we can also achieve the following results for the local stochastic zeroth-order oracle.

**Theorem 3.** *Under Assumptions 1, 2, 4, and 5, Algorithm 1 with the local stochastic zeroth-order oracle (Algorithm 4) by taking* $\delta' = \delta/2$, $K = \mathcal{O}(\delta^{-1}\epsilon^{-1})$, $T = \mathcal{O}(d\epsilon^{-2})$, $R = \tilde{\mathcal{O}}(\gamma^{-1/2})$, $\eta = \mathcal{O}(\delta\epsilon^3)$, *and* $D = \mathcal{O}(\delta\epsilon^2)$ *can output* $\{\mathbf{w}_i^{\text{out}}\}_{i=1}^n$ *such that* $\mathbb{E}\left[\|\nabla f(\mathbf{w}_i^{\text{out}})\|_\delta\right] \leq \epsilon$ *for all* $i \in [n]$.

**Corollary 4.** *Following the setting of Theorem 1, each client can achieve an* $(\delta, \epsilon)$-*Goldstein stationary point of the objective within the overall stochastic zeroth-order oracle complexity of* $\mathcal{O}(d\delta^{-1}\epsilon^{-3})$, *the computation rounds of* $\mathcal{O}(d\delta^{-1}\epsilon^{-3})$, *and the communication rounds of* $\tilde{\mathcal{O}}(d\gamma^{-1/2}\delta^{-1}\epsilon^{-3})$.

# 4 THE COMPLEXITY ANALYSIS

This section provides the brief sketch for the proofs of our main results and the details are deferred in supplementary materials. In the remains, we use the bold letter with a bar to denote the mean vector, e.g., $\bar{\mathbf{x}}^{k,t} = \frac{1}{n} \sum_{i=1}^n \mathbf{x}_i^{k,t}$, $\bar{\mathbf{y}}^{k,t} = \frac{1}{n} \sum_{i=1}^n \mathbf{y}_i^{k,t}$, and $\bar{\boldsymbol{\Delta}}^{k,t} = \frac{1}{n} \sum_{i=1}^n \boldsymbol{\Delta}_i^{k,t}$.

We first introduce the following proposition for the subroutine of multi-consensus steps with Chebyshev acceleration (Algorithm 2) (Ye et al., 2023).

**Proposition 2** (Ye et al. (2023, Proposition 1))**.** *For Algorithm 2, we denote* $\bar{\mathbf{z}} = \frac{1}{n} \sum_{i=1}^n \mathbf{z}_i^{(0)}$, *then it holds that* $\frac{1}{n} \sum_{i=1}^n \mathbf{z}_i^{(R)} = \bar{\mathbf{z}}$ *and*

$$\sum_{i=1}^n \|\mathbf{z}_i^{(R)} - \bar{\mathbf{z}}\|^2 \leq 14 \left(1 - \left(1 - \frac{1}{\sqrt{2}}\right)\sqrt{\gamma}\right)^{2R} \sum_{i=1}^n \|\mathbf{z}_i^{(0)} - \bar{\mathbf{z}}\|^2.$$

Based on Proposition 2, performing the gossip steps on variables $\{\boldsymbol{\Delta}_i^{k,t+1/2}\}_{i=1}^n$ and $\{\mathbf{y}_i^{k,t}\}_{i=1}^n$ (lines 13 and 18 of Algorithm 1) achieves the upper bounds for the consensus errors as follows.

**Lemma 2.** *Under Assumptions 3 and 5, Algorithm 1 with*

$$R \geq \left\lceil \frac{1}{(1 - 1/\sqrt{2})\sqrt{\gamma}} \log \frac{\sqrt{14n(n-1)}D}{\epsilon'} \right\rceil \tag{3}$$

*satisfies* $\|\boldsymbol{\Delta}_i^{k,t+1/2} - \bar{\boldsymbol{\Delta}}^{k,t+1/2}\| \leq \epsilon'$ *and* $\|\boldsymbol{\Delta}_i^{k,t+1/2}\| \leq D + \epsilon'$ *for all* $i \in [n]$ *and* $\epsilon' > 0$.

**Lemma 3.** *Under the setting of Lemma 2, Algorithm 1 holds*

$$\|\bar{\mathbf{y}}^{k,t} - \mathbf{y}_i^{k,t}\| \leq \frac{(D + \epsilon')\epsilon'}{D - \epsilon'},$$

*for all* $i \in [n]$ *and* $\epsilon' < D$.

We first consider the decrease of the objective function value at the mean vector after one epoch in the smooth case. The update rule of Algorithm 1 indicates

$$
\mathbb{E}[f(\bar{\mathbf{x}}^{k,T}) - f(\bar{\mathbf{x}}^{k,0})]
$$

$$
= \sum_{t=1}^{T} \mathbb{E}_{i^t} \mathbb{E}_{\mathcal{S}^{k,t}, \boldsymbol{\xi}_{i^t}^{k,t}} [\langle \boldsymbol{\nabla}^{k,t}, \boldsymbol{\Delta}_{i^t}^{k,t-1/2} - \bar{\boldsymbol{\Delta}}^{k,t-1/2} \rangle] + \sum_{t=1}^{T} \mathbb{E}_{i^t} \mathbb{E}_{\mathcal{S}^{k,t}, \boldsymbol{\xi}_{i^t}^{k,t}} [\langle \boldsymbol{\nabla}^{k,t} - \mathbf{g}_{i^t}^{k,t}, \bar{\boldsymbol{\Delta}}^{k,t-1/2} \rangle]
$$

$$
+ \sum_{t=1}^{T} \mathbb{E}_{i^t} \mathbb{E}_{\mathcal{S}^{k,t}, \boldsymbol{\xi}_{i^t}^{k,t}} [\langle \mathbf{g}_{i^t}^{k,t}, \mathbf{u}^k \rangle] + \sum_{t=1}^{T} \mathbb{E}_{i^t} \mathbb{E}_{\mathcal{S}^{k,t}, \boldsymbol{\xi}_{i^t}^{k,t}} [\langle \mathbf{g}_{i^t}^{k,t}, \bar{\boldsymbol{\Delta}}^{k,t-1/2} - \mathbf{u}^k \rangle],
$$

$$
\tag{4}
$$

holds for all $\mathbf{u}^k \in \mathbb{R}^d$, where $\boldsymbol{\nabla}^{k,t} = \int_0^1 \nabla f(\bar{\mathbf{x}}^{k,t-1} + s\boldsymbol{\Delta}_{i^t}^{k,t-1/2}) \, \mathrm{d}s$, $\mathcal{S}^{k,t} = \{s_i^{k,t}\}_{i=1}^n$ and $\boldsymbol{\xi}_{i^t}^{k,t}$ corresponds to the random variable $\boldsymbol{\xi}$ achieved from Algorithm 3 or 4 (line 1 of Algorithm 1). For the first term of equation (4), we use Cauchy–Schwarz inequality and Chebyshev acceleration to make the term sufficiently small, that is

$$
\sum_{t=1}^{T} \mathbb{E}_{i^t} \mathbb{E}_{\mathcal{S}^{k,t}, \boldsymbol{\xi}_{i^t}^{k,t}} [\langle \boldsymbol{\nabla}^{k,t}, \boldsymbol{\Delta}_{i^t}^{k,t-1/2} - \bar{\boldsymbol{\Delta}}^{k,t-1/2} \rangle]
$$

$$
\leq \sum_{t=1}^{T} \mathbb{E}_{i^t} \mathbb{E}_{\mathcal{S}^{k,t}, \boldsymbol{\xi}_{i^t}^{k,t}} \left[ \left\| \boldsymbol{\nabla}^{k,t} \right\| \left\| \boldsymbol{\Delta}_{i^t}^{k,t-1/2} - \bar{\boldsymbol{\Delta}}^{k,t-1/2} \right\| \right] \leq L\epsilon' T.
$$

$$
\tag{5}
$$

For the second term of equation (4), we use the following lemma to provide its upper bound.

**Lemma 4.** *Under Assumptions 1, 2, 3 and 5, we further suppose each $f_i$ is $H$-smooth, then Algorithm 1 with the local stochastic first-order oracle (Algorithm 3) by taking $\mu = 0$ in Algorithm 3 holds*

$$
\sum_{t=1}^{T} \mathbb{E}_{i^t} \mathbb{E}_{\mathcal{S}^{k,t}, \boldsymbol{\xi}_{i^t}^{k,t}} [\langle \boldsymbol{\nabla}^{k,t} - \mathbf{g}_{i^t}^{k,t}, \bar{\boldsymbol{\Delta}}^{k,t-1/2} \rangle] \leq \frac{2D^2 H\epsilon' T}{D - \epsilon'}.
$$

$$
\tag{6}
$$

For the third term of equation (4), we take $\mathbf{u}^k = -D \frac{\sum_{t=1}^T \sum_{i=1}^n \nabla f_i(\mathbf{w}_i^{k,t})}{\left\| \sum_{t=1}^T \sum_{i=1}^n \nabla f_i(\mathbf{w}_i^{k,t}) \right\|}$. Then we can show that

$$
\sum_{t=1}^{T} \mathbb{E}_{i^t} \mathbb{E}_{\mathcal{S}^{k,t}, \boldsymbol{\xi}_{i^t}^{k,t}} \left[ \langle \mathbf{g}_{i^t}^{k,t}, \mathbf{u}^k \rangle \right] \leq -D\mathbb{E}_{\mathcal{S}^{k,t}, \boldsymbol{\xi}_{i^t}^{k,t}} \left[ \left\| \frac{1}{n} \sum_{t=1}^{T} \sum_{i=1}^{n} \nabla f_i(\mathbf{w}_i^{k,t}) \right\| \right] + \frac{D\sigma\sqrt{T}}{\sqrt{n}}.
$$

$$
\tag{7}
$$

For the last term of equation (4), we use the following lemma to provide its upper bound.

**Lemma 5.** *Under the settings of Lemma 4, we have*

$$
\sum_{t=1}^{T} \mathbb{E}_{i^t} \mathbb{E}_{\mathcal{S}^{k,t}, \boldsymbol{\xi}_{i^t}^{k,t}} \left[ \langle \mathbf{g}_{i^t}^{k,t}, \bar{\boldsymbol{\Delta}}^{k,t-1/2} - \mathbf{u}^k \rangle \right] \leq G\epsilon' T + \frac{\eta G^2 T}{2} + \frac{D^2}{2\eta} + \frac{(4D + \epsilon')\epsilon' T}{2\eta},
$$

$$
\tag{8}
$$

*for all $\left\| \mathbf{u}^k \right\| \leq D$.*

We can also bound the difference between the gradients of the global and local functions as follows.

**Lemma 6** (Sahinoglu & Shahrampour (2024, Lemma 2)). *Let the functions $\{f_i\}_{i=1}^n$ be $H$-smooth and $f(\mathbf{x}) = \frac{1}{n} \sum_{i=1}^n f_i(\mathbf{x})$. For given sequence $\{\mathbf{w}_i^t\}_{t,i=1}^{T,n}$, we suppose there exists some $r > 0$ such that $\|\mathbf{w}_i^t - \bar{\mathbf{w}}^t\| \leq r$ for all $i \in [n]$ and $t \in [T]$, then it holds*

$$
\left\| \frac{1}{nT} \sum_{t=1}^{T} \sum_{i=1}^{n} \nabla f(\mathbf{w}_i^t) \right\| \leq \left\| \frac{1}{nT} \sum_{t=1}^{T} \sum_{i=1}^{n} \nabla f_i(\mathbf{w}_i^t) \right\| + 2rH,
$$

*where $\bar{\mathbf{w}}^t = \frac{1}{n} \sum_{i=1}^n \mathbf{w}_i^t$.*

Combing above results of Lemmas 4–6 and equations (4)-(8), we achieve the theoretical guarantee for our method in the smooth case.

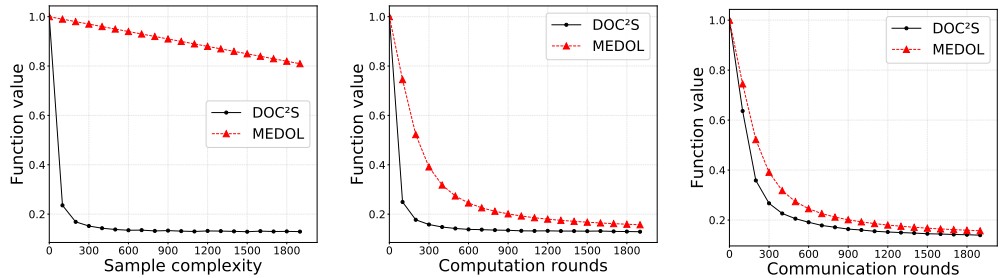

Figure 1: The results of first-order methods for binary classification on dataset "rcv1".

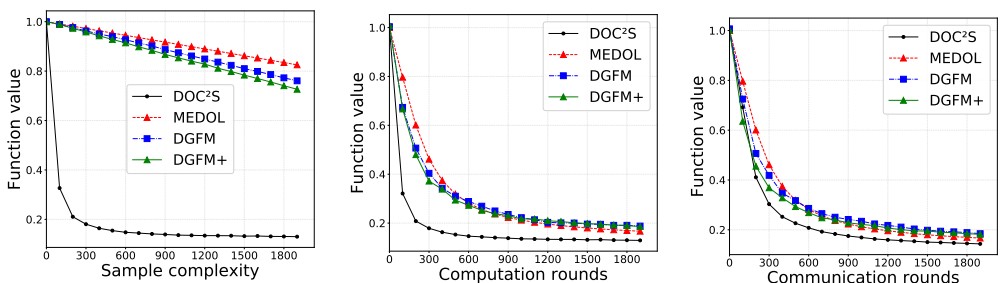

Figure 2: The results of zeroth-order methods for binary classification on dataset "rcv1".

**Lemma 7.** *Under the settings of Lemma 4, running Algorithms 1, 2 and 3 with parameters $T = \mathcal{O}(\epsilon^{-2})$, $K = \mathcal{O}(\delta^{-1}\epsilon^{-1})$, $D = \mathcal{O}(\delta\epsilon^2)$, $R = \tilde{\mathcal{O}}(\gamma^{-1/2})$, and $\eta = \mathcal{O}(\delta\epsilon^3)$ can output $\{\mathbf{w}_i^{\mathrm{out}}\}_{i=1}^n$ such that $\mathbb{E}\left[\|\nabla f(\mathbf{w}_i^{\mathrm{out}})\|_\delta\right] \leq \epsilon$ holds for all $i \in [n]$.*

Connecting the following lemma with Lemma 7, we can establish our main result for the nonsmooth case in Theorem 1. The result in Theorem 3 can be achieved in the similar way.

**Lemma 8** (Sahinoglu & Shahrampour (2024, Proposition 2))**.** *From Proposition 1 and Definition 3, we have $\|\nabla f(\mathbf{x})\|_\delta \leq \|\nabla f_{a\delta}(\mathbf{x})\|_{(1-a)\delta}$ for all $a \in (0,1)$.*

To the best of our knowledge, client sampling techniques previously have been studied in the context of smooth optimization, where convergence analysis relies crucially on the Lipschitz continuity of the gradient (Bai et al., 2024; Liu et al., 2024; Luo et al., 2022). However, in our nonsmooth setting, the gradient is not Lipschitz continuous, rendering existing convergence analyses inapplicable.

Compared with the full participated method ME-DOL for nonsmooth nonconvex optimization (Sahinoglu & Shahrampour, 2024), the proposed DOC²S (Algorithm 1) requires only requires one client to perform its computation per iteration. This results the key step for bounding the consensus error $\|\Delta_i^{k,t+1/2} - \bar{\Delta}^{k,t+1/2}\|$ in our analysis (the proof of Lemma 2 in Appendix A.1) being different from that of ME-DOL in the following aspects.

- The ME-DOL perform online mirror descent on all clients per iteration (Sahinoglu & Shahrampour, 2024, Algorithm 4), which ensures that $\|\Delta_i^{k,t+1/2}\| \leq D$ always holds. This allows the analysis to directly apply Lemma 1 of Shahrampour & Jadbabaie (2017) to bound the consensus error.

- Our DOC²S only performs online mirror descent on one client per iteration. Consequently, the updates of $x_i^{k,t}$ and $\Delta_i^{k,t}$ (when $i = i^t$) in Lines 10 and 16 of Algorithm 1 include an additional factor of $n$ preceding the term $\Delta_i^{k,t-1/2}$ and the min operator, respectively. Moreover, Algorithm 1 incorporates an additional communication step in Line 18. These modifications ensure that $\|\Delta_i^{k,t+1/2} - \bar{\Delta}^{k,t+1/2}\|$ can be effectively bounded, even though only one client participates in the computation and the condition $\|\Delta_i^{k,t+1/2}\| \leq D$ (as required by Sahinoglu & Shahrampour (2024)) is not necessarily satisfied. Specifically, the analysis in Appendix A.1 shows that $\|\Delta_i^{k,t+1/2} - \bar{\Delta}^{k,t+1/2}\| \leq \epsilon'$ and $\|\Delta_i^{k,t+1/2}\| \leq D + \epsilon'$. By choosing an appropriate accuracy $\epsilon'$ and employing Chebyshev acceleration, the consensus error is sufficiently controlled to achieve the desired theoretical guarantees.

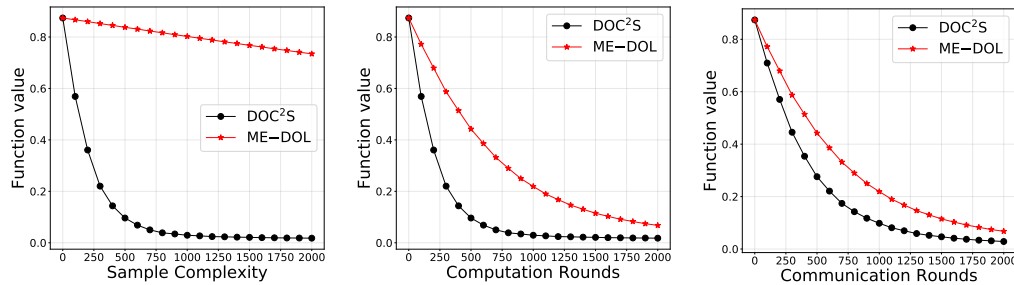

Figure 3: The results of first-order methods for multi-class classification on dataset "MNIST".

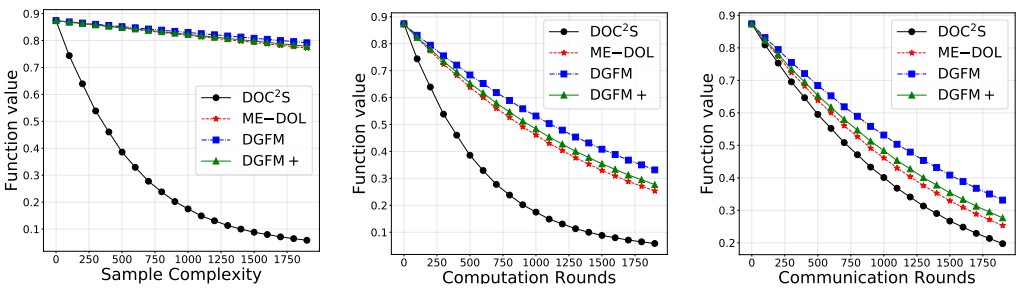

Figure 4: The results of zeroth-order methods for multi-class classification on dataset "MNIST".

## 5 NUMERICAL EXPERIMENTS

This section empirically compare our DOC$^2$S with baseline methods, including ME-DOL (Shahram-pour & Jadbabaie, 2017) for both first-order and zeroth-order settings, as well as DGFM (Lin et al., 2024) for the zeroth-order setting. We conduct experiments on the following two models:

- Nonconvex SVM with capped-$\ell_1$ penalty for binary classification on datasets "rcv1" and "a9a".
- Multilayer perceptron with ReLU activation for multi-class classification on datasets "MNIST" and "fashion-MNIST".

We provide the detailed descriptions for the models in Appendix E.

We perform our numerical experiments on $n = 16$ clients associated with the network of the ring topology. For DOC$^2$S and ME-DOL, we tune the stepsize $\eta$ and diameter $D$ from $\{0.01, 0.05, 0.1\}$ and $\{0.05, 0.01, 0.005, 0.001\}$, respectively. For DGFM and DGFM+, we tune the stepsize $\eta$ from $\{0.001, 0.005, 0.01\}$. Additionally, we set the iteration number of Chebyshev acceleration as $R = 2$ in our DOC$^2$S.

We evaluate the performance of our method and baselines through sample complexity, computation rounds, and communication rounds. We present the experimental results for datasets "rcv1" and "MINST" in Figures 1–4. Due to the space limitation, we defer the results for datasets "a9a" and "Fashion-MNIST" (Figures 5–8) to Appendix E. We can observe that the proposed DOC$^2$S performs better than baselines with respect to all measures. In particular, the client sampling strategy makes the sample complexity of our method significantly superior to that of baselines. All of the empirical results support the shaper upper bounds achieved in our theoretical analysis.

## 6 CONCLUSION

The paper presents a novel stochastic optimization methods for decentralized nonsmooth nonconvex problem. We provide the theoretical analysis to show involving the steps of client sampling and Chebyshev acceleration significantly improve the computation and the communication efficiencies. Additionally, our methods work for both stochastic first-order and zeroth-order oracles. The advantage of proposed method is also validated by empirical studies. In future work, it is possible to extend our ideas to solve decentralized nonsmooth nonconvex problem in time varying networks (Kovalev et al., 2024).

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

## A  UPPER BOUNDS FOR CONSENSUS ERRORS

We present the proofs for Lemmas 2 and 3, which provide upper bounds for consensus errors.

### A.1  PROOF OF LEMMA 2

*Proof.* We let $c = 1 - (1 - 1/\sqrt{2})\sqrt{\gamma}$. According to the line 18 in Algorithm 1 and applying Proposition 2, we have

$$\sum_{i=1}^{n} \left\| \mathbf{\Delta}_i^{k,t+1/2} - \bar{\mathbf{\Delta}}^{k,t+1/2} \right\|^2 \le 14 c^{2R} \sum_{i=1}^{n} \left\| \mathbf{\Delta}_i^{k,t} - \bar{\mathbf{\Delta}}^{k,t} \right\|^2. \tag{9}$$

Based on the update rule of $\mathbf{\Delta}_i^{k,t}$ in Algorithm 1 (line 16) and Proposition 2, we have

$$\bar{\mathbf{\Delta}}^{k,t+1/2} = \bar{\mathbf{\Delta}}^{k,t} = \frac{1}{n}\sum_{i=1}^{n} \mathbf{\Delta}_i^{k,t} = \frac{1}{n}\mathbf{\Delta}_{i^t}^{k,t}.$$

Therefore, it holds

$$
\begin{aligned}
&\sum_{i=1}^{n} \left\| \mathbf{\Delta}_i^{k,t} - \bar{\mathbf{\Delta}}^{k,t} \right\|^2 \\
&= \left\| \mathbf{\Delta}_{i^t}^{k,t} - \frac{1}{n}\mathbf{\Delta}_{i^t}^{k,t} \right\|^2 + \sum_{i \ne i^t} \left\| \mathbf{\Delta}_i^{k,t} - \frac{1}{n}\mathbf{\Delta}_{i^t}^{k,t} \right\|^2 \\
&= \left\| \mathbf{\Delta}_{i^t}^{k,t} - \frac{1}{n}\mathbf{\Delta}_{i^t}^{k,t} \right\|^2 + (n-1)\left\| \mathbf{0} - \frac{1}{n}\mathbf{\Delta}_{i^t}^{k,t} \right\|^2 \\
&= \frac{n-1}{n}\left\| \mathbf{\Delta}_{i^t}^{k,t} \right\|^2 \le n(n-1)D^2,
\end{aligned}
\tag{10}
$$

where the last step is based on the fact $\left\| \mathbf{\Delta}_{i^t}^{k,t} \right\| \le nD$.

Combing above results, we have

$$
\begin{aligned}
&\left\| \mathbf{\Delta}_i^{k,t+1/2} - \bar{\mathbf{\Delta}}^{k,t+1/2} \right\|^2 \\
&\le \sum_{j=1}^{n} \left\| \mathbf{\Delta}_j^{k,t+1/2} - \bar{\mathbf{\Delta}}^{k,t+1/2} \right\|^2 \\
&\le 14 c^{2R} \sum_{j=1}^{n} \left\| \mathbf{\Delta}_j^{k,t} - \bar{\mathbf{\Delta}}^{k,t} \right\|^2 \\
&\le 14 c^{2R} n(n-1)D^2
\end{aligned}
$$

for all $i \in [n]$, where the second inequality is based on equation (9), the third inequality is based on equation (10). Recall that $c = 1 - (1 - 1/\sqrt{2})\sqrt{\gamma}$. Therefore, the setting of $R$ and Proposition 2 implies

$$\left\| \mathbf{\Delta}_i^{k,t+1/2} - \bar{\mathbf{\Delta}}^{k,t+1/2} \right\| \le \epsilon' \tag{11}$$

for all $i \in [n]$. Consequently, we have

$$
\begin{aligned}
&\left\| \mathbf{\Delta}_i^{k,t+1/2} \right\| \\
&= \left\| \bar{\mathbf{\Delta}}^{k,t+1/2} + (\mathbf{\Delta}_i^{k,t+1/2} - \bar{\mathbf{\Delta}}^{k,t+1/2}) \right\| \\
&\le \left\| \bar{\mathbf{\Delta}}^{k,t+1/2} \right\| + \left\| \mathbf{\Delta}_i^{k,t+1/2} - \bar{\mathbf{\Delta}}^{k,t+1/2} \right\| \\
&= \left\| \frac{1}{n}\mathbf{\Delta}_{i^t}^{k,t} \right\| + \left\| \mathbf{\Delta}_i^{k,t+1/2} - \bar{\mathbf{\Delta}}^{k,t-1/2} \right\| \\
&\le D + \epsilon',
\end{aligned}
$$

where the last step is based on equation (11) and the fact $\left\| \mathbf{\Delta}_{i^t}^{k,t} \right\| \le nD$.  □

## A.2    PROOF OF LEMMA 3

*Proof.* Based on the update rule of $\mathbf{x}_i^{k,t}$ in Algorithm 1 (line 10), we denote

$$\mathbf{e}_i^{k,t} = \mathbf{x}_i^{k,t} - \mathbf{y}_i^{k,t-1} = \begin{cases} n\boldsymbol{\Delta}_i^{k,t-1/2}, & i = i^t \\ \mathbf{0}, & i \neq i^t \end{cases}.$$

Then we have $\mathbf{x}_i^{k,t} = \mathbf{y}_i^{k,t-1} + \mathbf{e}_i^{k,t}$ for all $i \in [n]$ and

$$\bar{\mathbf{x}}^{k,t} = \bar{\mathbf{y}}^{k,t-1} + \bar{\mathbf{e}}^{k,t}, \tag{12}$$

where $\bar{\mathbf{e}}^{k,t} = \frac{1}{n}\sum_{i=1}^n \mathbf{e}_i^{k,t}$. Furthermore, we get

$$\sum_{i=1}^n \left\| \mathbf{e}_i^{k,t} - \bar{\mathbf{e}}^{k,t} \right\|^2$$

$$= \sum_{i=1}^n \left\| \mathbf{e}_i^{k,t} - \frac{1}{n}\sum_{i=1}^n \mathbf{e}_i^{k,t} \right\|^2$$

$$= \left\| \mathbf{e}_{i^t}^{k,t} - \boldsymbol{\Delta}_{i^t}^{k,t-1/2} \right\|^2 + \sum_{i \neq i^t} \left\| \mathbf{e}_i^{k,t} - \boldsymbol{\Delta}_{i^t}^{k,t-1/2} \right\|^2 \tag{13}$$

$$= \left\| n\boldsymbol{\Delta}_{i^t}^{k,t-1/2} - \boldsymbol{\Delta}_{i^t}^{k,t-1/2} \right\|^2 + (n-1)\left\| \mathbf{0} - \boldsymbol{\Delta}_{i^t}^{k,t-1/2} \right\|^2$$

$$= n(n-1)\left\| \boldsymbol{\Delta}_{i^t}^{k,t-1/2} \right\|^2 \leq n(n-1)(D+\epsilon')^2,$$

where the last inequality is based on the fact $\|\boldsymbol{\Delta}_{i^t}^{k,t-1/2}\| \leq D + \epsilon'$ from Lemma 2.

Applying Proposition 2 and noticing that $\mathbf{y}_i^{k,0} = \mathbf{0}$, we get

$$\sqrt{\sum_{j=1}^n \left\| \mathbf{y}_j^{k,t} - \bar{\mathbf{y}}^{k,t} \right\|^2}$$

$$\leq \sqrt{14}c^R \sqrt{\sum_{j=1}^n \left\| \mathbf{x}_j^{k,t} - \bar{\mathbf{x}}^{k,t} \right\|^2}$$

$$= \sqrt{14}c^R \sqrt{\sum_{j=1}^n \left\| \mathbf{y}_j^{k,t-1} + \mathbf{e}_j^{k,t} - \bar{\mathbf{x}}^{k,t} \right\|^2} \tag{14}$$

$$= \sqrt{14}c^R \sqrt{\sum_{j=1}^n \left\| \mathbf{y}_j^{k,t-1} + \mathbf{e}_j^{k,t} - \bar{\mathbf{y}}^{k,t-1} - \bar{\mathbf{e}}^{k,t} \right\|^2}$$

$$\leq \sqrt{14}c^R \sqrt{\sum_{j=1}^n \left\| \mathbf{y}_j^{k,t-1} - \bar{\mathbf{y}}^{k,t-1} \right\|^2} + \sqrt{14}c^R \sqrt{\sum_{j=1}^n \left\| \mathbf{e}_j^{k,t} - \bar{\mathbf{e}}^{k,t} \right\|^2},$$

where $\bar{\mathbf{x}}^{k,t} = \frac{1}{n}\sum_{i=1}^n \mathbf{x}_i^{k,t}$ and $\bar{\mathbf{y}}^{k,t} = \frac{1}{n}\sum_{i=1}^n \mathbf{y}_i^{k,t}$. In the derivation of equation (15), the second inequality is based on Proposition 2, the equalities are based on the definition of $\mathbf{e}_i^{k,t}$ and equation (12), and the last step is based on the triangle inequality of Frobenius norm. Hence, we achieve the recursion

$$\sqrt{\sum_{j=1}^n \left\| \mathbf{y}_j^{k,t} - \bar{\mathbf{y}}^{k,t} \right\|^2} \leq \sqrt{14}c^R \sqrt{\sum_{j=1}^n \left\| \mathbf{y}_j^{k,t-1} - \bar{\mathbf{y}}^{k,t-1} \right\|^2} + \sqrt{14}c^R \sqrt{\sum_{j=1}^n \left\| \mathbf{e}_j^{k,t} - \bar{\mathbf{e}}^{k,t} \right\|^2}. \tag{15}$$

We then use induction to prove

$$\sqrt{\sum_{j=1}^n \left\| \mathbf{y}_j^{k,t} - \bar{\mathbf{y}}^{k,t} \right\|^2} \leq \frac{(D+\epsilon')\epsilon'}{D-\epsilon'}$$

for all $t \geq 1$ and $\epsilon' < D$ as follows.

**Induction Base:** For $t = 0$, we have

$$\sqrt{\sum_{j=1}^{n} \left\| \mathbf{y}_j^{k,0} - \bar{\mathbf{y}}^{k,0} \right\|^2} = 0 \leq \frac{(D+\epsilon')\epsilon'}{D-\epsilon'}.$$

**Induction Step:** We suppose

$$\sqrt{\sum_{j=1}^{n} \left\| \mathbf{y}_j^{k,t-1} - \bar{\mathbf{y}}^{k,t-1} \right\|^2} \leq \frac{(D+\epsilon')\epsilon'}{D-\epsilon'} \tag{16}$$

holds. Substituting the induction hypothesis (16) and equation (13) into equation (15) implies

$$\sqrt{\sum_{j=1}^{n} \left\| \mathbf{y}_j^{k,t} - \bar{\mathbf{y}}^{k,t} \right\|^2} \tag{17}$$

$$\leq \sqrt{14} c^R \cdot \frac{(D+\epsilon')\epsilon'}{D-\epsilon'} + \sqrt{14} c^R \sqrt{n(n-1)}(D+\epsilon') \leq \frac{(D+\epsilon')\epsilon'}{D-\epsilon'}, \tag{18}$$

where we take

$$R \geq \left\lceil \frac{1}{(1-1/\sqrt{2})\sqrt{\gamma}} \log \frac{\sqrt{14n(n-1)}D}{\epsilon'} \right\rceil.$$

$\square$

## B  PROOFS FOR THE SMOOTH CASE

This section provides proofs for the results of our method with stochastic first-order oracle in the smooth case. We first provide two basic lemmas.

**Lemma 9** ((Parikh et al., 2014, Section 6.5), (Shahrampour & Jadbabaie, 2017, Lemma 6)). *For given* $\mathbf{g}, \boldsymbol{\Delta} \in \mathbb{R}^d$ *and* $D > 0$*, the problem*

$$\min_{\|\mathbf{x}\| \leq D} \left\{ \langle \mathbf{x}, \mathbf{g} \rangle + \frac{1}{2\eta} \|\mathbf{x} - \boldsymbol{\Delta}\|^2 \right\} \tag{19}$$

*has the unique solution*

$$\mathbf{x}^* = \min \left\{ 1, \frac{D}{\|\boldsymbol{\Delta} - \eta\mathbf{g}\|} \right\} (\boldsymbol{\Delta} - \eta\mathbf{g}).$$

*Additionally, we have*

$$\langle \mathbf{g}, \mathbf{x}^* - \mathbf{u} \rangle \leq \frac{1}{2\eta} \|\boldsymbol{\Delta} - \mathbf{u}\|^2 - \frac{1}{2\eta} \|\mathbf{u} - \mathbf{x}^*\|^2 - \frac{1}{2\eta} \|\boldsymbol{\Delta} - \mathbf{x}^*\|^2,$$

*for all* $\mathbf{u} \in \mathbb{R}^d$.

**Lemma 10.** *Under the setting of Lemma 2, we have*

$$\frac{1}{2\eta} \sum_{t=1}^{T} \mathbb{E}_{i^t} \mathbb{E}_{\mathcal{S}^{k,t}, \boldsymbol{\xi}_{i^t}^{k,t}} \left[ \left\| \boldsymbol{\Delta}_{i^t}^{k,t-1/2} - \mathbf{u}^k \right\|^2 - \left\| \frac{\boldsymbol{\Delta}_{i^t}^{k,t}}{n} - \mathbf{u}^k \right\|^2 \right]$$

$$\leq \frac{D^2}{2\eta} + \frac{(4D+\epsilon')\epsilon'T}{2\eta}$$

for all $\left\| \mathbf{u}^k \right\| \leq D$.

*Proof.* The left-hand side of the above equation can be decomposed as follows:

$$\frac{1}{2\eta} \sum_{t=1}^{T} \mathbb{E}_{i^t} \mathbb{E}_{\mathcal{S}^{k,t}, \boldsymbol{\xi}_{i^t}^{k,t}} \left[ \left\| \boldsymbol{\Delta}_{i^t}^{k,t-1/2} - \mathbf{u}^k \right\|^2 - \left\| \frac{\boldsymbol{\Delta}_{i^t}^{k,t}}{n} - \mathbf{u}^k \right\|^2 \right]$$

$$= \frac{1}{2\eta} \sum_{t=1}^{T} \mathbb{E}_{i^1,\dots,i^T} \mathbb{E}_{\mathcal{S}^{k,t}, \boldsymbol{\xi}_{i^t}^{k,t}} \left[ \left\| \boldsymbol{\Delta}_{i^t}^{k,t-1/2} - \mathbf{u}^k \right\|^2 - \left\| \frac{\boldsymbol{\Delta}_{i^t}^{k,t}}{n} - \mathbf{u}^k \right\|^2 \right]$$

$$= \frac{1}{2\eta} \sum_{t=1}^{T} \mathbb{E}_{i^1,\dots,i^T} \mathbb{E}_{\mathcal{S}^{k,t}, \boldsymbol{\xi}_{i^t}^{k,t}} \left[ \left\| \boldsymbol{\Delta}_{i^t}^{k,t-1/2} - \mathbf{u}^k \right\|^2 - \left\| \boldsymbol{\Delta}_{i^{t+1}}^{k,t+1/2} - \mathbf{u}^k \right\|^2 \right]$$

$$+ \frac{1}{2\eta} \sum_{t=1}^{T} \mathbb{E}_{i^1,\dots,i^T} \mathbb{E}_{\mathcal{S}^{k,t}, \boldsymbol{\xi}_{i^t}^{k,t}} \left[ \left\| \boldsymbol{\Delta}_{i^{t+1}}^{k,t+1/2} - \mathbf{u}^k \right\|^2 - \left\| \frac{\boldsymbol{\Delta}_{i^t}^{k,t}}{n} - \mathbf{u}^k \right\|^2 \right]. \tag{20}$$

For the first term in equation (20), we obtain

$$\frac{1}{2\eta} \sum_{t=1}^{T} \mathbb{E}_{i^1,\dots,i^T} \mathbb{E}_{\mathcal{S}^{k,t}, \boldsymbol{\xi}_{i^t}^{k,t}} \left[ \left\| \boldsymbol{\Delta}_{i^t}^{k,t-1/2} - \mathbf{u}^k \right\|^2 - \left\| \boldsymbol{\Delta}_{i^{t+1}}^{k,t+1/2} - \mathbf{u}^k \right\|^2 \right]$$

$$= \frac{1}{2\eta} \mathbb{E}_{i^1,\dots,i^T} \mathbb{E}_{\mathcal{S}^{k,t}, \boldsymbol{\xi}_{i^1}^{k,t}, \dots, \boldsymbol{\xi}_{i^T}^{k,t}} \left[ \sum_{t=1}^{T} \left( \left\| \boldsymbol{\Delta}_{i^t}^{k,t-1/2} - \mathbf{u}^k \right\|^2 - \left\| \boldsymbol{\Delta}_{i^{t+1}}^{k,t+1/2} - \mathbf{u}^k \right\|^2 \right) \right]$$

$$= \frac{1}{2\eta} \mathbb{E}_{i^1,\dots,i^T} \mathbb{E}_{\mathcal{S}^{k,t}, \boldsymbol{\xi}_{i^1}^{k,t}, \dots, \boldsymbol{\xi}_{i^T}^{k,t}} \left[ \left\| \boldsymbol{\Delta}_{i^1}^{k,1/2} - \mathbf{u}^k \right\|^2 - \left\| \boldsymbol{\Delta}_{i^{T+1}}^{k,T+1/2} - \mathbf{u}^k \right\|^2 \right]$$

$$\leq \frac{1}{2\eta} \mathbb{E}_{i^1,\dots,i^T} \mathbb{E}_{\mathcal{S}^{k,t}, \boldsymbol{\xi}_{i^1}^{k,t}, \dots, \boldsymbol{\xi}_{i^T}^{k,t}} \left[ \left\| \boldsymbol{\Delta}_{i^1}^{k,1/2} - \mathbf{u}^k \right\|^2 \right]$$

$$= \frac{1}{2\eta} \mathbb{E}_{i^1,\dots,i^T} \mathbb{E}_{\mathcal{S}^{k,t}, \boldsymbol{\xi}_{i^1}^{k,t}, \dots, \boldsymbol{\xi}_{i^T}^{k,t}} \left[ \left\| \mathbf{u}^k \right\|^2 \right] \leq \frac{D^2}{2\eta},$$

where the last equality due to $\boldsymbol{\Delta}_{i^1}^{k,1/2} = \mathbf{0}$ and the last inequality is based on the fact $\|\mathbf{u}^k\| \leq D$.

For the second term of equation (20), we notice:

$$\left\| \boldsymbol{\Delta}_{i^{t+1}}^{k,t+1/2} - \mathbf{u}^k \right\|^2 - \left\| \frac{\boldsymbol{\Delta}_{i^t}^{k,t}}{n} - \mathbf{u}^k \right\|^2$$

$$= \left( \left\| \boldsymbol{\Delta}_{i^{t+1}}^{k,t+1/2} - \mathbf{u}^k \right\| + \left\| \frac{\boldsymbol{\Delta}_{i^t}^{k,t}}{n} - \mathbf{u}^k \right\| \right) \left( \left\| \boldsymbol{\Delta}_{i^{t+1}}^{k,t+1/2} - \mathbf{u}^k \right\| - \left\| \frac{\boldsymbol{\Delta}_{i^t}^{k,t}}{n} - \mathbf{u}^k \right\| \right)$$

then we have

$$\frac{1}{2\eta} \sum_{t=1}^{T} \mathbb{E}_{i^1,\dots,i^T} \mathbb{E}_{\mathcal{S}^{k,t}, \boldsymbol{\xi}_{i^t}^{k,t}} \left[ \left\| \boldsymbol{\Delta}_{i^{t+1}}^{k,t+1/2} - \mathbf{u}^k \right\|^2 - \left\| \frac{\boldsymbol{\Delta}_{i^t}^{k,t}}{n} - \mathbf{u}^k \right\|^2 \right]$$

$$\leq \frac{1}{2\eta} \sum_{t=1}^{T} \mathbb{E}_{i^1,\dots,i^T} \mathbb{E}_{\mathcal{S}^{k,t}, \boldsymbol{\xi}_{i^t}^{k,t}} \left[ \left( \left\| \boldsymbol{\Delta}_{i^{t+1}}^{k,t+1/2} \right\| + \|\mathbf{u}^k\| + \left\| \frac{\boldsymbol{\Delta}_{i^t}^{k,t}}{n} \right\| + \|\mathbf{u}^k\| \right) \left\| \boldsymbol{\Delta}_{i^{t+1}}^{k,t+1/2} - \frac{\boldsymbol{\Delta}_{i^t}^{k,t}}{n} \right\| \right]$$

$$\leq \frac{1}{2\eta} \sum_{t=1}^{T} \mathbb{E}_{i^1,\dots,i^T} \mathbb{E}_{\mathcal{S}^{k,t}, \boldsymbol{\xi}_{i^t}^{k,t}} \left[ (4D + \epsilon') \left\| \boldsymbol{\Delta}_{i^{t+1}}^{k,t+1/2} - \frac{\boldsymbol{\Delta}_{i^t}^{k,t}}{n} \right\| \right]$$

$$\leq \frac{(4D + \epsilon')\epsilon' T}{2\eta},$$

where the first inequality follows the triangle inequality $\|\mathbf{a}\| - \|\mathbf{b}\| \leq \|\mathbf{a} - \mathbf{b}\|$ with $\mathbf{a} = \boldsymbol{\Delta}_{i^{t+1}}^{k,t+1/2} - \mathbf{u}^k$ and $\mathbf{b} = \boldsymbol{\Delta}_{i^t}^{k,t}/n - \mathbf{u}^k$, the second inequality is based on the fact $\|\boldsymbol{\Delta}_{i^t}^{k,t}/n\| \leq D$, $\|\mathbf{u}^k\| \leq D$,

and $\|\boldsymbol{\Delta}_{i^{t+1}}^{k,t+1/2}\| \leq D + \epsilon'$ from Lemma 2. The the last inequality in above derivation is achieved as follows

$$\left\|\boldsymbol{\Delta}_{i^{t+1}}^{k,t+1/2} - \frac{\boldsymbol{\Delta}_{i^t}^{k,t}}{n}\right\| = \left\|\boldsymbol{\Delta}_{i^{t+1}}^{k,t+1/2} - \bar{\boldsymbol{\Delta}}^{k,t}\right\| = \left\|\boldsymbol{\Delta}_{i^{t+1}}^{k,t+1/2} - \bar{\boldsymbol{\Delta}}^{k,t+1/2}\right\| \leq \epsilon',$$

where the first step is based on the update rule of $\boldsymbol{\Delta}_i^{k,t}$ (line 16 of Algorithm 1), the second step is based on the update rule of $\boldsymbol{\Delta}_i^{k,t+1/2}$ (line 18 of Algorithm 1) and Proposition 2, and the last step is based on Lemma 2. □

We then provide the proofs of lemmas for the smooth case in Section 4

### B.1 PROOF OF LEMMA 5

*Proof.* Split the equation into the sum of three equations, we have

$$\sum_{t=1}^{T} \mathbb{E}_{i^t} \mathbb{E}_{\mathcal{S}^{k,t}, \boldsymbol{\xi}_{i^t}^{k,t}} [\langle \mathbf{g}_{i^t}^{k,t}, \bar{\boldsymbol{\Delta}}^{k,t-1/2} - \mathbf{u}^k \rangle]$$

$$= \sum_{t=1}^{T} \mathbb{E}_{i^t} \mathbb{E}_{\mathcal{S}^{k,t}, \boldsymbol{\xi}_{i^t}^{k,t}} \left[ \langle \mathbf{g}_{i^t}^{k,t}, \bar{\boldsymbol{\Delta}}^{k,t-1/2} - \boldsymbol{\Delta}_{i^t}^{k,t-1/2} \rangle \right]$$

$$+ \sum_{t=1}^{T} \mathbb{E}_{i^t} \mathbb{E}_{\mathcal{S}^{k,t}, \boldsymbol{\xi}_{i^t}^{k,t}} \left[ \left\langle \mathbf{g}_{i^t}^{k,t}, \boldsymbol{\Delta}_{i^t}^{k,t-1/2} - \frac{\boldsymbol{\Delta}_{i^t}^{k,t}}{n} \right\rangle \right]$$

$$+ \sum_{t=1}^{T} \mathbb{E}_{i^t} \mathbb{E}_{\mathcal{S}^{k,t}, \boldsymbol{\xi}_{i^t}^{k,t}} \left[ \langle \mathbf{g}_{i_t}^{k,t}, \frac{\boldsymbol{\Delta}_{i_t}^{k,t}}{n} - \mathbf{u}^k \rangle \right] \tag{21}$$

We now consider the upper bounds of equation (21). Line 14 of Algorithm 1 with the stochastic first-order oracle (Algorithm 3) and Assumption 3 imply

$$\mathbb{E}_{i^t} \mathbb{E}_{\mathcal{S}^{k,t}, \boldsymbol{\xi}_{i^t}^{k,t}} [\|\mathbf{g}_{i^t}^{k,t}\|] \leq \sqrt{\mathbb{E}_{i^t} \mathbb{E}_{\mathcal{S}^{k,t}, \boldsymbol{\xi}_{i^t}^{k,t}} [\|\mathbf{g}_{i^t}^{k,t}\|^2]} \leq G. \tag{22}$$

For the first term in equation (21), we have

$$\mathbb{E}_{i^t} \mathbb{E}_{\mathcal{S}^{k,t}, \boldsymbol{\xi}_{i^t}^{k,t}} [\langle \mathbf{g}_{i^t}^{k,t}, \bar{\boldsymbol{\Delta}}^{k,t-1/2} - \boldsymbol{\Delta}_{i^t}^{k,t-1/2} \rangle]$$

$$\leq \mathbb{E}_{i^t} \mathbb{E}_{\mathcal{S}^{k,t}, \boldsymbol{\xi}_{i^t}^{k,t}} [\|\mathbf{g}_{i^t}^{k,t}\| \|\bar{\boldsymbol{\Delta}}^{k,t-1/2} - \boldsymbol{\Delta}_{i^t}^{k,t-1/2}\|] \leq G\epsilon', \tag{23}$$

where the first step is based on Cauchy–Schwarz inequality and the second step is based on equation (22) and the result $\|\boldsymbol{\Delta}_{i^t}^{k,t-1/2} - \bar{\boldsymbol{\Delta}}^{k,t-1/2}\| \leq \epsilon'$ from Lemma 2.

For the second term in equation (21), we have

$$\mathbb{E}_{i^t} \mathbb{E}_{\mathcal{S}^{k,t}, \boldsymbol{\xi}_{i^t}^{k,t}} \left[ \left\langle \mathbf{g}_{i^t}^{k,t}, \boldsymbol{\Delta}_{i^t}^{k,t-1/2} - \frac{\boldsymbol{\Delta}_{i^t}^{k,t}}{n} \right\rangle \right]$$

$$\leq \frac{\eta}{2} \mathbb{E}_{i^t} \mathbb{E}_{\mathcal{S}^{k,t}, \boldsymbol{\xi}_{i^t}^{k,t}} [\|\mathbf{g}_{i^t}^{k,t}\|^2] + \frac{1}{2\eta} \mathbb{E}_{i^t} \mathbb{E}_{\mathcal{S}^{k,t}, \boldsymbol{\xi}_{i^t}^{k,t}} \left[ \left\|\boldsymbol{\Delta}_{i^t}^{k,t-1/2} - \frac{\boldsymbol{\Delta}_{i^t}^{k,t}}{n}\right\|^2 \right] \tag{24}$$

$$\leq \frac{\eta G^2}{2} + \frac{1}{2\eta} \mathbb{E}_{i^t} \mathbb{E}_{\mathcal{S}^{k,t}, \boldsymbol{\xi}_{i^t}^{k,t}} \left[ \left\|\boldsymbol{\Delta}_{i^t}^{k,t-1/2} - \frac{\boldsymbol{\Delta}_{i^t}^{k,t}}{n}\right\|^2 \right],$$

where the first step is based on Young's inequality and the second step is based on equation (22).

For the third term in equation (21), we apply Lemma 9 with $\mathbf{g} = \mathbf{g}_{i^t}^{k,t}$, $\mathbf{\Delta} = \mathbf{\Delta}_{i^t}^{k,t-1/2}$, $\mathbf{u} = \mathbf{u}^k$, and $\mathbf{x}^* = \mathbf{\Delta}_{i^t}^{k,t}/n$ and the update rule of $\mathbf{\Delta}_{i^t}^{k,t}$ (line 16 of Algorithm 1) to achieve

$$
\mathbb{E}_{i^t}\mathbb{E}_{\mathcal{S}^{k,t},\boldsymbol{\xi}_{i^t}^{k,t}}\left[\langle \mathbf{g}_{i^t}^{k,t}, \frac{\mathbf{\Delta}_{i^t}^{k,t}}{n} - \mathbf{u}^k \rangle\right]
$$

$$
\leq \mathbb{E}_{i^t}\mathbb{E}_{\mathcal{S}^{k,t},\boldsymbol{\xi}_{i^t}^{k,t}}\left[\frac{1}{2\eta}\|\mathbf{\Delta}_{i^t}^{k,t-1/2} - \mathbf{u}^k\|^2 - \frac{1}{2\eta}\left\|\mathbf{u}^k - \frac{\mathbf{\Delta}_{i^t}^{k,t}}{n}\right\|^2 - \frac{1}{2\eta}\left\|\mathbf{\Delta}_{i^t}^{k,t-1/2} - \frac{\mathbf{\Delta}_{i^t}^{k,t}}{n}\right\|^2\right]. \tag{25}
$$

Substituting equations equations (23), (24), and (25) into equation (21), we achive

$$
\mathbb{E}_{i^t}\mathbb{E}_{\mathcal{S}^{k,t},\boldsymbol{\xi}_{i^t}^{k,t}}\left[\sum_{t=1}^T \langle \mathbf{g}_{i^t}^{k,t}, \bar{\mathbf{\Delta}}^{k,t-1/2} - \mathbf{u}^k \rangle\right]
$$

$$
\leq G\epsilon'T + \frac{\eta G^2 T}{2} + \sum_{t=1}^T \mathbb{E}_{i^t}\mathbb{E}_{\mathcal{S}^{k,t},\boldsymbol{\xi}_{i^t}^{k,t}}\left[\frac{1}{2\eta}\|\mathbf{\Delta}_{i^t}^{k,t-1/2} - \mathbf{u}^k\|^2 - \frac{1}{2\eta}\left\|\mathbf{u}^k - \frac{\mathbf{\Delta}_{i^t}^{k,t}}{n}\right\|^2\right]
$$

$$
\leq G\epsilon'T + \frac{\eta G^2 T}{2} + \frac{D^2}{2\eta} + \frac{(4D + \epsilon')\epsilon'T}{2\eta},
$$

where the last inequality is based on Lemma 10. $\qquad\square$

## B.2 Proof of Lemma 4

*Proof.* Recall that

$$
\boldsymbol{\nabla}^{k,t} = \int_0^1 \nabla f(\bar{\mathbf{x}}^{k,t-1} + s\mathbf{\Delta}_{i^t}^{k,t-1/2})\, \mathrm{d}s.
$$

We split the left-hand side of equation (6) as

$$
\sum_{t=1}^T \mathbb{E}_{i^t}\mathbb{E}_{\mathcal{S}^{k,t},\boldsymbol{\xi}_{i^t}^{k,t}}[\langle \boldsymbol{\nabla}^{k,t} - \mathbf{g}_{i^t}^{k,t}, \bar{\mathbf{\Delta}}^{k,t-1/2} \rangle]
$$

$$
= \sum_{t=1}^T \mathbb{E}_{i^t}\mathbb{E}_{\mathcal{S}^{k,t},\boldsymbol{\xi}_{i^t}^{k,t}}[\langle \bar{\boldsymbol{\nabla}}^{k,t} - \mathbf{g}_{i^t}^{k,t}, \bar{\mathbf{\Delta}}^{k,t-1/2} \rangle] + \sum_{t=1}^T \mathbb{E}_{i^t}\mathbb{E}_{\mathcal{S}^{k,t},\boldsymbol{\xi}_{i^t}^{k,t}}[\langle \boldsymbol{\nabla}^{k,t} - \bar{\boldsymbol{\nabla}}^{k,t}, \bar{\mathbf{\Delta}}^{k,t-1/2} \rangle],
$$

$$\tag{26}$$

where

$$
\bar{\boldsymbol{\nabla}}^{k,t} = \frac{1}{n}\sum_{i=1}^n \boldsymbol{\nabla}_i^{k,t} \qquad \text{and} \qquad \boldsymbol{\nabla}_i^{k,t} = \int_0^1 \nabla f_i(\mathbf{y}_i^{k,t-1} + s\mathbf{\Delta}_i^{k,t-1/2})\, \mathrm{d}s.
$$

We now consider the upper bounds of equation (26). Line 14 of Algorithm 1 with the stochastic first-order oracle (Algorithm 3 with $\mu = 0$) and Assumption 3 imply

$$
\mathbb{E}_{\mathcal{S}^{k,t},\boldsymbol{\xi}_{i^t}^{k,t}}\mathbb{E}_{i^t}[\mathbf{g}_{i^t}^{k,t}] = \frac{1}{n}\sum_{i=1}^n \mathbb{E}_{\mathcal{S}^{k,t},\boldsymbol{\xi}_i^{k,t}}[\mathbf{g}_i^{k,t}] = \frac{1}{n}\sum_{i=1}^n \boldsymbol{\nabla}_i^{k,t} = \bar{\boldsymbol{\nabla}}^{k,t}, \tag{27}
$$

because of $i^t \sim \mathrm{Unif}(\{1,\ldots,n\})$, where $\mathbf{g}_i^{k,t}$ is the output of the First-Order-Estimator/Zeroth-Order-Estimator in line 14 of Algorithm 1 when $i^t = i$. Therefore, the first term of equation (26) equal to 0 so that we only need to consider the second term in equation (26).

We have

$$\left\|\boldsymbol{\nabla}^{k,t} - \bar{\boldsymbol{\nabla}}^{k,t}\right\|$$

$$= \left\|\frac{1}{n} \int_0^1 \left(\nabla f\big(\bar{\mathbf{x}}^{k,t-1} + s\boldsymbol{\Delta}_{i^t}^{k,t-1/2}\big) - \nabla f_i\big(\mathbf{y}_i^{k,t-1} + s\boldsymbol{\Delta}_i^{k,t-1/2}\big)\right) \mathrm{d}s\right\|$$

$$\leq \frac{1}{n} \sum_{i=1}^n \int_0^1 \left\|\nabla f_i\big(\bar{\mathbf{y}}^{k,t-1} + s\boldsymbol{\Delta}_{i^t}^{k,t-1/2}\big) - \nabla f_i\big(\mathbf{y}_i^{k,t-1} + s\boldsymbol{\Delta}_i^{k,t-1/2}\big)\right\| \mathrm{d}s \quad (28)$$

$$\leq \frac{H}{n} \sum_{i=1}^n \int_0^1 \left\|\bar{\mathbf{y}}^{k,t-1} + s\boldsymbol{\Delta}_{i^t}^{k,t-1/2} - \mathbf{y}_i^{k,t-1} - s\boldsymbol{\Delta}_i^{k,t-1/2}\right\| \mathrm{d}s$$

$$\leq \frac{H}{n} \sum_{i=1}^n \left\|\bar{\mathbf{y}}^{k,t-1} - \mathbf{y}_i^{k,t-1}\right\| + \frac{H}{2n} \sum_{i=1}^n \left\|\boldsymbol{\Delta}_{i^t}^{k,t-1/2} - \boldsymbol{\Delta}_i^{k,t-1/2}\right\|,$$

where the second inequality is based on the $H$-smoothness of the function $f_i$.

According to Lemma 3, we have

$$\|\bar{\mathbf{y}}^{k,t-1} - \mathbf{y}_i^{k,t-1}\| \leq \frac{(D+\epsilon')\epsilon'}{D-\epsilon'}. \quad (29)$$

According to Lemma 2, we have

$$\|\bar{\boldsymbol{\Delta}}^{k,t-1/2} - \boldsymbol{\Delta}_i^{k,t-1/2}\| \leq \epsilon'$$

for all $i \in [n]$, which means

$$\left\|\boldsymbol{\Delta}_{i^t}^{k,t-1/2} - \boldsymbol{\Delta}_i^{k,t-1/2}\right\| \leq \left\|\bar{\boldsymbol{\Delta}}^{k,t-1/2} - \boldsymbol{\Delta}_{i^t}^{k,t-1/2}\right\| + \left\|\bar{\boldsymbol{\Delta}}^{k,t-1/2} - \boldsymbol{\Delta}_i^{k,t-1/2}\right\| \leq 2\epsilon'. \quad (30)$$

Substituting equations (29) and (30) into equation (28), we have

$$\|\boldsymbol{\nabla}^{k,t} - \bar{\boldsymbol{\nabla}}^{k,t}\| \leq \frac{2DH\epsilon'}{D-\epsilon'}. \quad (31)$$

Therefore, the second term in equation (26) holds

$$\sum_{t=1}^T \mathbb{E}_{i^t} \mathbb{E}_{\mathcal{S}^{k,t}, \boldsymbol{\xi}_{i^t}^{k,t}} [\langle \boldsymbol{\nabla}^{k,t} - \mathbf{g}_{i^t}^{k,t}, \bar{\boldsymbol{\Delta}}^{k,t-1/2}\rangle]$$

$$= \sum_{t=1}^T \mathbb{E}_{i^t} \mathbb{E}_{\mathcal{S}^{k,t}, \boldsymbol{\xi}_{i^t}^{k,t}} [\langle \boldsymbol{\nabla}^{k,t} - \bar{\boldsymbol{\nabla}}^{k,t}, \bar{\boldsymbol{\Delta}}^{k,t-1/2}\rangle]$$

$$\leq \sum_{t=1}^T \mathbb{E}_{i^t} \mathbb{E}_{\mathcal{S}^{k,t}, \boldsymbol{\xi}_{i^t}^{k,t}} [\|\boldsymbol{\nabla}^{k,t} - \bar{\boldsymbol{\nabla}}^{k,t}\|\|\bar{\boldsymbol{\Delta}}^{k,t-1/2}\|]$$

$$\leq \frac{2D^2 H\epsilon' T}{D-\epsilon'},$$

where the first step is based on equations (26) and (27), the second step is based on Cauchy–Schwarz inequality, and the last step is based on equation (31) and the fact $\|\bar{\boldsymbol{\Delta}}^{k,t-1/2}\| = \|\boldsymbol{\Delta}_{i^{t-1}}^{k,t-1}/n\| \leq D$ from the update rule in line 16 of Algorithm 1. $\qquad\square$

### B.3 PROOF OF LEMMA 7

*Proof.* According to the update rule of $\mathbf{x}_i^{k,t}$ in Algorithm 1 (line 10), we have:

$$\bar{\mathbf{x}}^{k,t} = \frac{1}{n} \sum_{i=1}^n \mathbf{y}_i^{k,t-1} + \boldsymbol{\Delta}_{i^t}^{k,t-1/2} = \bar{\mathbf{x}}^{k,t-1} + \boldsymbol{\Delta}_{i^t}^{k,t-1/2},$$

where the last step is based on the doubly stochastic assumption of matrix $\mathbf{P}$ (Assumption 5).

Recall that

$$\boldsymbol{\nabla}^{k,t} = \int_0^1 \nabla f(\bar{\mathbf{x}}^{k,t-1} + s\boldsymbol{\Delta}_{i^t}^{k,t-1/2})\mathrm{d}s$$

then the continuity of the function $f$ means

$$
\begin{aligned}
&f(\bar{\mathbf{x}}^{k,t}) - f(\bar{\mathbf{x}}^{k,t-1})\\
&= \int_0^1 \langle \nabla f(\bar{\mathbf{x}}^{k,t-1} + s\boldsymbol{\Delta}_{i^t}^{k,t-1/2}), \boldsymbol{\Delta}_{i^t}^{k,t-1/2}\rangle \mathrm{d}s\\
&= \langle \boldsymbol{\nabla}^{k,t}, \boldsymbol{\Delta}_{i^t}^{k,t-1/2}\rangle\\
&= \langle \boldsymbol{\nabla}^{k,t}, \boldsymbol{\Delta}_{i^t}^{k,t-1/2} - \bar{\boldsymbol{\Delta}}^{k,t-1/2}\rangle + \langle \boldsymbol{\nabla}^{k,t} - \mathbf{g}_{i^t}^{k,t}, \bar{\boldsymbol{\Delta}}^{k,t-1/2}\rangle\\
&\quad + \langle \mathbf{g}_{i^t}^{k,t}, \mathbf{u}^k\rangle + \langle \mathbf{g}_{i^t}^{k,t}, \bar{\boldsymbol{\Delta}}^{k,t-1/2} - \mathbf{u}^k\rangle.
\end{aligned}
\tag{32}
$$

Summing equation (32) over $t$ and taking expectation on $\boldsymbol{\xi}_i^{k,t} \sim \mathcal{D}_i$ and $s_i^{k,t} \sim \mathrm{Unif}[0,1]$ yields

$$
\begin{aligned}
&\mathbb{E}[f(\bar{\mathbf{x}}^{k,T}) - f(\bar{\mathbf{x}}^{k,0})]\\
&= \sum_{t=1}^T \mathbb{E}_{i^t}\mathbb{E}_{\mathcal{S}^{k,t},\boldsymbol{\xi}_{i^t}^{k,t}}[\langle \boldsymbol{\nabla}^{k,t}, \boldsymbol{\Delta}_{i^t}^{k,t-1/2} - \bar{\boldsymbol{\Delta}}^{k,t-1/2}\rangle] + \sum_{t=1}^T \mathbb{E}_{i^t}\mathbb{E}_{\mathcal{S}^{k,t},\boldsymbol{\xi}_{i^t}^{k,t}}[\langle \boldsymbol{\nabla}^{k,t} - \mathbf{g}_{i^t}^{k,t}, \bar{\boldsymbol{\Delta}}^{k,t-1/2}\rangle]\\
&\quad + \sum_{t=1}^T \mathbb{E}_{i^t}\mathbb{E}_{\mathcal{S}^{k,t},\boldsymbol{\xi}_{i^t}^{k,t}}[\langle \mathbf{g}_{i^t}^{k,t}, \mathbf{u}^k\rangle] + \sum_{t=1}^T \mathbb{E}_{i^t}\mathbb{E}_{\mathcal{S}^{k,t},\boldsymbol{\xi}_{i^t}^{k,t}}[\langle \mathbf{g}_{i^t}^{k,t}, \bar{\boldsymbol{\Delta}}^{k,t-1/2} - \mathbf{u}^k\rangle]
\end{aligned}
\tag{33}
$$

hold for all $\mathbf{u}^k \in \mathbb{R}^d$, where we define $\mathbf{x}_i^{k,0} = \mathbf{y}_i^{k,0}$ and $\bar{\mathbf{x}}^{k,0} = \frac{1}{n}\sum_{i=1}^n \mathbf{x}_i^{k,0} = \frac{1}{n}\sum_{i=1}^n \mathbf{y}_i^{k,0}$.

For the first term of equation (33), we have:

$$
\begin{aligned}
&\sum_{t=1}^T \mathbb{E}_{i^t}\mathbb{E}_{\mathcal{S}^{k,t},\boldsymbol{\xi}_{i^t}^{k,t}}[\langle \boldsymbol{\nabla}^{k,t}, \boldsymbol{\Delta}_{i^t}^{k,t-1/2} - \bar{\boldsymbol{\Delta}}^{k,t-1/2}\rangle]\\
&\leq \sum_{t=1}^T \mathbb{E}_{i^t}\mathbb{E}_{\mathcal{S}^{k,t},\boldsymbol{\xi}_{i^t}^{k,t}}\left[\left\|\boldsymbol{\nabla}^{k,t}\right\|\left\|\boldsymbol{\Delta}_{i^t}^{k,t-1/2} - \bar{\boldsymbol{\Delta}}^{k,t-1/2}\right\|\right] \leq L\epsilon'T,
\end{aligned}
\tag{34}
$$

where the first step is based on the Cauchy–Schwarz inequality and the second step due to the result $\|\boldsymbol{\Delta}_{i^t}^{k,t-1/2} - \bar{\boldsymbol{\Delta}}^{k,t-1/2}\| \leq \epsilon'$ from Lemma 2 and the fact that $\|\nabla f(\mathbf{x})\| \leq L$. For the upper bound of $\|\nabla f(\mathbf{x})\|$, notice that we have

$$\|\nabla f_i(\mathbf{x})\| = \|\mathbb{E}[\nabla F_i(\mathbf{x};\boldsymbol{\xi}_i)]\| \leq \mathbb{E}[\|\nabla F_i(\mathbf{x};\boldsymbol{\xi}_i)\|] \leq \mathbb{E}[L(\boldsymbol{\xi}_i)] \leq \sqrt{\mathbb{E}[L(\boldsymbol{\xi}_i)^2]} \leq L$$

for all $x \in \mathbb{R}^d$ and $i \in [n]$, where we use Jensen's inequality and Assumption 1. Hence, we have

$$\|\nabla f(\mathbf{x})\| = \left\|\frac{1}{n}\sum_{i=1}^n \nabla f_i(\mathbf{x})\right\| \leq \frac{1}{n}\sum_{i=1}^n \|\nabla f_i(\mathbf{x})\| \leq \frac{1}{n}\sum_{i=1}^n L = L.$$

For the second term of equation (33), we apply Lemma 4 to achieve

$$\sum_{t=1}^T \mathbb{E}_{i^t}\mathbb{E}_{\mathcal{S}^{k,t},\boldsymbol{\xi}_{i^t}^{k,t}}[\langle \boldsymbol{\nabla}^{k,t} - \bar{\boldsymbol{\nabla}}^{k,t}, \bar{\boldsymbol{\Delta}}^{k,t-1/2}\rangle] \tag{35}$$

$$\leq \frac{2D^2 H\epsilon'T}{D - \epsilon'}. \tag{36}$$

For the third term of equation (33), we take

$$\mathbf{u}^k = -D\frac{\sum_{t=1}^T \sum_{i=1}^n \nabla f_i(\mathbf{w}_i^{k,t})}{\left\|\sum_{t=1}^T \sum_{i=1}^n \nabla f_i(\mathbf{w}_i^{k,t})\right\|},$$

which means

$$\sum_{t=1}^{T}\mathbb{E}_{i^t}\mathbb{E}_{\mathcal{S}^{k,t},\boldsymbol{\xi}_{i^t}^{k,t}}\left[\left\langle \mathbf{g}_{i^t}^{k,t},\mathbf{u}^k\right\rangle\right]$$

$$=\mathbb{E}_{\mathcal{S}^{k,t},\boldsymbol{\xi}_{i^t}^{k,t}}\left[\left\langle \frac{1}{n}\sum_{t=1}^{T}\sum_{i=1}^{n}\nabla f_i(\mathbf{w}_i^{k,t}),\mathbf{u}^k\right\rangle\right]$$

$$+\left[\left\langle \frac{1}{n}\sum_{t=1}^{T}\sum_{i=1}^{n}\mathbb{E}_{\mathcal{S}^{k,t},\boldsymbol{\xi}_i^{k,t}}[\mathbf{g}_i^{k,t}]-\frac{1}{n}\sum_{t=1}^{T}\sum_{i=1}^{n}\mathbb{E}_{\mathcal{S}^{k,t},\boldsymbol{\xi}_{i^t}^{k,t}}[\nabla f_i(\mathbf{w}_i^{k,t})],\mathbf{u}^k\right\rangle\right]$$

$$\leq -D\mathbb{E}_{\mathcal{S}^{k,t},\boldsymbol{\xi}_{i^t}^{k,t}}\left[\left\|\frac{1}{n}\sum_{t=1}^{T}\sum_{i=1}^{n}\nabla f_i(\mathbf{w}_i^{k,t})\right\|\right]+\frac{D}{n}\mathbb{E}_{\mathcal{S}^{k,t},\boldsymbol{\xi}_{i^t}^{k,t}}\left[\left\|\sum_{t=1}^{T}\sum_{i=1}^{n}(\nabla f_i(\mathbf{w}_i^{k,t})-\mathbf{g}_i^{k,t})\right\|\right]$$

$$\leq -DT\left[\left\|\frac{1}{nT}\sum_{t=1}^{T}\sum_{i=1}^{n}\nabla f_i(\mathbf{w}_i^{k,t})\right\|\right]+\frac{D\sigma\sqrt{T}}{\sqrt{n}},\tag{37}$$

where the first inequality is based on Cauchy–Schwarz inequality and the fact $\left\|\mathbf{u}^k\right\|\leq D$; the last step is due to

$$\mathbb{E}_{s^{k,t},\boldsymbol{\xi}^{k,t}}\left[\left\|\sum_{t=1}^{T}\sum_{i=1}^{n}\left(\nabla f_i(\mathbf{w}_i^{k,t})-\mathbf{g}_i^{k,t}\right)\right\|\right]$$

$$\leq\sqrt{\mathbb{E}_{s^{k,t},\boldsymbol{\xi}^{k,t}}\left[\left\|\sum_{t=1}^{T}\sum_{i=1}^{n}\left(\nabla f_i(\mathbf{w}_i^{k,t})-\mathbf{g}_i^{k,t}\right)\right\|^2\right]}$$

$$\leq\sqrt{\sum_{t=1}^{T}\sum_{i=1}^{n}\mathbb{E}_{s^{k,t},\boldsymbol{\xi}^{k,t}}\left[\left\|\nabla f_i(\mathbf{w}_i^{k,t})-\mathbf{g}_i^{k,t}\right\|^2\right]}$$

$$\leq\sqrt{nT}\sigma.$$

Here, the first inequality is based on Jensen's inequality, the second inequality is based on the fact

$$\mathbb{E}_{\mathcal{S}^{k,t},\boldsymbol{\xi}_{i^t}^{k,t}}[\mathbf{g}_i^{k,t}]=\mathbb{E}_{\mathcal{S}^{k,t},\boldsymbol{\xi}_{i^t}^{k,t}}[\nabla F_i(\mathbf{w}_i^{k,t})]=\mathbb{E}_{\mathcal{S}^{k,t}}[\nabla f_i(\mathbf{w}_i^{k,t})],$$

and third inequality is based on the fact from Assumption 3.

For the last term of equation (33), we apply Lemma 5 to achieve

$$\mathbb{E}^{i^t}\mathbb{E}_{\mathcal{S}^{k,t},\boldsymbol{\xi}_{i^t}^{k,t}}\left[\sum_{t=1}^{T}\langle \mathbf{g}_{i^t}^{k,t},\bar{\boldsymbol{\Delta}}^{k,t-1/2}-\mathbf{u}^k\rangle\right]\leq G\epsilon'T+\frac{\eta G^2T}{2}+\frac{D^2}{2\eta}+\frac{(4D+\epsilon')\epsilon'T}{2\eta}.\tag{38}$$

Next, we target to bound the distance from $\{\mathbf{w}_i^{t,k}\}$ to $\mathbf{w}_i^{\text{out}}$. For all $i,j\in[n]$, $k\in[K]$, and $t_1,t_2\in[T]$ such that $t_1>t_2$, we have

$$\left\|\mathbf{w}_i^{k,t_1}-\mathbf{w}_j^{k,t_2}\right\|\leq\left\|\mathbf{w}_i^{k,t_1}-\bar{\mathbf{w}}^{k,t_1}\right\|+\left\|\bar{\mathbf{w}}^{k,t_1}-\bar{\mathbf{w}}^{k,t_2}\right\|+\left\|\bar{\mathbf{w}}^{k,t_2}-\mathbf{w}_j^{k,t_2}\right\|.\tag{39}$$

The update rule of $\mathbf{w}_i^{k,t}$ (line 11 of Algorithm 1) implies

$$\left\|\mathbf{w}_i^{k,t}-\bar{\mathbf{w}}^{k,t}\right\|$$

$$=\left\|\mathbf{y}_i^{k,t-1}+s_i^{k,t}\boldsymbol{\Delta}_i^{k,t-1/2}-\bar{\mathbf{y}}^{k,t-1}-\frac{1}{n}\sum_{i=1}^{n}s_i^{k,t}\boldsymbol{\Delta}_i^{k,t-1/2}\right\|$$

$$\leq\left\|\mathbf{y}_i^{k,t-1}-\bar{\mathbf{y}}^{k,t-1}\right\|+\left\|\boldsymbol{\Delta}_i^{k,t-1/2}\right\|+\left\|\frac{1}{n}\sum_{i=1}^{n}s_i^{k,t}\boldsymbol{\Delta}_i^{k,t-1/2}\right\|\tag{40}$$

$$\leq\frac{(D+\epsilon')\epsilon'}{D-\epsilon'}+2(D+\epsilon')$$

$$\leq 3(D+\epsilon'),$$

for all $t \in [T]$ and $\epsilon' \leq D/2$, where the first inequality is based on the fact $s_i^{k,t} \leq 1$ and the second inequality is based on the result $\|\mathbf{\Delta}_i^{k,t-1/2}\| \leq D + \epsilon'$ from Lemmas 2 and 3. Consequently, we have

$$
\begin{aligned}
&\left\| \bar{\mathbf{w}}^{k,t_1} - \bar{\mathbf{w}}^{k,t_2} \right\| \\
&\leq \sum_{t=t_2}^{t_1-1} \left\| \bar{\mathbf{w}}^{k,t+1} - \bar{\mathbf{w}}^{k,t} \right\| \\
&\leq \sum_{t=1}^{T} \left\| \bar{\mathbf{w}}^{k,t+1} - \bar{\mathbf{w}}^{k,t} \right\| \\
&= \sum_{t=1}^{T-1} \left\| \bar{\mathbf{y}}^{k,t} + \frac{1}{n} \sum_{i=1}^{n} s_i^{k,t+1} \mathbf{\Delta}_i^{k,t+1/2} - \bar{\mathbf{y}}^{k,t-1} - \frac{1}{n} \sum_{i=1}^{n} s_i^{k,t} \mathbf{\Delta}_i^{k,t-1/2} \right\| \\
&= \sum_{t=1}^{T-1} \left\| \bar{\mathbf{y}}^{k,t-1} + \mathbf{\Delta}_{i^t}^{k,t-1/2} + \frac{1}{n} \sum_{i=1}^{n} s_i^{k,t+1} \mathbf{\Delta}_i^{k,t+1/2} - \bar{\mathbf{y}}^{k,t-1} - \frac{1}{n} \sum_{i=1}^{n} s_i^{k,t} \mathbf{\Delta}_i^{k,t-1/2} \right\| \\
&= \sum_{t=1}^{T-1} \left\| \mathbf{\Delta}_{i^t}^{k,t-1/2} + \frac{1}{n} \sum_{i=1}^{n} s_i^{k,t+1} \mathbf{\Delta}_i^{k,t+1/2} - \frac{1}{n} \sum_{i=1}^{n} s_i^{k,t} \mathbf{\Delta}_i^{k,t-1/2} \right\| \\
&\leq \sum_{t=1}^{T} \left( \left\| \mathbf{\Delta}_{i^t}^{k,t-1/2} \right\| + \left\| \frac{1}{n} \sum_{i=1}^{n} s_i^{k,t+1} \mathbf{\Delta}_i^{k,t+1/2} \right\| + \left\| \frac{1}{n} \sum_{i=1}^{n} s_i^{k,t} \mathbf{\Delta}_i^{k,t-1/2} \right\| \right) \\
&\leq 3(D + \epsilon')(T - 1) \leq 3(D + \epsilon')T,
\end{aligned}
\tag{41}
$$

where the third step is based on the update rule of $\mathbf{w}_i^{k,t}$ in (line 11 of Algorithm 1); the fourth step is based on the update rule of $\mathbf{x}_i^{k,t}$ (line 10 of Algorithm 1) and the fact $\bar{\mathbf{y}}^{k,t} = \bar{\mathbf{x}}^{k,t}$ from the update rule of $\mathbf{y}_i^{k,t}$ (line 13 of Algorithm 1) and Proposition 2; the last line is based on the result of $\|\mathbf{\Delta}_i^{k,t-1/2}\| \leq D + \epsilon'$ for all $i \in [n]$ from Lemma 2 and the setting $s_i^{k,t} \in [0, 1]$.

Substituting equations (40) and (41) into equation (39), we have

$$
\left\| \mathbf{w}_i^{k,t_1} - \mathbf{w}_j^{k,t_2} \right\| \leq 3(D + \epsilon') + 3(D + \epsilon')T + 3(D + \epsilon') \leq \delta,
\tag{42}
$$

where the last inequality is based on taking

$$
D = \frac{\delta}{4T}, \qquad T > 6, \qquad \text{and} \qquad \epsilon' \leq \frac{T - 6}{3T + 6} D.
\tag{43}
$$

We set $\eta = D/(G\sqrt{T})$ for equation (38) to achieve

$$
\begin{aligned}
&\mathbb{E}_{i^t} \mathbb{E}_{\mathcal{S}^{k,t}, \boldsymbol{\xi}_{i^t}^{k,t}} \left[ \sum_{t=1}^{T} \langle \mathbf{g}_{i^t}^{k,t}, \bar{\mathbf{\Delta}}^{k,t-1/2} - u^k \rangle \right] \\
&\leq G\epsilon' T + \frac{\eta G^2 T}{2} + \frac{D^2}{2\eta} + \frac{(4D + \epsilon')\epsilon' T}{2\eta} \\
&\leq G\epsilon' T + GD\sqrt{T} + \frac{(4D + \epsilon')\epsilon' G T^{3/2}}{2D}.
\end{aligned}
\tag{44}
$$

Substituting equations equations (34), (35), (37), (44) into equation (33):

$$
\begin{aligned}
&\mathbb{E}[f(\bar{\mathbf{x}}^{k,T}) - f(\bar{\mathbf{x}}^{k,0})] \\
&= \sum_{t=1}^{T} \mathbb{E}_{i^t} \mathbb{E}_{\mathcal{S}^{k,t}, \boldsymbol{\xi}_{i^t}^{k,t}} [\langle \boldsymbol{\nabla}^{k,t}, \mathbf{\Delta}_{i^t}^{k,t-1/2} - \bar{\mathbf{\Delta}}^{k,t-1/2} \rangle] + \sum_{t=1}^{T} \mathbb{E}_{i^t} \mathbb{E}_{\mathcal{S}^{k,t}, \boldsymbol{\xi}_{i^t}^{k,t}} [\langle \boldsymbol{\nabla}^{k,t} - \mathbf{g}_{i^t}^{k,t}, \bar{\mathbf{\Delta}}^{k,t-1/2} \rangle] \\
&\quad + \sum_{t=1}^{T} \mathbb{E}_{i^t} \mathbb{E}_{\mathcal{S}^{k,t}, \boldsymbol{\xi}_{i^t}^{k,t}} [\langle \mathbf{g}_{i^t}^{k,t}, \mathbf{u}^k \rangle] + \sum_{t=1}^{T} \mathbb{E}_{i^t} \mathbb{E}_{\mathcal{S}^{k,t}, \boldsymbol{\xi}_{i^t}^{k,t}} [\langle \mathbf{g}_{i^t}^{k,t}, \bar{\mathbf{\Delta}}^{k,t-1/2} - \mathbf{u}^k \rangle]
\end{aligned}
$$

$$\leq L\epsilon'T + \frac{2D^2 H\epsilon'T}{D-\epsilon'} - DT\mathbb{E}_{\mathcal{S}^{k,t},\boldsymbol{\xi}_i^{k,t}} \left[\left\|\frac{1}{nT}\sum_{t=1}^{T}\sum_{i=1}^{n}\nabla f_i(\mathbf{w}_i^{k,t})\right\|\right] + \frac{D\sigma\sqrt{T}}{\sqrt{n}}$$

$$+ G\epsilon'T + GD\sqrt{T} + \frac{(4D+\epsilon')\epsilon'GT^{3/2}}{2D}.$$

Taking the average on above inequality over $k = 1, \ldots, K$ and dividing $DT$, we achieve

$$\mathbb{E}_{\mathcal{S}^{k,t},\boldsymbol{\xi}_i^{k,t}} \left[\frac{1}{K}\sum_{k=1}^{K}\left\|\frac{1}{nT}\sum_{t=1}^{T}\sum_{i=1}^{n}\nabla f_i(\mathbf{w}_i^{k,t})\right\|\right]$$

$$\leq \frac{\mathbb{E}[f(\bar{\mathbf{x}}^{1,0}) - f(\bar{\mathbf{x}}^{K,T})]}{DKT} + \frac{\sigma}{\sqrt{nT}} + \frac{G}{\sqrt{T}} + \left(\frac{2DH}{D-\epsilon'} + \frac{G+L}{D} + \frac{(4D+\epsilon')G\sqrt{T}}{2D^2}\right)\epsilon', \tag{45}$$

Now we start to show the desired approximate stationary point can be achieved at each client. According to Lemma 6 with $r = 3(D+\epsilon')$, we have

$$\left\|\frac{1}{nT}\sum_{t=1}^{T}\sum_{i=1}^{n}\nabla f(\mathbf{w}_i^{k,t})\right\| \leq \left\|\frac{1}{nT}\sum_{t=1}^{T}\sum_{i=1}^{n}\nabla f_i(\mathbf{w}_i^{k,t})\right\| + 6H(D+\epsilon'). \tag{46}$$

Additionally, we have

$$\left\|\hat{\mathbf{w}}_i^k - \mathbf{w}_j^{k,t}\right\| = \left\|\frac{1}{T}\sum_{\tau=1}^{T}\mathbf{w}_i^{k,\tau} - \mathbf{w}_j^{k,t}\right\| \leq \left\|\frac{1}{T}\sum_{\tau=1}^{T}\left(\mathbf{w}_i^{k,\tau} - \mathbf{w}_j^{k,t}\right)\right\| \leq \delta \tag{47}$$

for all $i, j \in [n]$, $k \in [K]$, and $t \in [T]$, where the first step is based on the setting of $\hat{\mathbf{w}}_i^k$ (line 11 of Algorithm 1) and the last step is based on equation (42). Combing above results, we have

$$\mathbb{E}[\left\|\nabla f(\mathbf{w}_i^{\text{out}})\right\|_\delta]$$

$$= \mathbb{E}\left[\frac{1}{K}\sum_{k=1}^{K}\left\|\nabla f(\hat{\mathbf{w}}_i^k)\right\|_\delta\right]$$

$$\leq \mathbb{E}\left[\frac{1}{K}\sum_{k=1}^{K}\left\|\frac{1}{nT}\sum_{t=1}^{T}\sum_{i=1}^{n}\nabla f(\mathbf{w}_i^{k,t})\right\|\right]$$

$$\leq \frac{\mathbb{E}[f(\bar{\mathbf{x}}^{1,0}) - f(\bar{\mathbf{x}}^{K,T})]}{DKT} + \frac{\sigma}{\sqrt{nT}} + \frac{G}{\sqrt{T}}$$

$$+ \left(\frac{2DH}{D-\epsilon'} + \frac{G+L}{D} + \frac{(4D+\epsilon')G\sqrt{T}}{2D^2}\right)\epsilon' + 6H(D+\epsilon'),$$

where the first step is based on the setting of $\mathbf{w}_i^{\text{out}}$ (line 25 of Algorithm 1), the second step is based on the definition of $\|\nabla f(\cdot)\|_\delta$ (Definition 3), and the last step is based on using equations (45) and (46). We substitute the settings of $\epsilon' < D$ and $D = \delta/(4T)$ (see equation (43)) into above result and assume $\delta \leq 1$, then it holds

$$\mathbb{E}\left[\left\|\nabla f(\mathbf{w}_i^{\text{out}})\right\|_\delta\right]$$

$$\leq \frac{4\mathbb{E}[f(\bar{\mathbf{x}}^{1,0}) - f(\bar{\mathbf{x}}^{K,T})]}{\delta K} + \frac{\sigma}{\sqrt{nT}} + \frac{G}{\sqrt{T}} + \frac{3H\delta}{2T}$$

$$+ \left(\frac{2DH}{D-\epsilon'} + \frac{G+L}{D} + \frac{(4D+\epsilon')G\sqrt{T}}{2D^2} + 6H\right)\epsilon'$$

$$\leq \frac{4\nu}{\delta K} + \frac{1}{\sqrt{T}}\left(\frac{\sigma}{\sqrt{n}} + G + \frac{3H}{2}\right) + \left(\frac{2DH}{D-\epsilon'} + \frac{G+L}{D} + \frac{5G\sqrt{T}}{2D} + 6H\right)\epsilon', \tag{48}$$

where we define $\nu = f(\bar{\mathbf{x}}^{1,0}) - \inf_{\mathbf{x}\in\mathbb{R}^d} f(\mathbf{x}) = f(\mathbf{0}) - \inf_{\mathbf{x}\in\mathbb{R}^d} f(\mathbf{x})$.

We denote

$$h_1 = \frac{\sigma}{\sqrt{n}} + G + \frac{3H}{2},$$

and take

$$K = \left\lceil \frac{12\nu}{\delta\epsilon} \right\rceil \qquad \text{and} \qquad T = \left\lceil \frac{9{h_1}^2}{\epsilon^2} \right\rceil,$$

then the first two terms in the last line of equation (48) holds

$$\frac{4\nu}{\delta K} \le \frac{\epsilon}{3} \qquad \text{and} \qquad \frac{1}{\sqrt{T}} \left( \frac{\sigma}{\sqrt{n}} + G + \frac{3H}{2} \right) \le \frac{\epsilon}{3}.$$

For the last term in the last line of equation (48), equation (43) implies

$$D = \frac{\delta}{4T} = \frac{\delta}{4 \left\lceil 9{h_1}^2/\epsilon^2 \right\rceil} \qquad \text{and} \qquad \epsilon' \le \frac{T-6}{3T+6} D = \frac{(\left\lceil 9{h_1}^2/\epsilon^2 \right\rceil - 6)\delta}{12 \left\lceil 9{h_1}^2/\epsilon^2 \right\rceil^2 + 24 \left\lceil 9{h_1}^2/\epsilon^2 \right\rceil}.$$

Combining above results, we can take

$$\epsilon' \le \min \left\{ \frac{(\left\lceil \frac{9h_1^2}{\epsilon^2} \right\rceil - 6)\delta}{12 \left( \left\lceil \frac{9h_1^2}{\epsilon^2} \right\rceil \right)^2 + 24 \left\lceil \frac{9h_1^2}{\epsilon^2} \right\rceil}, \left( 9H + \frac{4(G+L) \left\lceil \frac{9h_1^2}{\epsilon^2} \right\rceil}{\delta} + \frac{10G \left\lceil \frac{9h_1^2}{\epsilon^2} \right\rceil^{3/2}}{\delta} \right)^{-1} \frac{\epsilon}{3} \right\}.$$

and $R = \tilde{\mathcal{O}}(1/\sqrt{\gamma})$ to guarantee

$$\mathbb{E} \left[ \left\| \nabla f(\mathbf{w}_i^{\text{out}}) \right\|_\delta \right] \le \frac{\epsilon}{3} + \frac{\epsilon}{3} + \frac{\epsilon}{3} = \epsilon.$$

$\square$

# C  PROOFS FOR THE NONSMOOTH CASE

This section follows the proof of Lemma 7 to achieve the results in the nonsmooth case.

## C.1  PROOF OF THEOREM 1

*Proof.* Recall that the setting of Theorem 1 takes $\mu = \delta/2$ in the stochastic first-order oracle (Algorithm 3), then Proposition 1 means the function $f_{\delta/2}$ is $\delta L/2$-Lipschitz and $2c_0 L\sqrt{d}/\delta$-smooth. In the view of minimizing the smooth function $f_{\delta/2}$ by Algorithm 1, we can follow the first step in the derivation of equation (48) (in the proof of Lemma 7) by replacing $\delta$, $f$, $\sigma$, and $H$ by $\delta/2$, $f_{\delta/2}$, $G$, and $2c_0 L\sqrt{d}/\delta$, respectively. This implies

$$\begin{aligned}
\mathbb{E}[\|\nabla f_{\delta/2}(\mathbf{w}_i^{\text{out}})\|_{\delta/2}] & \\
\le \frac{8\mathbb{E}[f_{\delta/2}(\bar{\mathbf{x}}^{1,0}) - f_{\delta/2}(\bar{\mathbf{x}}^{K,T})]}{\delta K} &+ \frac{G}{\sqrt{nT}} + \frac{G}{\sqrt{T}} + \frac{3c_0 L\sqrt{d}}{T} \\
&+ \left( \frac{c_0 L D\sqrt{d}}{\delta(D-\epsilon')} + \frac{G+L}{D} + \frac{5G\sqrt{T}}{2D} + \frac{12c_0 L\sqrt{d}}{\delta} \right) \epsilon'.
\end{aligned} \tag{49}$$

We let $\nu = f(\bar{\mathbf{x}}^{1,0}) - \inf_{\mathbf{x} \in \mathbb{R}^d} f(\mathbf{x}) = f(\mathbf{0}) - \inf_{\mathbf{x} \in \mathbb{R}^d} f(\mathbf{x})$, then we have

$$\begin{aligned}
&f_{\delta/2}(\bar{\mathbf{x}}^{1,0}) - f_{\delta/2}(\bar{\mathbf{x}}^{K,T}) \\
\le& f(\bar{\mathbf{x}}^{1,0}) - f(\bar{\mathbf{x}}^{K,T}) + f_{\delta/2}(\bar{\mathbf{x}}^{1,0}) - f(\bar{\mathbf{x}}^{1,0}) - f_{\delta/2}(\bar{\mathbf{x}}^{K,T}) + f(\bar{\mathbf{x}}^{K,T}) \\
\le& \nu + |f_{\delta/2}(\bar{\mathbf{x}}^{1,0}) - f(\bar{\mathbf{x}}^{1,0})| + |f_{\delta/2}(\bar{\mathbf{x}}^{K,T}) - f(\bar{\mathbf{x}}^{K,T})| \\
\le& \nu + \delta L/2 + \delta L/2 \le \nu + L\delta \le \nu + L.
\end{aligned}$$

Let $\nu' = \nu + L$, then combining above results achives

$$\mathbb{E}[\|\nabla f_{\delta/2}(\mathbf{w}_i^{\text{out}})\|_{\delta/2}]$$

$$\leq \frac{8\nu'}{\delta K} + \frac{1}{\sqrt{T}}\left(\frac{G}{\sqrt{n}} + G + 3c_0 L\right) + \left(\frac{c_0 L D \sqrt{d}}{\delta(D - \epsilon')} + \frac{G + L}{D} + \frac{5G\sqrt{T}}{2D} + \frac{12c_0 L \sqrt{d}}{\delta}\right)\epsilon',$$

where we take $T \geq d$. We denote

$$h_2 = \frac{G}{\sqrt{n}} + G + 3c_0 L,$$

and first consider the first two terms in equation (49) By taking

$$K = \left\lceil \frac{24\nu'}{\delta\epsilon} \right\rceil \qquad \text{and} \qquad T = \left\lceil \frac{9h_2^2}{\epsilon^2} \right\rceil + d,$$

then it holds

$$\frac{8\nu'}{\delta K} \leq \frac{\epsilon}{3} \qquad \text{and} \qquad \frac{1}{\sqrt{T}}\left(\frac{G}{\sqrt{n}} + G + 3c_0 L\right) \leq \frac{\epsilon}{3}.$$

We then consider the last term in equation (49). Based on the equation (43) that

$$D = \frac{\delta}{8T} = \frac{\delta}{8\left\lceil 9h_2^2/\epsilon^2 \right\rceil} \qquad \text{and} \qquad \epsilon' \leq \frac{T - 6}{3T + 6}D = \frac{(\left\lceil 9h_2^2/\epsilon^2 \right\rceil - 6)\delta}{24\left\lceil 9h_2^2/\epsilon^2 \right\rceil^2 + 48\left\lceil 9h_2^2/\epsilon^2 \right\rceil},$$

we take

$$\epsilon' \leq \min\left\{ \frac{(\left\lceil \frac{9h_2^2}{\epsilon^2} \right\rceil - 6)\delta}{24\left(\left\lceil \frac{9h_2^2}{\epsilon^2} \right\rceil\right)^2 + 48\left\lceil \frac{9h_2^2}{\epsilon^2} \right\rceil}, \left(\frac{27c_0 L \sqrt{d}}{2\delta} + \frac{4(G + L)\left\lceil \frac{9h_2^2}{\epsilon^2} \right\rceil}{\delta} + \frac{10G\left\lceil \frac{9h_2^2}{\epsilon^2} \right\rceil^{\frac{3}{2}}}{\delta}\right)^{-1}\frac{\epsilon}{3} \right\},$$

Based on the fact $D - \epsilon' \geq (2T + 12)D/(3T + 6) \geq 2D/3$, it holds

$$\left(\frac{c_0 L D \sqrt{d}}{\delta(D - \epsilon')} + \frac{G + L}{D} + \frac{5G\sqrt{T}}{2D} + \frac{12c_0 L \sqrt{d}}{\delta}\right)\epsilon'$$

$$\leq \left(\frac{3c_0 L \sqrt{d}}{2\delta} + \frac{G + L}{D} + \frac{5G\sqrt{T}}{2D} + \frac{12c_0 L \sqrt{d}}{\delta}\right)\epsilon'$$

$$\leq \frac{\epsilon}{3}.$$

Finally, by using Lemma 8, we achieve

$$\mathbb{E}\left[\left\|\nabla f(\mathbf{w}_i^{\text{out}})\right\|_\delta\right] \leq \mathbb{E}[\|\nabla f_{\delta/2}(\mathbf{w}_i^{out})\|_{\delta/2}] \leq \frac{\epsilon}{3} + \frac{\epsilon}{3} + \frac{\epsilon}{3} = \epsilon$$

for all $i \in [n]$. Hence, for $\epsilon < \mathcal{O}(\sqrt{d})$, we can achieve the desired $(\delta, \epsilon)$-Goldstein stationary point on each client with $T = \mathcal{O}(\epsilon^{-2})$. $\qquad\square$

## C.2 Proof of Corollary 2

*Proof.* According to the proof of Theorem 1, we achieve an $(\delta, \epsilon)$-Goldstein stationary point of the objective within the the computation rounds of $KT = \mathcal{O}(\delta^{-1}\epsilon^{-3})$. Since we sample one client for update per round, the overall stochastic first-order oracle complexity is $\mathcal{O}(\delta^{-1}\epsilon^{-3})$. We perform $R$ communication rounds each time and $R = \tilde{\mathcal{O}}(\gamma^{-1/2})$ from Lemma 2. Thus the communication rounds is $\tilde{\mathcal{O}}(\gamma^{-1/2}\delta^{-1}\epsilon^{-3})$. $\qquad\square$

### C.3 PROOF OF THEOREM 3

*Proof.* Recall that the setting of Theorem 1 takes $\mu = \delta/2$ in the stochastic first-order oracle (Algorithm 3), then Proposition 1 means the function $f_{\delta/2}$ is $\delta L/2$-Lipschitz, and $2c_0 L\sqrt{d}/\delta$-smooth. In the view of minimizing the smooth function $f_{\delta/2}$ by Algorithm 1, we can follow the first step in the derivation of equation (48) (in the proof of Lemma 7) by replacing $\delta$, $f$, $G$ and $\sigma$, $H$ by $\delta/2$, $f_{\delta/2}$, $\sqrt{16\sqrt{2\pi}}L$ and $2c_0 L\sqrt{d}/\delta$, respectively. This implies

$$\mathbb{E}[\|\nabla f_{\delta/2}(\mathbf{w}_i^{out})\|_{\delta/2}]$$

$$\leq \frac{8\mathbb{E}[f_{\delta/2}(\bar{\mathbf{x}}^{1,0}) - f_{\delta/2}(\bar{\mathbf{x}}^{K,T})]}{\delta K} + \frac{\sqrt{16\sqrt{2\pi}}dL}{\sqrt{nT}} + \frac{\sqrt{16\sqrt{2\pi}}dL}{\sqrt{T}} + \frac{3c_0 L\sqrt{d}}{T}$$

$$+ \left( \frac{c_0 L D\sqrt{d}}{\delta(D - \epsilon')} + \frac{\sqrt{16\sqrt{2\pi}}dL + L}{D} + \frac{5\sqrt{16\sqrt{2\pi}}dL\sqrt{T}}{2D} + \frac{12c_0 L\sqrt{d}}{\delta} \right) \epsilon'. \tag{50}$$

Let $\nu' = \nu + L$, then combining above results achives

$$\mathbb{E}[\|\nabla f_{\delta/2}(\mathbf{w}_i^{out})\|_{\delta/2}]$$

$$\leq \frac{8\nu'}{\delta K} + \frac{\sqrt{d}}{\sqrt{T}} \left( \frac{\sqrt{16\sqrt{2\pi}}L}{\sqrt{n}} + \sqrt{16\sqrt{2\pi}}L + 3c_0 L \right)$$

$$+ \left( \frac{c_0 L D\sqrt{d}}{\delta(D - \epsilon')} + \frac{\sqrt{16\sqrt{2\pi}}dL + L}{D} + \frac{5\sqrt{16\sqrt{2\pi}}dL\sqrt{T}}{2D} + \frac{12c_0 L\sqrt{d}}{\delta} \right) \epsilon',$$

where the inequality is based on $\sqrt{T} \leq T$.

We denote

$$h_3 = \sqrt{16\sqrt{2\pi}}L \qquad \text{and} \qquad h_4 = \frac{h_3}{\sqrt{n}} + h_3 + 3c_0 L.$$

We first consider the first two terms in equation (49) By taking

$$K = \left\lceil \frac{24\nu'}{\delta\epsilon} \right\rceil \qquad \text{and} \qquad T = \left\lceil \frac{9h_4^2 d}{\epsilon^2} \right\rceil,$$

then it holds

$$\frac{8\nu'}{\delta K} \leq \frac{\epsilon}{3} \qquad \text{and} \qquad \frac{\sqrt{d}}{\sqrt{T}} \left( \frac{h_3}{\sqrt{n}} + h_3 + 3c_0 L \right) \leq \frac{\epsilon}{3}.$$

We then consider the last term in equation (49). Based on the equation (43) that

$$D = \frac{\delta}{8T} = \frac{\delta}{8\lceil 9h_4^2 d/\epsilon^2 \rceil} \qquad \text{and} \qquad \epsilon' \leq \frac{(\lceil 9h_4^2 d/\epsilon^2 \rceil - 6)\delta}{24\lceil 9h_4^2 d/\epsilon^2 \rceil^2 + 48\lceil 9h_4^2 d/\epsilon^2 \rceil}.$$

We take the value of $\epsilon'$ less than or equal to

$$\min\left\{ \frac{(\lceil \frac{9h_4^2 d}{\epsilon^2} \rceil - 6)\delta}{24\lceil \frac{9h_4^2 d}{\epsilon^2} \rceil^2 + 48\lceil \frac{9h_4^2 d}{\epsilon^2} \rceil}, \left( \frac{27c_0 L\sqrt{d}}{2\delta} + \frac{4(h_3 + L)\lceil \frac{9h_4^2 d}{\epsilon^2} \rceil}{\delta} + \frac{10h_3 \lceil \frac{9h_4^2 d}{\epsilon^2} \rceil^{\frac{3}{2}}}{\delta} \right)^{-1} \frac{\epsilon}{3} \right\}.$$

Based on the fact $D - \epsilon' \leq 2/3$, it holds

$$\left( \frac{c_0 L D\sqrt{d}}{\delta(D - \epsilon')} + \frac{h_3 + L}{D} + \frac{5h_3\sqrt{T}}{2D} + \frac{12c_0 L\sqrt{d}}{\delta} \right) \epsilon'$$

$$\leq \left( \frac{3c_0 L\sqrt{d}}{2\delta} + \frac{h_3 + L}{D} + \frac{5h_3\sqrt{T}}{2D} + \frac{12c_0 L\sqrt{d}}{\delta} \right) \epsilon' \leq \frac{\epsilon}{3}.$$

Finally, by using Lemma 8, we achieve

$$\mathbb{E}\left[ \|\nabla f(\mathbf{w}_i^{out})\|_\delta \right] \leq \mathbb{E}[\|\nabla f_{\delta/2}(\mathbf{w}_i^{out})\|_{\delta/2}] \leq \frac{\epsilon}{3} + \frac{\epsilon}{3} + \frac{\epsilon}{3} = \epsilon.$$

Thus, we find a $(\delta, \epsilon)$-stationary with computation rounds $KT = \mathcal{O}(d\delta^{-1}\epsilon^{-3})$. $\qquad \square$

### C.4 Proof of Corollary 4

*Proof.* According to the proof of Theorem 3, we obtain an $(\delta, \epsilon)$-Goldstein stationary point of the objective within $KT = \mathcal{O}(d\delta^{-1}\epsilon^{-3})$ computation rounds. Since we update one client per round, the overall stochastic first-order oracle complexity is $\mathcal{O}(d\delta^{-1}\epsilon^{-3})$. We perform $R$ communication rounds each time, where $R = \tilde{\mathcal{O}}(\gamma^{-1/2})$ from Lemma 2. Therefore, the total number of communication rounds is $\tilde{\mathcal{O}}(d\gamma^{-1/2}\delta^{-1}\epsilon^{-3})$. □

## D Revisiting the Results of ME-DOL

This section shows the the iteration numbers of ME-DOL (Sahinoglu & Shahrampour, 2024) indeed contains the dependency on $n$, which is not explicitly showed in the presentation of .

We follow the notations of Sahinoglu & Shahrampour (2024). According to the proof of their Theorem 2 (page 16) for their first-order method case, it requires

$$c_8(\delta N)^{-1/3} \le \epsilon, \tag{51}$$

where

$$c_8 = \frac{12\gamma\sqrt{n}}{1-\rho} \left( \frac{(1-\rho)(2G + 2c_1\sqrt{n} + cL\sqrt{d}(1-\rho)c_3)}{16\gamma n} \right)^{2/3} = \Omega(n^{1/3}),$$

$$c_1 = 4\sqrt{\frac{G^2(1-\rho) + 4G(L+G)\sqrt{n}}{2(1-\rho)}} = \Omega(n^{1/4}),$$

$$c_3 = \frac{3\sqrt{n}}{1-\rho} + 5 = \Omega(\sqrt{n}).$$

Therefore, we require the computation rounds of $N = \mathcal{O}(n(1-\rho)^{-2}\delta^{-1}\epsilon^{-3})$. Similarly, the other complexity of ME-DOL also contain the dependency on $n$.

## E More Details of Our Numerical Experiments

This section provides the detailed description of the models used in our experiments, as well as the additional experimental results on dataset "a9a" and "Fashion-MNIST".

### E.1 Nonconvex SVM with Capped-$\ell_1$ Penalty

We first consider the model of nonconvex penalized SVM with capped-$\ell_1$ regularizer (Zhang, 2010b), which targets to train the binary classifier $\mathbf{x} \in \mathbb{R}^d$ on dataset $\{(\mathbf{a}_i, b_i)\}_{i=1}^m$, where $\mathbf{a}_i \in \mathbb{R}^d$ and $b_i \in \{-1, 1\}$ are the feature vector and label for the $i$-th sample. We formulate this problem as the following nonsmooth nonconvex problem

$$\min_{\mathbf{x}\in\mathbb{R}^d} f(\mathbf{x}) \triangleq \frac{1}{m} \sum_{i=1}^m g_i(\mathbf{x}),$$

where $g_i(\mathbf{x}) = l(b_i\mathbf{a}_i^\top \mathbf{x}) + \nu(\mathbf{x})$, $l(z) = \max\{1 - z, 0\}$, $\nu(\mathbf{x}) = \lambda \sum_{j=1}^d \min\{|x(j)|, \alpha\}$, and $\lambda, \alpha > 0$. Here, the notation $x(j)$ means the $j$th coordinate of $\mathbf{x}$. We evenly divide functions $\{g_i\}_{i=1}^m$ into $m$ clients. We set $\lambda = 10^{-5}/m$ and $\alpha = 2$ in our experiments.

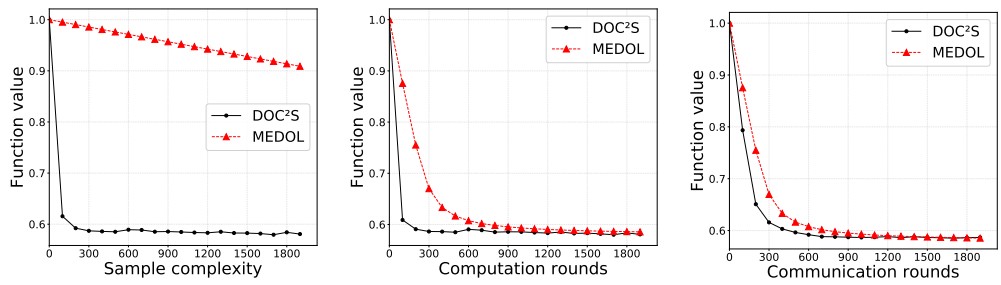

Figure 5: The results of first-order methods for binary classification on dataset "a9a".

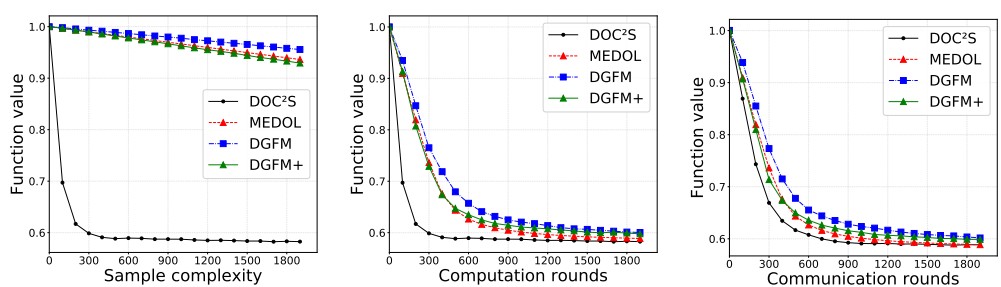

Figure 6: The results of zeroth-order methods for binary classification on dataset "a9a".

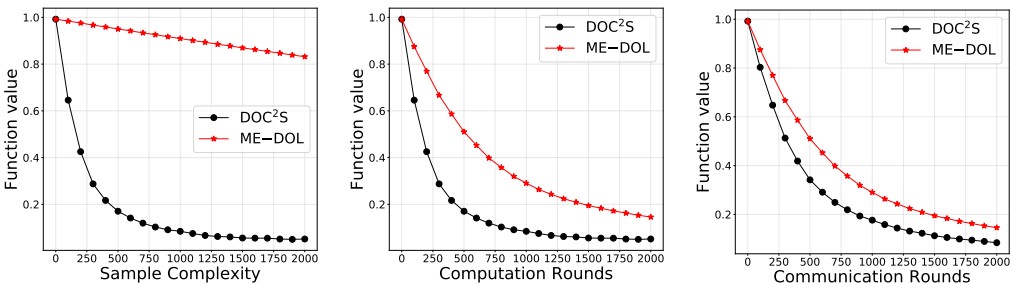

Figure 7: The results of first-order methods for multi-class classification on dataset "fashion-MNIST".

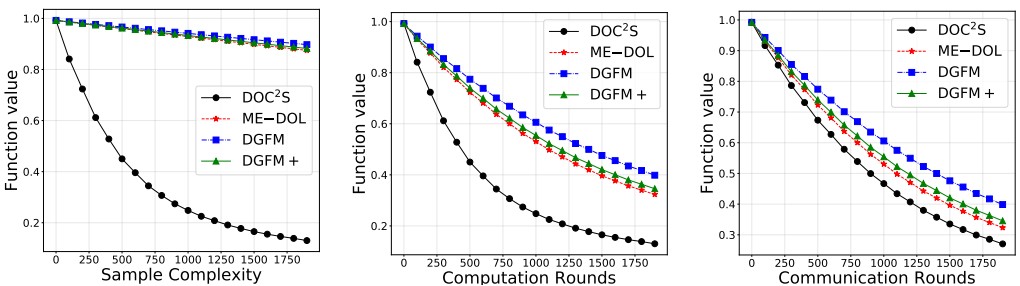

Figure 8: The results of zeroth-order methods for multi-class classification on dataset "fashion-MNIST".

### E.2 MULTILAYER PERCEPTRON WITH RELU ACTIVATION

We have additionally conducted the applications of image classification on datasets "MNIST" and "fashion-MNIST" ($28 \times 28$ pixels for each image, 10 classes). Specifically, we consider the two-layer Multilayer Perceptron (MLP) with ReLU activation and a 256-dimensional hidden layer. Specifically, the local function at the $i$-th client can be written as

$$f_i(\mathbf{x}) \triangleq \frac{1}{m_i} \sum_{j=1}^{m_i} \ell\big(g(\mathbf{x}; \mathbf{a}_i^j), b_i^j\big) + \lambda \|\mathbf{x}\|_2^2,$$

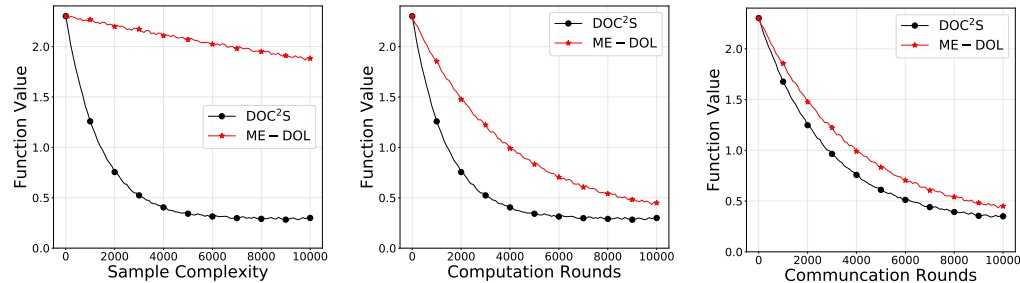

Figure 9: The results of first-order methods for multi-class classification on dataset "CIFAR-10".

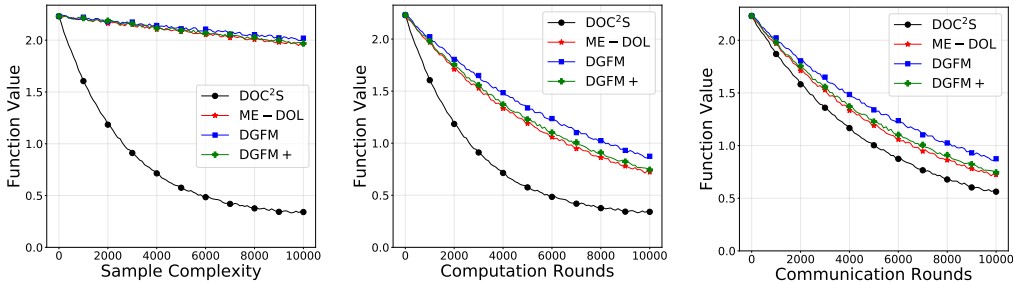

Figure 10: The results of zeroth-order methods for multi-class classification on dataset "CIFAR-10".

where we organize the parameters of the model as $\mathbf{x} = (\mathbf{W}_1, \mathbf{c}_1, \mathbf{W}_2, \mathbf{c}_2)$ with $\mathbf{W}_1 \in \mathbb{R}^{256 \times 784}$, $\mathbf{c}_1 \in \mathbb{R}^{256}$, $\mathbf{W}_2 \in \mathbb{R}^{10 \times 256}$, $\mathbf{c}_2 \in \mathbb{R}^{10}$ and denote $g(\mathbf{x}; \mathbf{a}_i^j) = \mathbf{W}_2 \cdot \text{ReLU}(\mathbf{W}_1 \mathbf{a}_i^j + \mathbf{c}_1) + \mathbf{c}_2$ with $\text{ReLU}(x) = \max(0, x)$. Additionally, we let

$$\ell(\hat{\mathbf{y}}, y) = -\sum_{k=0}^{9} \mathbf{1}_{[y=k]} \log \left( \frac{\exp(\hat{y}_{[k]})}{\sum_{j=0}^{9} \exp(\hat{y}_{[j]})} \right),$$

where $\hat{y}_j$ is the $j$-th coordinate of $\hat{\mathbf{y}}$. We also denote $\mathbf{a}_i^j \in \mathbb{R}^{784}$ and $b_i^j \in \{0, 1, \ldots, 9\}$ as the feature (flattened $28 \times 28$ images) of the $j$th sample on the $i$th client and its corresponding label.

### E.3 ADDITIONAL NUMERICAL RESULTS

We present the experimental results for datasets "a9a" and "Fashion-MNIST" in Figures 5–8. Similar to the observation in Section 5, the proposed DOC$^2$S also performs better than baselines with respect to all measures.

## F MORE EXPERIMENTS FOR REBUTTAL

In this section, we provide more experiments for rebuttal.

### F.1 EXPERIMENTAL RESULTS ON DATASETS "CIFAR-10" AND "CIFAR-100"

We have additionally conducted applications of image classification on the larger-scale datasets "CIFAR-10" (6,0000 images, 10 classes) and "CIFAR-100" (6,0000 images, 100 classes). Each image of these datasets consists of $32 \times 32$ RGB pixels, i.e., the feature dimension is $32 \times 32 \times 3$.

We adopt the standard ResNet-18 architecture citation. This model is composed of an initial convolutional layer, followed by four stages of "Residual Blocks" (totaling 17 convolutional layers) that utilize skip connections to enable effective training of deep networks. The architecture also includes Batch Normalization layers after each convolution. The network is finalized by a Global Average Pooling layer and a single fully connected (linear) layer to produce the classification logits.

We perform our numerical experiments on $n = 16$ clients associated with the network of the ring topology. For DOC$^2$S and ME-DOL, we tune the stepsize $\eta$ and diameter $D$ from $\{0.01, 0.05, 0.1\}$

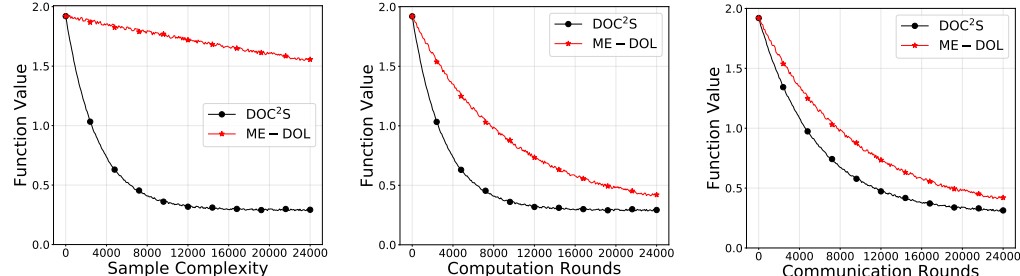

Figure 11: The results of first-order methods for multi-class classification on dataset "CIFAR-100".

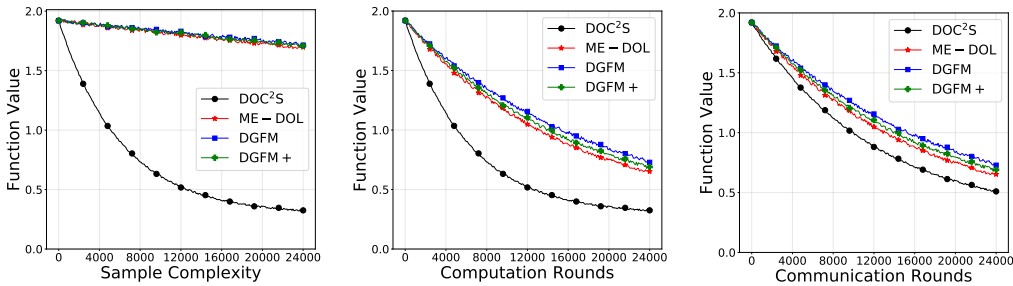

Figure 12: The results of zeroth-order methods for multi-class classification on dataset "CIFAR-100".

and $\{0.05, 0.01, 0.005, 0.001\}$, respectively. For DGFM and DGFM+, we tune the stepsize $\eta$ from $\{0.001, 0.005, 0.01\}$. Additionally, we set the iteration number of Chebyshev acceleration as $R = 2$ in our DOC$^2$S. We have conduct the empirical results for "CIFAR-10" and "CIFAR-100" in Figures 9–10 and Figures 11–12. We can observe that the proposed methods demonstrate superior performance over baselines.

## F.2    SENSITIVITY TO STEP SIZE

We provide experiments in Figures 13–14 to study the performance sensitive to step size. Specifically, we fix the radius parameter $D = 0.01$ and set $\eta = 0.05, 0.01, 0.1$, respectively. The experimental results demonstrate that an excessively large $\eta$ exhibits a faster descent at early state, while it does not exhibits good performance finally.

## F.3    SPECTRAL GAP

We additionally evaluate the effect of network connectivity on the ring-based graphs (Sahinoglu & Shahrampour, 2024) with $n = 16$ clients and set the number of neighbors from $\{3, 5, 7, 9\}$. The corresponding values of $\gamma$ are $0.0507, 0.1476, 0.2818, 0.4414$, respectively. Intuitively, the graph becomes more connected (the value of $\gamma$ increases) as the number of neighbors increases. We provide the experimental results in Figures 15–16 to illustrate the performance of our DOC$^2$S with varying spectral gap $\gamma$ for binary classification on the dataset "a9a" and multi-class classification on the dataset "MNIST", respectively. We can observe that better connectivity (larger $\gamma$) results in a faster convergence, which validate our theoretical results.

## F.4    CONSENSUS ERROR

We additionally present the consensus error $\frac{1}{n} \sum_{i=1}^{n} ||\mathbf{x}_i^{k,t} - \bar{\mathbf{x}}^{k,t}||$ against the number of computation rounds achieved by our DOC$^2$S with different number of rounds ($R \in \{1, 2, 3, 4\}$) in Chebyshev acceleration subroutine and that of the baseline method ME-DOL on the binary classification task ("a9a") and the multi-class classification task ("MNIST"). The results in Figures 17 and 18 demonstrate the larger $R$ achieves the faster consensus error decay. Hence, DOC$^2$S does benefit from better consensus due to adopting FastGossip.

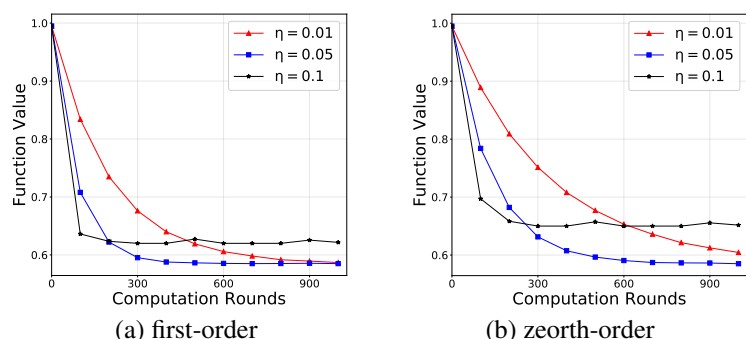

(a) first-order      (b) zeorth-order

Figure 13: The results for binary classification on the dataset "a9a" with step sizes $\eta = 0.01, 0.05, 0.1$.

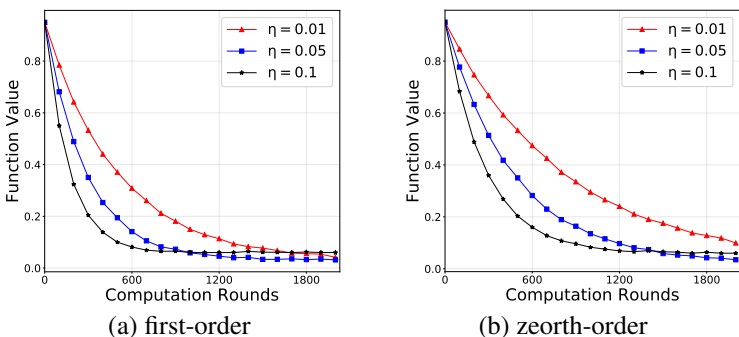

(a) first-order      (b) zeorth-order

Figure 14: The results for multi-class classification on the dataset "MNIST" with step sizes $\eta = 0.01, 0.05, 0.1$.

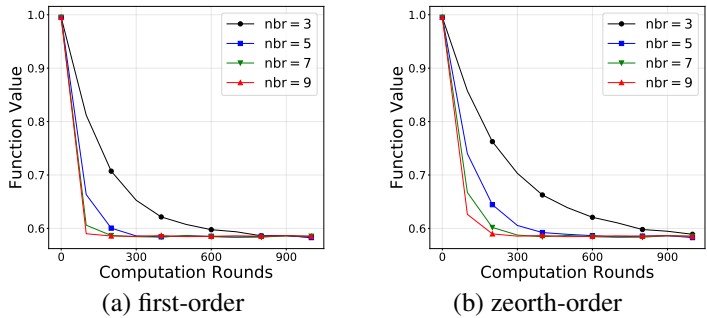

(a) first-order      (b) zeorth-order

Figure 15: The results for binary classification on the dataset "a9a" and the ring-based network of the number of neighbors from $\{3, 5, 7, 9\}$.

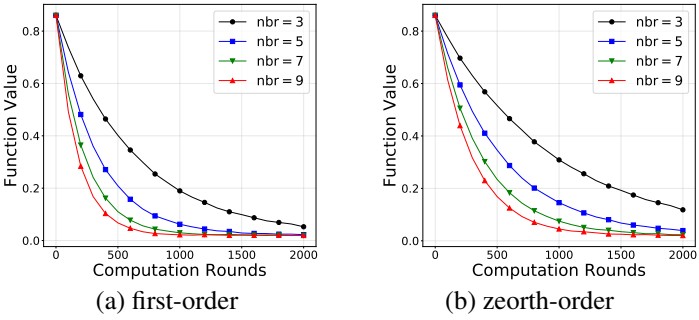

(a) first-order      (b) zeorth-order

Figure 16: The results for multi-class classification on the dataset "MNIST" and the ring-based network of the number of neighbors from $\{3, 5, 7, 9\}$.

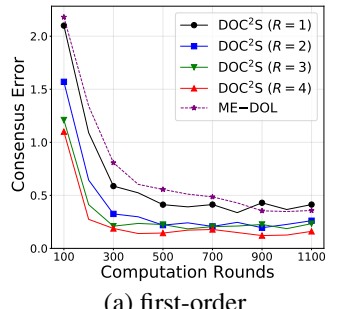 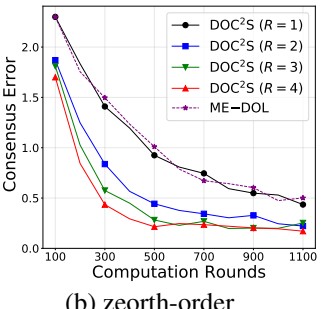

(a) first-order  (b) zeorth-order

Figure 17: The results of consensus error against the number of computation rounds for binary classification on the dataset "a9a" by the algorithm with different communication rounds in the subroutine.

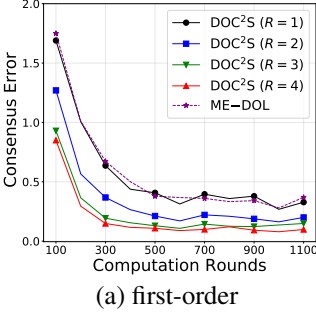 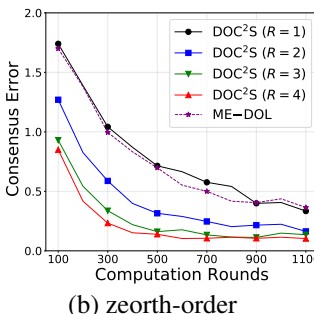

(a) first-order  (b) zeorth-order

Figure 18: The results of consensus error against the number of computation rounds for multi-class classification on the dataset "MNIST" by the algorithm with different communication rounds in the subroutine.

