# OpenReview forum: "Decentralized Nonsmooth Nonconvex Optimization with Client Sampling"
_ICLR.cc/2026/Conference — Submitted to ICLR 2026_

### Official Review · Reviewer_DtU4 · 2025-10-30

**Soundness:** 2
**Presentation:** 3
**Contribution:** 2
**Rating:** 4
**Confidence:** 3

**Summary:**

Prior work (ME-DOL (Sahinoglu & Shahrampour, 2024)) proposed a decentralized algorithm by extending the optimal zero-th order nonsmooth nonconvex stochastic optimization algorithm (Kornowski & Shamir (2024)) to the decentralized setting. This paper is further extending ME-DOL to a client sampling setting where each iteration only requires one agent to compute its local gradient oracle, while adopting Chebyshev accelerated gossip communication.

**Strengths:**

- The numerical experiment shows significant improvement in terms of sample and computation complexity, potentially due to the use of Chebyshev accelerated gossiping.

**Weaknesses:**

- The algorithm design is not clearly motivated. For instance, it is not explained why the algorithm requires performing gossip on two variables $y_i^{k, t}$ and $\Delta_i^{k, t+1/2}$ in every iteration. Similarly, it is not explained why a double loop structure (iterations $k, t$) is necessary.
- While client sampling requires only one agent on the network to access its gradient oracle per iteration, FastGossip in line 13 and line 18 of Algorithm 1 still requires all agents to participate on the communication of every iteration.
- In Lemma 7, I suggest rephrasing "running Algorithm 3" as "running Algorithm 1, 2 and 3".

**Questions:**

- Is the application of FastGossip in line 13 and 18 crucial in achieving convergence of Algorithm 1? Can FastGossip (Chebyshev accelerated gossip) be replaced by a one step decentralized gossip $z_i = \sum_{j=1}^n p_{ij} z_j$?
- Can the authors provide additional plots on the consensus error during training to give a better picture on whether the speedup of DOC${ }^2$S actually benefits from better consensus due to adopting FastGossip (Chebyshev accelerated gossip)?
- What does "Computation Rounds" in Figure 1 refers to? Does it refers to iterations?
- What does "Communication Rounds" in Figure 1 refers to? Is FastGossip counted as 1 communication round or $R$ communication rounds? It would be better if the x-axis is presented as clear metrics such as "Total Network Transmission in Bytes".
- What is the benefit of adopting randomized smoothing (i.e., $\mu > 0$) in the first-order oracle (Algorithm 3)? It seems that the result of Theorem 1 depends on the condition $\mu = 0$ due to Lemma 4, therefore randomized smoothing is not adopted when a first-order oracle is used.

---

> ### Author Response · Authors · 2025-11-21
> **Authors' Response to Reviewer DtU4**
>
> We thank the reviewer DtU4 for the detailed and insightful comments.
>
> **Q1:**
> Is the application of FastGossip in line 13 and 18 crucial in achieving convergence of Algorithm 1? Can FastGossip (Chebyshev accelerated gossip) be replaced by a one step decentralized gossip $\mathbf{z}\_i = \sum_{j=1}^n p_{ij} \mathbf{z}\_j$?
>
> **A1:**
> Recall that the existing method ME-DOL (Sahinoglu \& Shahrampour, 2024) just employs the one-step decentralized gossip $\mathbf{z}\_i = \sum_{j=1}^n p_{ij}\mathbf{z}\_j$, which requires the communication rounds of $\mathcal{O}(n\gamma^{-2}\delta^{-1}\epsilon^{-3})$.
> In contrast, we employs FastGossip to achieve the tighter upper bound $\tilde{\mathcal{O}}(\gamma^{-{1/2}}\delta^{-1}\epsilon^{-3})$ for the communication rounds.
>
> We explain why FastGossip is necessary to our algorithm and analysis as follows.
>
> 1. We first use FastGossip with $R=\mathcal{O}(\gamma^{-1/2}\log(nD\epsilon'^{-1}))=\tilde{\mathcal{O}}(\gamma^{-1/2})$ to achieve the result of Lemma 2, i.e.,
> \begin{align*}
> \|\|\mathbf{\Delta}\_i^{k,t+1/2}-\bar{\mathbf{\Delta}}^{k,t+1/2}\|\| \le \epsilon'
> \quad \text{and} \quad \|\|\mathbf{\Delta}\_i^{k,t+1/2}\|\| \le D+\epsilon'
> \end{align*}
> by  for all $\epsilon'>0$, where the notation $\tilde{\mathcal{O}}(\cdot)$ omits the logarithmic factors in the complexity.
>
> 2. We then use Lemma 2 to prove the result of Lemma 7.
> Specifically, in lines 1053-1058, we apply Lemma 2 to achieve
> $$
> \sum_{t=1}^{T} \mathbb{E}\_{i^t}\mathbb{E}\_{\mathcal{S}^{k,t},\mathbf{\xi}\_{i^t}^{k,t}}[\langle \mathbf{\nabla}^{k,t},\mathbf{\Delta}\_{i^t}^{k,t-1/2}-\bar{\mathbf{\Delta}}^{k,t-1/2}\rangle] \le L\epsilon'T
> $$
> holds for all $\epsilon'>0$.
> Consequently, we need to apply the above inequality with sufficient small $\epsilon'$ (lines 1260-1264) such that
> \begin{align*}
>     \epsilon' \leq \min\lbrace{\frac{(\lceil \frac{9h\_1^2}{\epsilon^2} \rceil - 6)\delta}{12 \left( \lceil \frac{9h\_1^2}{\epsilon^2} \rceil \right)^2 + 24 \lceil \frac{9h\_1^2}{\epsilon^2} \rceil},
>     \left( 9H + \frac{4(G + L)\lceil \frac{9h\_1^2}{\epsilon^2} \rceil}{\delta} + \frac{10G\lceil \frac{9h\_1^2}{\epsilon^2} \rceil^{3/2}}{\delta} \right)^{-1} \frac{\epsilon}{3}}\rbrace.
> \end{align*}
> to guarantee that the last term of equation (48) in lines 1239-1240 is no larger than $\mathcal{O}(\epsilon)$ , i.e.,
> \begin{align*}
>     \left(\frac{2DH}{D-\epsilon'}+\frac{G+L}{D}+\frac{5G\sqrt{T}}{2D}+6H\right)\epsilon' \leq \mathcal{O}(\epsilon).
> \end{align*}
> Based on Proposition 2, we requires taking $R=\tilde{\mathcal{O}}(1/\sqrt{\gamma})$ to guarantee $\epsilon'$ satisfies the above condition.
>
> In contrast, only using one step decentralized gossip $\mathbf{z}\_i = \sum_{j=1}^n p_{ij} \mathbf{z}\_j$ cannot guarantee $\epsilon'$ be sufficient small, so that we cannot achieve the desired approximate Goldstein stationary point.
>
> **Q2:**
> Can the authors provide additional plots on the consensus error during training to give a better picture on whether the speedup of DOC$^2$S actually benefits from better consensus due to adopting FastGossip (Chebyshev accelerated gossip)?
>
> **A2:**
>  We additionally present the consensus error $\frac{1}{n}\sum_{i=1}^n||\mathbf{x}\_i^{k,t}-\bar{\mathbf{x}}^{k,t}||$ against the number of computation rounds achieved by our DOC$^2$S with different number of rounds ($R\in \lbrace{1,2,3,4}\rbrace$) in Chebyshev acceleration subroutine and that of the baseline method ME-DOL on the binary classification task (''a9a'') and the multi-class classification task (''MNIST'').
> The results in **Figures 17–18** demonstrate the larger $R$ do achieve the faster consensus error decay.
> Hence, we verify that our DOC$^2$S does benefit from better consensus due to adopting FastGossip.
>
> **Q3:**
> What does "Computation Rounds" in Figure 1 refers to? Does it refers to iterations?
>
> **A3:**
> The computation rounds corresponds to the computation time to access the local stochastic first-order (zeroth-order) oracle. We use first-order methods as examples to illustrate this point.
>
> 1. For our DOC$^2$S (our Algorithm 1) with local stochastic first-order oracle, client $i_t$ employs First-Order-Estimator (line 14) to access one local stochastic first-order oracle $\mathbf{g}\_{i^t}^{k,t}$ at each iteration.
> This step takes one computation round (one unit time).
>
> 2. For ME-DOL (Algorithm 1 of Sahinoglu \& Shahrampour (2024)) with local stochastic first-order oracle, all clients employ First-Order-Estimator (their Algorithm 2) to access their local stochastic first-order oracles $\lbrace{g_{t,i}^{k}}\rbrace\_{i=1}^n$ in parallel at each iteration.
> This step also takes only one computation round (one unit time), since all $g_{t,i}^{k}$ are achieved simultaneously.

---

> > ### Author Response · Authors · 2025-11-21
> > **Authors' Response to Reviewer DtU4**
> >
> > **Q4:**
> > What does "Communication Rounds" in Figure 1 refers to? Is FastGossip counted as 1 communication round or $R$ communication rounds? It would be better if the x-axis is presented as clear metrics such as "Total Network Transmission in Bytes".
> >
> > **A4:**
> >  One communication round corresponds to the communication time during which all clients exchange information with their neighbors once. Specifically, one communication round can be characterized by the procedure that each client $i$ access the weighted average $\mathbf{x}\_i^+=\sum_{j=1}^n p_{ij}\mathbf{x}\_j$ via the mixing matrix $\mathbf{P}=[p_{ij}]$.
> > Therefore, one FastGossip call is counted as $R$ communication rounds. In this view, the communication rounds is proportion to "Total Network Transmission in Bytes". We tend to keep the terminal of communication rounds in the x-axis, since it is widely used in the presentation of the papers for decentralized optimization, e.g., [1, 2, 3, 4].
> >
> > **Q5:**
> > What is the benefit of adopting randomized smoothing (i.e., $\mu>0$) in the first-order oracle (Algorithm 3)? It seems that the result of Theorem 1 depends on the condition $\mu=0$ due to Lemma 4, therefore randomized smoothing is not adopted when a first-order oracle is used.
> >
> > **A5:**
> > Our analysis is based on the fact that the task of finding an $(\delta,\epsilon)$-Goldstein stationary point of the nonsmooth objective $f$ can be reduced to finding an $(\delta/2,\epsilon)$-Goldstein stationary point of its smooth surrogate $f_{\delta/2}$ (Lemma 8 with $\alpha=1/2$).
> >
> > We outline the idea of our proof based on the above reduction as follows.
> >
> > 1. We first consider finding the approximate Goldstein stationary point for the smooth objective by using the subroutine (Algorithm 3) with $\mu=0$ to achieve Lemmas 4 and 7 (lines 347--349 and 401-403).
> >
> > 2. In lines 1278-1281, we take $\mu=\delta/2$ in Algorithm 3 and use Proposition 1 to show that the smooth surrogate function $f_{\delta/2}$ is $\delta L/2$-Lipschitz and $2c_0 L\sqrt{d}/\delta$-smooth.
> > In the view of minimizing the smooth function $f_{\delta/2}$ by Algorithm 1, we follow the derivation of equation (48) in the proof of Lemma 7 by replacing $\delta$, $f$, $\sigma$, and $H$ by $\delta/2$, $f_{\delta/2}$, $G$, and $2c_0 L\sqrt{d}/\delta$ to show the algorithm guarantees $\mathbb{E}[\|\nabla f\_{\delta/2}(\mathbf{w}\_{i}^{\rm out})\|\_{\delta/2}]\leq \epsilon$ (lines 1281-1335), i.e., the point $\mathbf{w}\_{i}^{\rm out}$ is a $(\delta/2,\epsilon)$-Goldstein stationary point of $f\_{\delta/2}$ in expectation.
> >
> > 3. Finally, we use Lemma 8 to conclude that such $\mathbf{w}\_i^{\rm out}$ is also an $(\delta,\epsilon)$-Goldstein stationary point of our nonsmooth objective $f$ in expectation.
> >
> > **Q6:** It is not explained why a double loop structure (iterations $k, t$) is necessary.
> >
> > **A6:** We introduce the double loop structure because the output $\mathbf{w}\_i^{\rm out}$ is uniformly sampled from $\lbrace{\hat{\mathbf{w}}\_i^1, \dots, \hat{\mathbf{w}}\_i^K}\rbrace$ (line 21 of Algorithm 1), where  $\hat{\mathbf{w}}\_i^{k}=\frac{1}{T}\sum_{t=1}^T\mathbf{w}\_i^{k,t}$ is the average of the recent $T$ points generated from the algorithm.
> > We could use a single loop to present the algorithm, but this would make the presentations of $\hat{\mathbf{w}}\_i^1, \dots, \hat{\mathbf{w}}\_i^K$ (and the related equations in the proof) be complicated.
> >
> > **Q7:** While client sampling requires only one agent on the network to access its gradient oracle per iteration, FastGossip in line 13 and line 18 of Algorithm 1 still requires all agents to participate in the communication of every iteration.
> >
> > **A7:** We agree that the ideal decentralized methods are expected the partial participation for both the computation and the communication.
> > However, even for the smooth case, existing partial participated decentralized methods require all agents to participate in the communication at every iteration [3, 4].
> >
> > **Q8:** In Lemma 7, I suggest rephrasing ''running Algorithm 3'' as ''running Algorithm 1, 2 and 3''.
> >
> > **A8:** Thanks for your suggestion. We have replaced in our rebuttal revision.
> >
> > **Reference**
> >
> > [1] Kevin Scaman, Francis Bach, Sébastien Bubeck, Yin Tat Lee, Laurent Massoulié. Optimal algorithms for smooth and strongly convex distributed optimization in networks. In International Conference on Machine Learning, pages 3027-3036, 2017.
> >
> > [2] Guanghui Lan, Soomin Lee, and Yi Zhou. Communication-efficient algorithms for decentralized and stochastic optimization. Mathematical Programming, 180(1):237–284, 2020.
> >
> > [3] Yuxing Liu, Lesi Chen, Luo Luo. Decentralized convex finite-sum optimization with better dependence on condition numbers. In International Conference on Machine Learning, pages 30807-30841, 2024.
> >
> > [4] Yunyan Bai, Yuxing Liu, Luo Luo. On the complexity of finite-sum smooth optimization under the Polyak–Łojasiewicz condition. In International Conference on Machine Learning, pages:2392-2417, 2024.

---

> > > ### Comment · Reviewer_DtU4 · 2025-11-28
> > >
> > > > Q6: It is not explained why a double loop structure (iterations $k, t$) is necessary.
> > >
> > > > A6: We introduce the double loop structure because the output $\mathbf{w}_i^{\rm out}$ is uniformly sampled from ... be complicated.
> > >
> > > The proposed double-loop structure is inherit in the analysis (see $K={\cal O}(\epsilon^{-1}), T={\cal O}(\epsilon^{-2}))$, therefore it is not a matter of presentation. A follow-up question regarding the double-loop structure: why a reset to the variable $\Delta_{i}^{k, 1/2} = 0$ is necessary in line 5?
> > >
> > > Also, the authors does not respond to my question to why "the algorithm requires performing gossip on two variables $y_i^{k, t}$ and $\Delta_i^{k, t+1/2}$ in every iteration".
> > >
> > > These design choices should be explained in the paper clearly, e.g., in Section 3 before the theoretical analysis.

---

> > > > ### Author Response · Authors · 2025-11-30
> > > > **Reply to Reviewer DtU4**
> > > >
> > > > Thanks for your follow-up response. We address your questions as follows.
> > > >
> > > > **Q9:** The proposed double-loop structure is inherit in the analysis (see $K={\cal O}(\epsilon^{-1}), T={\cal O}(\epsilon^{-2}))$, therefore it is not a matter of presentation.
> > > >
> > > > **A9:**
> > > > 1. The double-loop structure of our Algorithm 1 follows the presentation of ME-DOL [1, Algorithm 1], which extends the Online-to-Non-Convex Conversion (presented by single-loop structure) of Cutkosky et al. [2, Algorithm 1].
> > > >
> > > > 2. Intuitively, the double-loop structure in our DOC$^2$S can be regarded as minimizing the $K$-shifting regret.
> > > > This is the regret with respect to an arbitrary sequence of $K$ vectors $\lbrace{\mathbf u^1}, \dots, {\mathbf u^K}\in\{\mathbf{u}:\|\|\mathbf{u}\|\| \le D}\rbrace $ that changes every $T$ iterations, i.e.,
> > > > $$
> > > > R_T({\mathbf u^1}, \dots, {\mathbf u^K})=\sum_{k=1}^K\sum_{t=1}^T\langle \mathbf{g}_{i^t}^{k,t},\bar{\mathbf{\Delta}}^{k,t-1/2}-\mathbf{u}^k \rangle.
> > > > $$
> > > > We desire the algorithm guarantees every $T$ iterations can achieve a shifting regret of $\mathcal{O}(K\sqrt{T})$, where $K={\cal O}(\epsilon^{-1})$ and $T={\cal O}(\epsilon^{-2})$.
> > > >
> > > > **Q10:** Why a reset to the variable $\mathbf{\Delta}_i^{k, 1/2} = 0$ is necessary in line 5?
> > > >
> > > > **A10:** Setting $\mathbf{\Delta}\_i^{k,1/2}=0$ is for the ease of presentation in our proof.
> > > > In general, any initial point $\mathbf{\Delta}\_i^{k,1/2}$ in the feasible set $\lbrace{\mathbf{\Delta}:\|\|\mathbf{\Delta}\|\| \le D\}\rbrace$ could work for our algorithm and analysis.
> > > >
> > > > **Q11:** The authors does not respond to my question to why "the algorithm requires performing gossip on two variables $\mathbf{y}_i^{k, t}$ and $\mathbf{\Delta}_i^{k, t+1/2}$ in every iteration".
> > > >
> > > > **A11:** In our response **A1**, we have explained why the algorithm requires performing gossip on variables $\lbrace{\mathbf{\Delta}\_i^{k, t+1/2}\rbrace}_{i=1}^n$ (line 18 of Algorithm 1). General speaking, the gossip step guarantees
> > > > the result of Lemma 2 holds, i.e., we have
> > > > $$\|\|\mathbf{\Delta}\_i^{k,t+1/2}-\bar{\mathbf{\Delta}}^{k,t+1/2}\|\| \le \epsilon'
> > > > \quad \text{and} \quad \|\|\mathbf{\Delta}_i^{k,t+1/2}\|\|\le D+\epsilon'$$
> > > > for all $\epsilon'>0$ by taking $R=\mathcal{O}(\gamma^{-1/2}\log(nD\epsilon'^{-1}))=\tilde{\mathcal{O}}(\gamma^{-1/2})$, where the notation $\tilde{\mathcal{O}}(\cdot)$ omits the logarithmic factors in the complexity.
> > > >
> > > > Similarly, we also perform FastGossip on $\lbrace{\mathbf{y}\_i^{k, t}\rbrace}\_{i=1}^n$
> > > > with $R=\tilde{\mathcal{O}}(\gamma^{-1/2})$ (line 13 of Algorithm 1) to achieve the result of Lemma 3, i.e.,
> > > > $$
> > > > \|\|\mathbf{y}\_i^{k,t}-\bar{\mathbf{y}}^{k,t}\|\|\le \frac{(D+\epsilon')\epsilon'}{D-\epsilon'}
> > > > $$
> > > > for all $\epsilon'>0$.
> > > > We then use Lemma 3 to prove the result of Lemma 4 **lines 974–990**, i.e.,
> > > > \begin{align*}
> > > > \sum_{t=1}^{T} \mathbb{E}\_{i^t}\mathbb{E}\_{\mathcal{S}^{k,t},\mathbf{\xi}_{i^t}^{k,t}}[\langle \mathbf{\nabla}^{k,t}-\bar{\mathbf{\nabla}}^{k,t},\bar{\mathbf{\Delta}}^{k,t-1/2}\rangle] \le \frac{2D^2HT\epsilon'}{D-\epsilon'}.
> > > > \end{align*}
> > > > holds for all $\epsilon'>0$. Then we use the result to prove Lemma 7 **lines 1071–1075**.
> > > >
> > > > Consequently, we need to apply the above inequality with sufficient small $\epsilon'$ **lines 1260-1264** such that
> > > > \begin{align*}
> > > >     \epsilon' \leq \min\lbrace{\frac{(\left\lceil \frac{9h_1^2}{\epsilon^2} \right\rceil - 6)\delta}{12 \left( \left\lceil \frac{9h_1^2}{\epsilon^2} \right\rceil \right)^2 + 24 \left\lceil \frac{9h_1^2}{\epsilon^2} \right\rceil}
> > > > , \left( 9H + \frac{4(G + L)\left\lceil \frac{9h_1^2}{\epsilon^2} \right\rceil}{\delta} + \frac{{10G\left\lceil \frac{9h_1^2}{\epsilon^2} \right\rceil^{3/2}}}{\delta} \right)^{-1}
> > > >     \frac{\epsilon}{3}\rbrace}.
> > > > \end{align*}
> > > > to guarantee that the last term of equation (48) in **lines 1239–1240** is no larger than $\mathcal{O}(\epsilon)$ , i.e.,
> > > > \begin{align*}
> > > >     \left(\frac{2DH}{D-\epsilon'}+\frac{G+L}{D}+\frac{5G\sqrt{T}}{2D}+6H\right)\epsilon' \le \mathcal{O}(\epsilon).
> > > > \end{align*}
> > > >
> > > > In contrast, only using one step decentralized gossip $\mathbf{z}\_i = \sum_{j=1}^n p\_{ij} \mathbf{z}\_j$ cannot guarantee $\epsilon'$ be sufficient small, so that we cannot achieve the desired approximate Goldstein stationary point.
> > > >
> > > > **Q12:** These design choices should be explained in the paper clearly, e.g., in Section 3 before the theoretical analysis.
> > > >
> > > > **A12:** Thanks for your suggestions. We have highlighted the intuition of our design for the double-loop structure and the gossip steps in blue.
> > > >
> > > > **References**
> > > >
> > > > [1] Emre Sahinoglu and Shahin Shahrampour. An online optimization perspective on first-order and zero-order decentralized nonsmooth nonconvex stochastic optimization. In International Conference on Machine Learning, pages:43043–43059, 2024.
> > > >
> > > > [2] Ashok Cutkosky, Harsh Mehta, and Francesco Orabona. Optimal stochastic non-smooth non-convex optimization through online-to-non-convex conversion. In International Conference on Machine Learning, pages:6643–6670, 2023.

---

### Official Review · Reviewer_27Yq · 2025-11-01

**Soundness:** 3
**Presentation:** 3
**Contribution:** 3
**Rating:** 6
**Confidence:** 3

**Summary:**

This paper studies decentralized stochastic optimization where each client’s function is nonsmooth, nonconvex, and Lipschitz continuous.
The authors propose **DOC$^2$S** (Decentralized Online-to-Nonconvex Conversion with Client Sampling), which integrates partial client participation and multi-consensus steps. **DOC$^2$S** achieves $(\delta,\epsilon)$–Goldstein stationary points with optimal sample complexity $\mathcal{O}(\delta^{-1}\epsilon^{-3})$, computation complexity $\mathcal{O}(\delta^{-1}\epsilon^{-3})$, and communication complexity $\tilde{\mathcal{O}}(\gamma^{-1/2}\delta^{-1}\epsilon^{-3})$, improving prior decentralized algorithms such as ME-DOL.
The paper also extends **DOC$^2$S** to the zeroth-order (gradient-free) case and presents both theoretical and empirical utilities.

**Strengths:**

1. Introduces the first decentralized nonsmooth nonconvex method supporting client sampling, enhancing the scalability of the proposed algorithm.

2. Achieves optimal sample complexity and sharper communication bounds than other state-of-the-art methods.

3.  Extends the proposed algorithm to the settings of zeroth-order optimization with dimension-dependent complexity $\mathcal{O}(d\delta^{-1}\epsilon^{-3})$.

4.  Theoretical results are well-grounded and consistent with known lower bounds.

5.  Comprehensive comparison with prior literature clarifies author's contributions.

**Weaknesses:**

Assumptions (Lipschitz continuity, fixed mixing matrix) may be restrictive in heterogeneous or time-varying networks.

**Questions:**

1. How sensitive is DOC2S to the network spectral gap $\gamma$ in practical decentralized systems?

2. Does partial participation introduce statistical bias when data are non-i.i.d.?

3. Could DOC2S be extended to dynamic or temporal communication graphs?

---

> ### Author Response · Authors · 2025-11-21
> **Authors' Response to Reviewer 27Yq**
>
> We thank the reviewer 27Yq for the detailed and insightful comments.
>
>
> **Q1:**
> How sensitive is DOC$^2$S to the network spectral gap $\gamma$ in practical decentralized systems?
>
> **A1:**
> We additionally evaluate the effect of network connectivity on the ring-based graphs [2] with $n = 16$ clients and set the number of neighbors from $\lbrace{3, 5, 7, 9}\rbrace$.
> The corresponding values of $\gamma$ are $0.0507, 0.1476, 0.2818, 0.4414$, respectively.
> Intuitively, the graph becomes more connected (the value of $\gamma$ increases) as the number of neighbors increases.
> We provide the experimental results in appendix **(page 31, Figures 15 and 16)** to illustrate the performance of our DOC$^2$S with varying spectral gap $\gamma$ for binary classification on the dataset ''a9a'' and multi-class classification on the dataset ''MNIST'', respectively.
> We can observe that better connectivity (larger $\gamma$) results in a faster convergence, which validates our theoretical results.
>
> **Q2:**
> Assumptions may be restrictive in heterogeneous.
> Does partial participation introduce statistical bias when data are non-i.i.d.?
>
> **A2:**
> We emphasize that our work does address **the heterogeneous setting**, since we assume the local function on the $i$th client has the form of $f_i(\mathbf{x}) \triangleq \mathbb{E}_{\mathbf{\xi}\_i \sim \mathcal{D}\_i}[F_i(\mathbf{x}; \mathbf{\xi}\_i)]$, where $\mathbf{\xi}\_i$ follows distribution $\mathcal{D}_i$ (see Equation (2)).
> It is worth noting that we do **not** require distributions $\mathcal{D}\_1,\dots,\mathcal{D}\_n$ be identical in our algorithm design and theoretical analysis.
>
> Additionally, the partial participation does not introduce statistical bias when data are non-i.i.d., because the client index $i_t$ is uniformly sampled from $\lbrace{1,\dots,n}\rbrace$.
> Please refer to equation (27) as an example, i.e.,
> $$
> \mathbb{E}\_{\mathcal{S}^{k,t},\mathbf{\xi}\_{i^t}^{k,t}}\mathbb{E}\_{i^t}[\mathbf{g}\_{i^t}^{k,t}]
> =\frac{1}{n}\sum_{i=1}^n \mathbb{E}\_{\mathcal{S}^{k,t},\mathbf{\xi}\_{i}^{k,t}}[\mathbf{g}\_{i}^{k,t}]
> =\frac{1}{n}\sum_{i=1}^n \mathbf{\nabla}\_{i}^{k,t}=\bar{\mathbf{\nabla}}^{k,t}.
> $$
>
> **Q3:**
> Assumptions may be restrictive in time-varying networks. Could DOC$^2$S be extended to dynamic or temporal communication graphs?
>
> **A3:**
> In fact, we can naturally extend DOC$^2$S to dynamic or temporal communication graphs.
> Specifically, we can replace our FastGossip subroutine (Algorithm 2) with the multiple consensus subroutine of Li and Lin [2, Section 2.4.2] to address the dynamic graphs.
> For the theoretical guarantee, we can follow the setting of Li and Lin [2] by supposing the mixing matrix at any time satisfies Assumption 5,
> then setting the number of communication rounds in each call of multiple consensus subroutine be $\tilde{\mathcal{O}}(1/\gamma)$ can achieve the overall communication complexity of $\tilde{\mathcal{O}}(1/(\gamma\delta\epsilon^{3}))$ and $\tilde{\mathcal{O}}(d/(\gamma\delta\epsilon^{3}))$ for the first-order and zeroth-order methods, respectively.
> Note that the Chebyshev acceleration does not work for the dynamic graphs, so that we cannot achieve the $1/\sqrt{\gamma}$ dependency like the static graph.
>
> **Reference**
>
> [1] Emre Sahinoglu and Shahin Shahrampour. An online optimization perspective on first-order and zero-order decentralized nonsmooth nonconvex stochastic optimization. In International Conference on
> Machine Learning, pp. 43043–43059, 2024.
>
> [2] Huan Li and Zhouchen Lin. Accelerated gradient tracking over time-varying graphs for decentralized optimization. Journal of Machine Learning Research 25(274):1-52, 2024.

---

### Official Review · Reviewer_HZ98 · 2025-11-02

**Soundness:** 3
**Presentation:** 2
**Contribution:** 2
**Rating:** 4
**Confidence:** 3

**Summary:**

The paper introduces Decentralized Online-to-nonconvex Conversion with Client Sampling (DOC²S), a decentralized algorithm for nonsmooth, nonconvex optimization that uses client sampling.
It integrates the partial participated computation and the multi-consensus steps into decentralized optimization.

Theoretical analysis shows that DOC²S with local stochastic first-order oracle achieves the $(\delta,\epsilon)$-Goldstein staionary points with sample complexity of $\mathcal{O}(\delta^{-1}\epsilon^{-3})$, the computation bounds of $\mathcal{O}(\delta^{-1}\epsilon^{-3})$ and the communication bounds of $\mathcal{O}(\gamma^{-0.5}\delta^{-1}\epsilon^{-3})$.
Furthermore, DOC²S with local stochastic zeroth-order oracle also achieves the $(\delta,\epsilon)$-Goldstein staionary points with sample complexity of $\mathcal{O}(d\delta^{-1}\epsilon^{-3})$, the computation bounds of $\mathcal{O}(d\delta^{-1}\epsilon^{-3})$ and the communication bounds of $\mathcal{O}(d\gamma^{-0.5}\delta^{-1}\epsilon^{-3})$. These upper bounds are sharper than existing methods.

The paper also conducts numerical experiments on two different models, which support the shaper upper bounds of the proposed methods compared to other methods.

**Strengths:**

1. The paper solves the decentralized nonsmooth nonconvex optimization problem to a sharper upper bound than existing methods.
2. It incorporates the steps of client sampling and Chebyshev acceleration into the framework of online-to-nonconvex conversion, which does not require all clients accessing their local oracle in per computation rounds, and thus reduces the computation cost.
3. The experimental results support the theoretical findings.

**Weaknesses:**

1. The numerical experiments are limited to only two models. More experiments on different models and more recent datasets would strengthen the empirical validation of the proposed method.

**Questions:**

1. In line 145, it may be "zeroth-order" instead of "first-order" , since the discussion is about LSZO.
2. Can you provide additional experiments on newer datasets and larger-scale tasks to further validate the effectiveness and scalability of the proposed method?
3. Is the performance sensitive to hyperparameters like step size? In the theoretical results, the step size needs to satisfy the rate of $\mathcal{O}(\delta \epsilon^3)$, but in the numerical experiments, it seems that a fixed step size is used. Can you add more discussion or ablation study on this?

---

> ### Author Response · Authors · 2025-11-21
> **Authors' Response to Reviewer HZ98**
>
> We thank the reviewer HZ98 for the detailed and insightful comments.
>
> **Q1:**
> In line 145, it may be "zeroth-order" instead of "first-order" , since the discussion is about LSZO.
>
> **A1:**
> Thank you for your careful review. We have corrected this typo in our rebuttal revision.
>
> **Q2:**
> The numerical experiments are limited to only two models. More experiments on different models and more recent datasets would strengthen the empirical validation of the proposed method.
> Can you provide additional experiments on newer datasets and larger-scale tasks to further validate the effectiveness and scalability of the proposed method?
>
> **A2:**
> We have additionally conducted applications of image classification on the larger-scale datasets ''CIFAR-10" (60,000 images, 10 classes) and "CIFAR-100" (60,000 images, 100 classes).
> Each image of these datasets consists of $32 \times 32$ RGB pixels, i.e., the feature dimension is $32 \times 32 \times 3$.
> We adopt the standard ResNet-18 architecture citation, which is composed of an initial convolutional layer, followed by four stages of "Residual Blocks" (totaling 17 convolutional layers) that utilize skip connections to enable effective training of deep networks.
>
> We perform our numerical experiments on $n = 16$ clients associated with the network of the ring topology.
> For DOC$^2$S and ME-DOL, we tune the stepsize $\eta$ and diameter $D$ from $\lbrace{0.01, 0.05, 0.1}\rbrace$ and $\lbrace{0.05, 0.01, 0.005, 0.001}\rbrace$, respectively.
> For DGFM and DGFM+, we tune the stepsize $\eta$ from $\lbrace{0.001, 0.005, 0.01}\rbrace$.
> Additionally, we set the iteration number of Chebyshev acceleration as $R=2$ in our DOC$^2$S.
> We have conduct the empirical results in appendix of our rebuttal revision **(pages 29–30, Figures 9–12)**.
>
> We can observe that the proposed DOC$^2$S outperforms
> than baselines with respect to all measures (sample complexity,  computation rounds, and communication rounds).
> In particular, the client sampling strategy makes
> the sample complexity of our method be significantly superior to that of baseline methods.
>
> **Q3:**
> Is the performance sensitive to hyperparameters like step size? In the theoretical results, the step size needs to satisfy the rate of $\mathcal{O}(\delta\epsilon^3)$
> , but in the numerical experiments, it seems that a fixed step size is used. Can you add more discussion or ablation study on this?
>
> **A3:**
> We provide additional experiments in appendix of our rebuttal revision (page 31, Figures 13–14) to study the performance sensitive to step size.
> Specifically, we fix the radius parameter $D=0.01$ and set $\eta=0.05, 0.01, 0.1$, respectively.
> The experimental results demonstrate that an excessively large $\eta$ exhibits a faster descent at early state, while it does not exhibit good performance finally.
>
> Although the step size needs to satisfy the order of $\mathcal{O}(\delta\epsilon^3)$ in theoretical, we typically do not require the values of $\delta$ and $\epsilon$ to be very small in practice. Therefore, we tune the step size $\eta$ from $\lbrace{0.05, 0.01, 0.1}\rbrace$ in our experiments and the empirical results show our algorithm performs well.

---

### Official Review · Reviewer_4RA4 · 2025-11-04

**Soundness:** 3
**Presentation:** 3
**Contribution:** 3
**Rating:** 6
**Confidence:** 4

**Summary:**

This paper proposes a new algorithm for decentralized stochastic non-smooth non-convex optimization using both first-order and zeroth-order oracles. The method uses an online-to-non-convex approach introduced by Cutkosky et al., 2023, as well as a stochastic smoothing technique for non-smooth functions. The authors provide theoretical guarantees for these results, as well as experiments that validate the theoretical findings.

**Strengths:**

1. The proposed results have theoretical guarantees

2. The proposed algorithm has significantly better performance compared to existing algorithms (see Table 1).

**Weaknesses:**

**Minor comments:**

P.18 line 966. $g_i^{k,t}$ is not defined in Alg 1 and Theorem 1, only $g_{i^t}^{k,t}$ is defined

P.23 line 1231. Can you explain why you assume $\delta < 1$? I think we can't select $\delta$.

**Typos:**

P.19 line 978 $y_i^{k, t-1/2} \to y_i^{k, t-1}$

P.19 line 984 $2n \to n$

P.19 line 1024 $x^{k,t-1} \to \overline{x}^{k, t-1} $

P.24 line 1262. $\sqrt{T} \to 1$

P.24 line 1295. $\nu + L \to \nu + L\delta$,  also for th 3

P.25 line 1319. $4 \to 8$

P.25 line 1319. $k \to c_0$

P.25 line 1327. $D - \epsilon’ \leq 2/3 \to D/(D - \epsilon’)  \leq 3/2$,  also for th 3

**Questions:**

I provided questions in minor comments

---

> ### Author Response · Authors · 2025-11-21
> **Authors' Response to Reviewer 4RA4**
>
> We thank the reviewer 4RA4 for the detailed and insightful comments.
>
> **Q1:**  P.18 line 966. $ \mathbf{g}\_{i}^{k,t} $ is not defined in Alg 1 and Theorem 1, only $  \mathbf{g}\_{i^t}^{k,t} $ is defined.
>
> **A1:** We use $\mathbf{g}\_{i}^{k,t}$ to denote the output of the First-Order-Estimator/Zeroth-Order-Estimator in line 14 of Algorithm 1 when $i^t=i$. We have highlighted the explanation for this notation in our rebuttal revision.
>
> **Q2:** P.23 line 1231. Can you explain why you assume $ \delta < 1 $? I think we can't select $ \delta$.
>
> **A2:** Recall that the notion of $(\delta,\epsilon)$-Goldstein stationarity is used to characterize the property of a point in its **small local neighbors** (with radius $\delta$). Therefore, we are mainly interested in the case of small $\delta$.
> In fact, it is very common to suppose such small quantity is less than some constant is the complexity analysis.
> For example, Sahinoglu \& Shahrampour (2024, Theorem 3) and
> Guy Kornowski \& Ohad Shamir (2024, Theorems 5 and 6) also suppose $\delta<1$ for the analysis of nonconvex nonsmooth optimization.
>
> **Q3:** Typos:
>
> P.19 line 978 $ \mathbf{y}\_i^{k,t-1/2} \rightarrow \mathbf{y}\_i^{k,t-1} $
>
> P.19 line 984 $ 2n \rightarrow n $
>
> P.19 line 1024 $ \mathbf{x}^{k,t-1} \rightarrow \bar{\bf{{x}}}^{k,t-1} $
>
> P.24 line 1262 $ \sqrt{T} \rightarrow 1 $
>
> P.24 line 1295 $ \nu + L \rightarrow \nu + L\delta $, also for th 3
>
> P.25 line 1319 $ 4 \rightarrow 8 $
>
> P.25 line 1319 $ k \rightarrow c_0 $
>
> P.25 line 1327 $ D - \epsilon' \leq 2/3 \rightarrow D/(D - \epsilon') \leq 3/2 $, also for th 3
>
> **A3:**
> We thank the reviewer 4RA4 for the detailed and helpful comments.
> We have address above issues as follows.
>
> **P.19 line 984:** We confirm that the term $2n$ is correct. Note that we have
> \begin{align*}
> & \frac{H}{n} \sum_{i=1}^{n} \int_{0}^{1}\|\|{\bar{\mathbf{y}}^{k,t-1} + s\mathbf{\Delta}\_{i^t}^{k,t-1/2} - \mathbf{y}\_{i}^{k,t-1} - s \mathbf{\Delta}\_{i}^{k,t-1/2}}\|\|\_2  {\rm d}s \\\\
> \le& \frac{H}{n} \sum_{i=1}^{n} \|\|{\bar{\mathbf{y}}^{k,t-1} - \mathbf{y}\_{i}^{k,t-1}}\|\|\_2 + \frac{H}{n} \sum_{i=1}^{n} \|\|{\mathbf{\Delta}\_{i^t}^{k,t-1/2} - \mathbf{\Delta}\_{i}^{k,t-1/2}}\|\|\_2 \int_{0}^{1} s{\rm d}s \\\\
> =& \frac{H}{n} \sum_{i=1}^{n} \|\|{\bar{\mathbf{y}}^{k,t-1} - \mathbf{y}\_{i}^{k,t-1}}\|\|\_2 + \frac{H}{2n} \sum_{i=1}^{n} \|\|{\mathbf{\Delta}\_{i^t}^{k,t-1/2} - \mathbf{\Delta}\_{i}^{k,t-1/2}}\|\|\_2,
> \end{align*}
> where the first step is based on triangle inequality and the second step is based on the fact $\int_{0}^{1} s{\rm d}s=\frac{1}{2}$.
>
> **P.24 line 1295:** We confirm this step is correct because $\nu + L\delta \le \nu + L$, which is based on the assumption $\delta\in (0,1)$.
>
> We have fixed the typos you mentioned in the other points and highlighted the modifications in our rebuttal revision.

---

### Author Response · Authors · 2025-11-27

Dear Area Chair and Reviewers,

We sincerely thank you for your time and effort to review our paper. We have addressed the questions raised by the reviewers and revised our submission. However, we have not received any further response from the reviewers.

We would be very appreciated if the reviewers could participate in the discussion, which would help us improve our manuscript.

Best regards,

Authors

---

### Author Response · Authors · 2025-11-30
**Summary of Rebuttal Responses**

We greatly thank area chairs and reviewers for their careful reading and insightful comments.  We thank all the reviewers for their appreciation on our work, e.g.,
* The proposed algorithm has significantly better performance compared to existing algorithms (see Table 1);
* The proposed algorithm does not require all clients accessing their local oracle in per computation rounds, and thus reduces the computation cost;
* The experimental results support the theoretical findings;
* Achieves optimal sample complexity and sharper communication bounds than other state-of-the-art methods;
* Theoretical results are well-grounded and consistent with known lower bounds;
* The numerical experiment shows significant improvement in terms of sample and computation complexity.

We have addressed the reviewers' concerns in our responses and revised our submission by highlighting the modification in blue.
Due to the discussion period has been cut short, we have only received the feedback from Reviewer DtU4.

We summarize the main points in our response as follows.

**1. Reviewer HZ98 (Q2) suggested strengthening the empirical validation.**

We have additionally conducted the experiments for image classification on the larger-scale datasets "CIFAR-10" (60,000 images, 10 classes) and "CIFAR-100" (60,000 images, 100 classes).
The empirical results in appendix of our rebuttal revision **(pages 29–30, Figures 9–12)** shows that the proposed DOC$^2$S outperforms than baselines with respect to all measures (sample complexity, computation rounds, and communication rounds). In particular, the client sampling strategy makes the sample complexity of our method be significantly superior to that of baseline methods, which supports our theoretical analysis.

**2. Reviewer HZ98 (Q3) was confused about the setting of step size in the nonsmooth nonconvex optimization.**

Although the step size needs to satisfy the order of $\mathcal{O}(\delta\epsilon^3)$ in theoretical, we typically do not require the values of $\delta$ and $\epsilon$ to be very small in practice. Thus, we empirically tune $\eta$ from $\lbrace0.05, 0.01, 0.1\rbrace$, and our algorithm performs well with these choices.

Additional experiments on step size sensitivity are in the appendix (page 31, Figs. 13-14). With a fixed radius $D=0.01$, we test $\eta=0.05, 0.01, 0.1$. Results show that while a large $\eta$ descends faster initially, it leads to worse final performance.

**3. Reviewer 27Yq (Q2) did not aware that our method can address the heterogeneity.**

We have emphasized that our work does address **the heterogeneous setting**, since we assume the local function on the $i$th client has the form of $f_i(\mathbf{x}) \triangleq \mathbb{E}_{\mathbf{\xi}_i \sim \mathcal{D}_i}[F_i(\mathbf{x}; \mathbf{\xi}_i)]$, where $\mathbf{\xi}_i$ follows distribution $\mathcal{D}_i$ (see Equation (2)). It is worth noting that we do **not** require distributions $\mathcal{D}_1,\dots,\mathcal{D}_n$ be identical in our algorithm design and theoretical analysis.

**4.
Reviewer DtU4 (Q1 and Q11) was confused about why the algorithm requires performing gossip on $y_i^{k, t}$ and $\Delta_i^{k, t+1/2}$.**

We use FastGossip with $R=\mathcal{O}(\gamma^{-1/2}\log(nD\epsilon'^{-1}))=\tilde{\mathcal{O}}(\gamma^{-1/2})$ to achieve the result of Lemmas 2 and 3, i.e.,
$$\|\|\mathbf{\Delta}\_i^{k,t+1/2}-\bar{\mathbf{\Delta}}^{k,t+1/2}\|\|\le \epsilon',
\quad\|\|\mathbf{\Delta}\_i^{k,t+1/2}\|\|\le D+\epsilon'
\quad\text{ and }\quad\|\|\mathbf{y}\_i^{k,t}-\bar{\mathbf{y}}^{k,t}\|\|\le \frac{(D+\epsilon')\epsilon'}{D-\epsilon'}$$
for all $\epsilon'>0$, where the notation $\tilde{\mathcal{O}}(\cdot)$ omits the logarithmic factors in the complexity.
In later analysis, we apply the above inequality with sufficient small $\epsilon'$ (lines 1260-1264) such that
\begin{align*}
    \epsilon' \leq \min\lbrace{\frac{(\left\lceil \frac{9h_1^2}{\epsilon^2} \right\rceil - 6)\delta}{12 \left( \left\lceil \frac{9h_1^2}{\epsilon^2} \right\rceil \right)^2 + 24 \left\lceil \frac{9h_1^2}{\epsilon^2} \right\rceil}
, \left( 9H + \frac{4(G + L)\left\lceil \frac{9h_1^2}{\epsilon^2} \right\rceil}{\delta} + \frac{{10G\left\lceil \frac{9h_1^2}{\epsilon^2} \right\rceil^{3/2}}}{\delta} \right)^{-1}
    \frac{\epsilon}{3}\rbrace}.
\end{align*}
to guarantee that the last term of equation (48) in lines 1239–1240 is no larger than $\mathcal{O}(\epsilon)$ , i.e.,
\begin{align*}
    \left(\frac{2DH}{D-\epsilon'}+\frac{G+L}{D}+\frac{5G\sqrt{T}}{2D}+6H\right)\epsilon' \leq \mathcal{O}(\epsilon).
\end{align*}

**5. Reviewer DtU4 (Q5) was confused about why we provide the result for the smooth problem:**

We have explained that the main idea of our proof is **reducing the nonsmooth problem to the smooth one**. Therefore, we first establish the result for finding the approximate Goldstein-stationary point of the smooth surrogate function (Theorem 1 for Algorithm 3 with $\mu=0$) before achieving our main result for the nonconvex case.

---

### Meta-Review · Area_Chair_xsRk · 2026-01-04

**Summary:**

This paper develops a decentralized algorithm for nonsmooth nonconvex optimization. The proposed method combines several existing techniques, such as FastGossip and client sampling, which have been well studied, to improve the convergence rate of ME-DOL. The paper also establishes convergence rates for the proposed algorithm and validates its performance on several machine learning models.

The concerns raised include limited evaluation, inappropriate hyperparameter settings, and unclear explanations of algorithmic details. Some of these issues were addressed during the rebuttal phase. For example, the authors provided additional experiments on larger-scale datasets. However, several concerns remain unresolved. In particular, the paper claims that client sampling is an advantage of the proposed algorithm, yet the FastGossip step still requires the participation of all clients, which contradicts this claim. Moreover, the hyperparameter settings used in the experiments are not aligned with the theoretical analysis, failing to validate the theoretical results. For instance, in Theorems 1 and 3, the radius $D$ has a lower-order dependence on $\epsilon$ than the learning rate  $\eta$, implying that the learning rate should be smaller than the radius. However, in the experiments, the learning rate is set larger than the radius, undermining the empirical validation of the theoretical claims.

Considering that some key concerns remain unresolved, I recommend rejection.

**Reviewer Concerns:**

The concerns raised include limited evaluation, inappropriate hyperparameter settings, and unclear explanations of algorithmic details. Some of these issues were addressed during the rebuttal phase. For example, the authors provided additional experiments on larger-scale datasets.

However, several concerns remain unresolved. In particular, the paper claims that client sampling is an advantage of the proposed algorithm, yet the FastGossip step still requires the participation of all clients, which contradicts this claim. Moreover, the hyperparameter settings used in the experiments are not aligned with the theoretical analysis, failing to validate the theoretical results. For instance, in Theorems 1 and 3, the radius $D$ has a lower-order dependence on $\epsilon$ than the learning rate  $\eta$, implying that the learning rate should be smaller than the radius. However, in the experiments, the learning rate is set larger than the radius, undermining the empirical validation of the theoretical claims.

**Reviewer Scores:**

The key concerns of the two negative reviewers remain unresolved, so the initial scores would be kept.

---

### Decision · Program_Chairs · 2026-01-26

Reject